# Hessian-guided Perturbed Wasserstein Gradient Flows for Escaping Saddle Points

**Naoya Yamamoto**
The University of Tokyo
yamamoto-naoya251@g.ecc.u-tokyo.ac.jp

**Juno Kim**[*]
UC Berkeley
junokim@berkeley.edu

**Taiji Suzuki**
The University of Tokyo, RIKEN AIP
taiji@mist.i.u-tokyo.ac.jp

## Abstract

Wasserstein gradient flow (WGF) is a common method to perform optimization over the space of probability measures. While WGF is guaranteed to converge to a first-order stationary point, for nonconvex functionals the converged solution does not necessarily satisfy the second-order optimality condition; i.e., it could converge to a saddle point. In this work, we propose a new algorithm for probability measure optimization, *perturbed Wasserstein gradient flow* (PWGF), that achieves second-order optimality for general nonconvex objectives. PWGF enhances WGF by injecting noisy perturbations near saddle points via a Gaussian process-based scheme. By pushing the measure forward along a random vector field generated from a Gaussian process, PWGF helps the solution escape saddle points efficiently by perturbing the solution towards the smallest eigenvalue direction of the Wasserstein Hessian. We theoretically derive the computational complexity for PWGF to achieve a second-order stationary point. Furthermore, we prove that PWGF converges to a global optimum in polynomial time for strictly benign objectives.

## 1 Introduction

We consider the general problem of optimizing a probability measure: $\min_{\mu \in \mathcal{P}_2(\mathbb{R}^d)} F(\mu)$, where $\mathcal{P}_2(\mathbb{R}^d)$ denotes the space of Borel probability measures on $\mathbb{R}^d$ with finite second moment and $F : \mathcal{P}_2(\mathbb{R}^d) \to \mathbb{R}$ is a given functional which is not necessarily convex. Optimization of measures appears extensively in machine learning and statistics either explicitly or implicitly, such as in sampling and variational inference (Jordan et al., 1998; Liu & Wang, 2016), generative models (Arbel et al., 2019; Chu et al., 2019) and training neural networks (Mei et al., 2018; Nitanda et al., 2022), and has garnered significant attention both theoretically and practically. To solve such problems, the **Wasserstein gradient flow** (WGF) is extensively employed:

$$\partial_t \mu_t + \nabla \cdot \left( -\nabla \frac{\delta F}{\delta \mu}(\mu_t) \mu_t \right) = 0, \quad (\mu_t) \subset \mathcal{P}_2(\mathbb{R}^d). \tag{1}$$

WGF is the continuity equation of the velocity field $\nabla \frac{\delta F}{\delta \mu}$ and can be interpreted as gradient descent with respect to the 2-Wasserstein metric (Jordan et al., 1998). Moreover, (1) is equivalent to the continuous-time and infinite-particle limit of first-order optimization algorithms such as gradient descent, and serves as the foundation for numerous machine learning methods; see Appendix A for a discussion of related works and applications. It is thus of paramount importance to understand the convergence of WGF, and to develop algorithms which guarantee well-behaved solutions.

---

[*]This work was primarily conducted while the author was at the University of Tokyo and RIKEN AIP.

**Convergence of WGF.**   A prominent line of work on the convergence of WGF is the study of neural network optimization using *mean-field theory* (Mei et al., 2018). Mean-field theory models the evolution of an interacting particle system as the number of particles tends to infinity, such as the training dynamics of an infinite-width neural network, through the WGF of the limiting distribution. For simple problem settings such as regression with two-layer networks, the corresponding loss functional is shown to be (linearly) convex, allowing for global convergence analysis of WGF (Chizat & Bach, 2018; Mei et al., 2018). Furthermore, Nitanda et al. (2022); Chizat (2022); Suzuki et al. (2023) obtained linear convergence under the log-Sobolev inequality (LSI) condition using mean-field Langevin dynamics. This adds isotropic noise via Brownian motion, corresponding to an additional entropy regularization which effectively makes the objective strongly convex.

**Non-convex objectives.**   While these works primarily rely on convexity, the vast majority of objectives arising in deep learning (such as the loss function for neural networks with three or more layers) are non-convex, even when lifted to the space of measures. However, due to the inherent difficulty of infinite-dimensional non-convex optimization, convergence guarantees for WGF in this regime has been extremely limited. Recently, Kim & Suzuki (2024) studied a transformer model combining a mean-field neural network with linear attention for in-context learning, resulting in a non-convex optimization problem. Through landscape analysis, they showed that the objective possesses a desirable benignity property: all second-order optima are either saddle points or global optima. A similar result has been demonstrated for energy kernels such as the MMD functional (Boufadene & Vialard, 2024). These findings suggest the necessity of probability measure optimization algorithms that account for second-order optimality. While the first-order optimality of WGF has been shown by Lanzetti et al. (2025), there are few studies that rigorously consider second-order conditions.

**Perturbation methods.**   The main difficulty of non-convex optimization is due to the existence of saddle points. A first-order method with naive initialization may end up converging to a saddle point. Moreover, Du et al. (2017) showed that gradient descent can be significantly delayed near saddle points, taking exponential time to converge. For optimization in finite-dimensional Euclidean space, Ge et al. (2015); Jin et al. (2017); Li (2019) proposed methods to overcome this issue by introducing perturbations in the vicinity of saddle points to 'fall off' the saddle and escape efficiently. These methods, often referred to as **perturbed gradient descent** (PGD), are particularly useful as they achieve second-order optimality in polynomial time while primarily relying on first-order techniques. In particular, PGD guarantees global convergence for problems satisfying a strict benignity property, such as matrix factorization (Jin et al., 2017) and tensor decomposition (Ge et al., 2015).

With these issues in mind, we ask the following question:

*Can we develop a perturbative version of Wasserstein gradient flow for non-convex objectives which converges efficiently to second-order optimal points?*

**Our Contributions.**   In this study, we propose a perturbative modification to WGF that efficiently avoids saddle points. Considering the tangent bundle structure of $\mathcal{P}_2(\mathbb{R}^d)$ induced by the Wasserstein distance, it is natural to extend the notion of perturbation in Euclidean space to measure space by defining perturbations of the drift function through randomly generated vector fields. Such a perturbative method was conjectured to improve convergence by Kim & Suzuki (2024), but without any theoretical guarantees. Our contributions are summarized as follows:

- We propose a new implementable algorithm, **perturbed Wasserstein gradient flow (PWGF)**, that guarantees second-order optimality for general smooth and non-convex functionals. Unlike the method proposed by Kim & Suzuki (2024), which injects isotropic noise near saddles into the WGF, we guarantee improvement by *directing* the noise using the (Wasserstein) Hessian. Specifically, PWGF pushes the measure along a random velocity field generated from a Gaussian process whose covariance is constructed from the Hessian of the objective.

- We prove that PWGF effectively avoids saddle points and reaches second-order optimal points in time that depends polynomially on precision parameters, enabling the optimization of non-convex distributional objectives. Compared to the finite-dimensional setting (Li, 2019), the analysis requires a careful treatment of an infinite dimensional objective. For this purpose, we utilize techniques from Wasserstein geometry, optimal transport and the theory of Gaussian processes.

**Organization.** The paper is organized as follows. Section 2 provides theoretical preliminaries, supplemented in Appendix B. Second-order optimality conditions are presented in Section 3. In Section 4, we introduce the proposed PWGF algorithm. Section 5 presents the convergence analysis, along with a rough sketch of the proof. Numerical experiments are provided in Appendix F.

## 2 Preliminaries

In this paper, we consider the optimization problem $\min_{\mu \in \mathcal{P}_2(\mathbb{R}^d)} F(\mu)$ over the space of probability distributions, where $F : \mathcal{P}(\mathbb{R}^d) \to \mathbb{R}$ is a real-valued lower-bounded functional defined on $\mathcal{P}_2(\mathbb{R}^d)$. This section introduces the problem of probability measure optimization and reviews key concepts in optimal transport and Wasserstein geometry. See Appendix B for further details.

**Notation.** Let Id be the identity map on $\mathbb{R}^d$. The canonical projection to the $i$th coordinate is denoted by $p_i$. The Euclidean inner product and operator norm are $\langle \cdot, \cdot \rangle$, $\|\cdot\|$. The Frobenius norm is $\|\cdot\|_{\mathrm{F}}$. The set of real-valued functions on $\mathbb{R}^d$ that are infinitely differentiable with compact support is $C_0^\infty(\mathbb{R}^d)$. The inner product and norm or operator norm in $L^2(\mu)^d$ is $\langle \cdot, \cdot \rangle_{L^2(\mu)}$, $\|\cdot\|_{L^2(\mu)}$. The trace norm of an operator is $\|\cdot\|_{\mathrm{Tr}, L^2(\mu)}$, and the Hilbert-Schmidt norm is $\|\cdot\|_{\mathrm{HS}, L^2(\mu)}$. $T \succeq O$ indicates that an operator $T$ is positive semi-definite. The smallest eigenvalue of a compact operator $T$ is denoted by $\lambda_{\min} T$. The exponential of an operator $T$ is denoted by $e^T$ or $\exp T$. We use $\tilde{O}$ to denote big $O$ notation ignoring logarithmic factors.

The set of all Borel probability measures on $\mathbb{R}^d$ with finite second moments is denoted by $\mathcal{P}_2(\mathbb{R}^d)$, and the subset of measures absolutely continuous with respect to Lebesgue measure is denoted by $\mathcal{P}_2^a(\mathbb{R}^d)$. The Dirac measure on $x \in \mathbb{R}^d$ is $\delta_x \in \mathcal{P}_2(\mathbb{R}^d)$. $f \# \mu$ denotes the pushforward of $\mu \in \mathcal{P}_2(\mathbb{R}^d)$ by a measurable map $f : \mathbb{R}^d \to \mathbb{R}^d$.

### 2.1 Wasserstein Geometry

**Definition 2.1** (Wasserstein metric). The 2-Wasserstein metric between $\mu, \nu \in \mathcal{P}_2(\mathbb{R}^d)$ is defined as

$$W_2(\mu, \nu)^2 := \min_{\gamma \in \Gamma(\mu, \nu)} \int \|x - y\|^2 \gamma(\mathrm{d}x \mathrm{d}y), \tag{2}$$

where $\Gamma(\mu, \nu)$ represents the set of all transport plans (or the set of couplings) of $\mu, \nu$, that is, all joint distributions on $\mathbb{R}^d \times \mathbb{R}^d$ whose marginal distributions are $\mu$ and $\nu$. We denote the set of all optimal transport plans by $\Gamma_o(\mu, \nu)$ and the optimal transport map by $\mathcal{T}_\mu^\nu$.

A fundamental dynamics in Wasserstein space is the continuity equation with velocity field $v_t$,

$$\partial_t \mu_t + \nabla \cdot (v_t \mu_t) = 0, \quad \mu_t \in \mathcal{P}_2(\mathbb{R}^d), \ t \in I. \tag{3}$$

Intuitively, (3) describes how a particle distribution $\mu_t$ evolves along a vector field $v_t$. In particular, WGF (1) moves particles according to the vector field $v_t = -\nabla \frac{\delta F}{\delta \mu}(\mu_t)$ (see Section 2.3). Moreover, the continuity equation with velocity field $v_t$ such that

$$v_t \in \mathrm{Tan}_{\mu_t} \mathcal{P}_2(\mathbb{R}^d) := \overline{\{\nabla \varphi \mid \varphi \in C_0^\infty(\mathbb{R}^d)\}}^{L^2(\mu_t)}.$$

can be locally approximated by a pushforward along $v_t$ (Proposition B.9). Therefore the continuity equation is computationally approximated by the pushforward representation:

$$\mu_{t+\Delta t} \leftarrow (\mathrm{Id} + \Delta t v_t) \# \mu_t. \tag{4}$$

The absolute continuity of the curve $\mu_t$ with respect to the Wasserstein distance is equivalent to satisfying (3) for some $v_t \in L^2(\mu_t)$ (Ambrosio et al. (2008), Theorem 8.3.1). In this sense, the continuity equation provides a concept of differentiation consistent with the Wasserstein metric. For further background on optimal transport theory, see Appendix B.

### 2.2 Wasserstein Gradient

The Wasserstein gradient is the fundamental quantity for first-order analysis in Wasserstein space.

**Definition 2.2** (Wasserstein gradient). The Wasserstein gradient of $F$ at $\mu \in \mathcal{P}_2(\mathbb{R}^d)$ is a vector field $\nabla_\mu F : \mathbb{R}^d \to \mathbb{R}^d$ such that for any $\nu \in \mathcal{P}_2(\mathbb{R}^d)$ and $\gamma \in \Gamma_o(\mu, \nu)$,

$$F(\nu) - F(\mu) = \int \nabla_\mu F(x)^\top (y - x) \gamma(\mathrm{d}x\mathrm{d}y) + O\big(W_2(\mu, \nu)^2\big).$$

The first variation also frequently appears in the context of probability measure optimization (cf. proximal Gibbs measure (Nitanda et al., 2022)).

**Definition 2.3** (First variation). The first variation $\frac{\delta F}{\delta \mu} : \mathcal{P}_2(\mathbb{R}^d) \times \mathbb{R}^d \to \mathbb{R}$ is defined as a functional satisfying for any $\nu \in \mathcal{P}_2(\mathbb{R}^d)$,

$$\frac{\mathrm{d}}{\mathrm{d}h}\bigg|_{h=0} F(\mu + h(\nu - \mu)) = \int \frac{\delta F}{\delta \mu}(\mu, x)(\mu - \nu)(\mathrm{d}x). \tag{5}$$

A naive computation might suggest that $\nabla_\mu F = \nabla \frac{\delta F}{\delta \mu}$; however, this is not generally true without additional conditions. Nevertheless, we do not distinguish between the two, see Appendix B.2.

**First-order optimality.** The Wasserstein gradient allows us to construct first-order approximations of functionals. Furthermore, Lanzetti et al. (2025) demonstrated two analogies to finite-dimensional optimization. The first is that $\nabla_\mu F = 0$ serves as a necessary condition for local optimality. The second is that $\nabla_\mu F = 0$ becomes a sufficient condition for global optimality in the convex case. Based on these considerations, we define the following.

**Definition 2.4** (First-order stationary point). Supoose that a functional $F : \mathcal{P}_2(\mathbb{R}^d) \to \mathbb{R}$ satisfies sufficient smoothness. We say that $\mu \in \mathcal{P}_2(\mathbb{R}^d)$ is a *first-order stationary point*, if $\mu$ satisfies $\nabla_\mu F = 0$ $\mu$-a.e.

### 2.3 Wasserstein Gradient Flow (WGF)

As a counterpart of gradient descent in Euclidean space, the WGF in Wasserstein space is defined as

$$\partial_t \mu_t + \nabla \cdot (-\nabla_\mu F(\mu_t)\mu_t) = 0, \quad (\mu_t) \subset \mathcal{P}_2(\mathbb{R}^d). \tag{6}$$

From previous observations, the direction $v_t = -\nabla_\mu F(\mu_t)$ yields the steepest descent direction of the objective $F$. Furthermore, by the chain rule (Proposition C.3) it holds that

$$\frac{\mathrm{d}}{\mathrm{d}t}F(\mu_t) = -\|\nabla_\mu F(\mu_t)\|^2_{L^2(\mu_t)} \leq 0$$

This indicates that the WGF monotonically decreases the objective function. Indeed, the WGF can be interpreted as a gradient descent method in the space of probability measures (Jordan et al., 1998). Moreover, WGF becomes stationary iff the solution is at a first-order stationary point (Definition 2.4).

## 3 Second Order Optimality on Probability Space

In order to study second-order behavior, we define a suitable class of sufficiently regular functionals.

**Definition 3.1** (Sufficient smoothness). A functional $F : \mathcal{P}_2(\mathbb{R}^d) \to \mathbb{R}$ is *sufficiently smooth* if

- $F$ admits a $L^2(\mu)$-integrable Wasserstein gradient $\nabla_\mu F$ at all $\mu \in \mathcal{P}_2(\mathbb{R}^d)$.

- $\nabla_\mu F(\mu, x)$ further admits Wasserstein gradient $\nabla^2_\mu F : \mathcal{P}_2(\mathbb{R}^d) \times \mathbb{R}^d \times \mathbb{R}^d \to \mathbb{R}^{d \times d}$ and is differentiable with respect to the second coordinate $x$ for any $\mu \in \mathcal{P}_2(\mathbb{R}^d)$ and $\mu$-a.e. $x$. Furthermore, $\nabla^2_\mu F(\mu)$ is $L^2(\mu \otimes \mu)$ integrable and $\mu$-ess $\sup\|\nabla\nabla_\mu F(\mu, x)\| < \infty$.

**Assumption 1.** *The objective $F : \mathcal{P}_2(\mathbb{R}^d) \to \mathbb{R}$ is a sufficiently smooth functional.*

Building upon the discussion of first-order optimality, we extend the analysis to second-order conditions. For simplicity of notation, we define the following operators $H_\mu, H'_\mu$ for $f \in L^2(\mathbb{R}^d)^d$:

$$H_\mu f(x) = \int \nabla^2_\mu F(\mu, x, y)f(y)\mu(\mathrm{d}y), \quad H'_\mu f(x) = \nabla\nabla_\mu F(\mu, x)f(x).$$

We establish the following proposition.

**Proposition 3.2.** *For $F : \mathcal{P}_2(\mathbb{R}^d) \to \mathbb{R}$ a sufficiently smooth functional, for all $v \in L^2(\mu)^d$,*

$$\frac{\mathrm{d}^2}{\mathrm{d}h^2}\bigg|_{h=0} F((\mathrm{Id} + hv)\#\mu) = \left\langle v, (H_\mu + H'_\mu)v \right\rangle_{L^2(\mu)}. \tag{7}$$

This demonstrates that the second order term of a vector field perturbation is characterized by the operator $H_\mu + H'_\mu$. For a more general version, see Proposition C.7. In particular, when examining stability at first-order stationary points, we have $\nabla_\mu F(\mu) = 0$, which implies $\nabla\nabla_\mu F(\mu) = 0$, that is, $H'_\mu = 0$. Consequently, the change in $F$ due to perturbations along vector fields is determined solely by the integral operator $H_\mu$ up to second order. Note that this does not necessarily hold for $\mu \notin \mathcal{P}_2^a(\mathbb{R}^d)$; for further details, refer to Appendix C.

Next, paralleling the work by Lanzetti et al. (2025), we demonstrate that $H_\mu \succeq O$ serves as a sufficient condition for local stability, establishing the analogy with second-order conditions in Euclidean space.

**Proposition 3.3** (second-order necessary condition). *Let $F : \mathcal{P}_2(\mathbb{R}^d) \to \mathbb{R}$ be sufficiently smooth. If $\mu^* \in \mathcal{P}_2^a(\mathbb{R}^d)$ is a local minimum of F, then it holds that $H_{\mu^*} \succeq O$.*

Based on this observation, we define the second-order optimality condition for probability measures.

**Definition 3.4** (second-order stationary point). *For $F : \mathcal{P}_2(\mathbb{R}^d) \to \mathbb{R}$ a sufficiently smooth functional and $\mu \in \mathcal{P}_2(\mathbb{R}^d)$,*

- We say that $\mu$ is a *second-order stationary point* if $\mu$ is a first-order stationary point and $H_\mu \succeq O$.

- We say that $\mu$ is a *saddle point* if $\mu$ is a first order stationary point and satisfies $H_\mu \not\succeq O$, i.e., the smallest eigenvalue of $H_\mu$ is strictly negative : $\lambda_{\min} H_\mu < 0$.

Since we seek approximate solutions for $F$, we also define *approximate* second order stationary points and saddle points.

**Definition 3.5** (approximate second-order stationary point). *Suppose that $F : \mathcal{P}_2(\mathbb{R}^d) \to \mathbb{R}$ is sufficiently smooth and $\mu \in \mathcal{P}_2(\mathbb{R}^d)$.*

- We say that $\mu$ is an $(\varepsilon, \delta)$-stationary point, if $\|\nabla_\mu F(\mu)\|_{L^2(\mu)} \leq \varepsilon$ and $\lambda_{\min} H_\mu \geq -\delta$.

- We say that $\mu$ is an $(\varepsilon, \delta)$-saddle point, if $\|\nabla_\mu F(\mu)\|_{L^2(\mu)} \leq \varepsilon$ and $\lambda_{\min} H_\mu < -\delta$.

While this definition is useful for convergence analysis, it is not fully appropriate in light of the second-order expansion (7). This is because the condition $\|\nabla_\mu F(\mu)\|_{L^2(\mu)} \leq \varepsilon$ only implies that $\nabla_\mu F(\mu)$ is small in the sense of $L^2$ norm and does not indicate that $\nabla\nabla_\mu F(\mu)$ is close to zero. Therefore, in that case, second order optimality should be determined by $H_\mu + H'_\mu$ rather than $H_\mu$. Consequently, this paper assumes that the $L^2$ norm of Wasserstein gradients being small implies that the supremum norm of their gradients is also small.

**Assumption 2.** *For any $\mu \in \mathcal{P}_2(\mathbb{R}^d)$, it holds that*

$$\mu\text{-}\operatorname*{ess\,sup}_{x \in \mathbb{R}^d} \|\nabla\nabla_\mu F(\mu, x)\| \leq R_2 \|\nabla_\mu F(\mu)\|_{L^2(\mu)}. \tag{8}$$

Under this assumption, a small $\|\nabla_\mu F(\mu)\|_{L^2(\mu)}$ implies that $\mu\text{-}\operatorname{ess\,sup}\|\nabla\nabla_\mu F(\mu, x)\| = \|H'_\mu\|_{L^2(\mu)}$ is also small, justifying the definition of $(\varepsilon, \delta)$-stationary points.

### 3.1 Global Convergence for Strictly Benign Objectives

It is known that certain non-convex optimization problems possess a desirable property called *benignity*, i.e. all local minima must be global minima. In Euclidean spaces, examples include tensor decomposition (Ge et al., 2015; Jin et al., 2017). With our definitions of (approximate) second-order optimality, a similar property can be considered for the Wasserstein space.

**Definition 3.6** (Strict benignity). *The functional $F : \mathcal{P}_2(\mathbb{R}^d) \to \mathbb{R}$ is said to be $(\varepsilon, \delta, \alpha)$-strictly benign if at least one of the following conditions holds for any $\mu \in \mathcal{P}_2(\mathbb{R}^d)$:*

   1. $\|\nabla_\mu F(\mu)\|_{L^2(\mu)} > \varepsilon$.

2. $\lambda_{\min} H_\mu < -\delta$.

3. $W_2(\mu, \mu^o) \leq \alpha$ for some global optima $\mu^o$.

We provide examples of non-convex objective functions that exhibit strict benignity. For details on the properties of each objective function, refer to the appendix and the cited papers.

**Example 1** (Matrix decomposition). This example is inspired by the fact that finite-dimensional tensor decomposition exhibits strict benignity. We use a mean-field two-layer neural network

$$h_\mu(z) = \int a\sigma(w^\top z)\mu(\mathrm{d}a\mathrm{d}w)$$

to learn a rank-one matrix induced by the target measure $\mu^o$, where $z$ is a data input that follows a certain distribution, the parameter is $x = (a, w) \in \mathbb{R}^{k+l}$, and $\sigma$ is an activation function, which we set as the sigmoid function. The objective functional can be expressed as follows:

$$F(\mu) = \mathrm{E}_z[\|h_{\mu^o}(z)h_{\mu^o}(z)^\top - h_\mu(z)h_\mu(z)^\top\|_{\mathrm{F}}^2]$$

We defer a detailed analysis of this objective to Appendix G.

**Example 2** (3-layer neural network). Consider a three-layer neural network model consisting of a mean-field two-layer network followed by a linear layer. Assuming realizability, the $L^2$ loss with respect to a teacher network $T^* h_{\mu^*}(z)$ can be written as follows:

$$\tilde{F}(\mu, T) = \mathrm{E}_z\left[\|T^* h_{\mu^*}(z) - T h_\mu(z)\|^2\right] \tag{9}$$

In this case, the optimization problem with respect to $T$ can be explicitly solved. Defining $\mu^o = (T^* \times \mathrm{Id}_{\mathbb{R}^l})\#\mu^*$ and $\Sigma_{\mu,\nu} = \mathrm{E}_z\left[h_\mu(z)h_\nu(z)^\top\right]$, the optimal $T$ satisfies $T = \Sigma_{\mu^o,\mu}\Sigma_{\mu,\mu}^{-1}$ and (9) reduces to the following probability measure optimization problem:

$$F(\mu) = \mathrm{E}_z\left[\|h_{\mu^o}(z) - \Sigma_{\mu^o,\mu}\Sigma_{\mu,\mu}^{-1}h_\mu(z)\|^2\right]. \tag{10}$$

Kim & Suzuki (2024) analyze this optimization problem and essentially show strict benignity, assuming $\Sigma_{\mu,\mu}$ is bounded away from degeneracy.

**Example 3** (Coulomb MMD). The maximum mean discrepancy (MMD) with Coulomb kernel can be expressed as:

$$F(\mu) = \int \frac{(\mu - \mu^o)^{\otimes 2}(\mathrm{d}x\mathrm{d}y)}{\|x - y\|^{d-2}}.$$

This energy functional has been studied by Boufadene & Vialard (2024), who showed that any absolutely continuous stationary point must be a global optimum. Therefore, $F$ becomes benign in regions that do not involve singular distributions.

## 4 Perturbed Wasserstein Gradient Flow

In this section, we introduce our proposed algorithm that incorporates perturbations in measure space along random velocity fields to escape saddle points.

### 4.1 Perturbations in Wasserstein Space

In Euclidean spaces, perturbed gradient descent (PGD) is a first-order optimization method capable of escaping saddle points efficiently by adding perturbations (Ge et al., 2015; Jin et al., 2017; Li, 2019). Despite relying solely on first order information, this method effectively achieves second order optimality, making it highly practical especially for problems with known benign structure. A typical perturbation technique in Euclidean spaces involves adding vectors sampled uniformly from a ball of small radius. Intuitively, such perturbations can uniformly explore all directions, making it likely to include the unstable direction corresponding to the smallest eigenvalue of the Hessian, thereby quickly 'falling off' the saddle.

To extend this idea to the space of probability measures, two issues must be addressed: **(1)** how to induce perturbations in Wasserstein space, and **(2)** whether the perturbation includes the unstable direction that maximally reduces the objective. For **(1)**, Kim & Suzuki (2024) proposed introducing

---

**Algorithm 1** PWGF (continuous-time)

---

    set hyperparameter $\eta_p = \tilde{O}\left(\delta^{\frac{3}{2}} \wedge \frac{\delta^3}{\varepsilon}\right)$, $T_{\text{thres}} = \tilde{O}\left(\frac{1}{\delta}\right)$, and $F_{\text{thres}} = \tilde{O}(\delta^3)$

    initialize $\mu^{(0)}$ and $t_p = -T_{\text{thres}}$

    **for** $t > 0$ **do**

        **if** $\|\nabla_\mu F(\mu_t)\|_{L^2(\mu_t)} \leq \varepsilon$ and $t - t_{\text{p}} > t_{\text{thres}}$ **then**

            $\xi \sim \text{GP}(0, K_{\mu_t})$, $\mu_t \leftarrow (\text{Id} + \eta_p \xi)\sharp \mu_t$, $t_{\text{p}} \leftarrow t$

        **end if**

        **if** $t = t_{\text{p}} + T_{\text{thres}}$ and $F(\mu_{t_{\text{p}}}) - F(\mu_t) \leq F_{\text{thres}}$ **then**

            **return** $\mu_{t_{\text{p}}}$

        **end if**

        $\partial_t \mu_t + \nabla \cdot (-\nabla_\mu F(\mu_t)\mu_t) = 0$

    **end for**

---

perturbations via a multivariate Gaussian process $\xi \sim \text{GP}(0, K)$ with a fixed positive semi-definite kernel $K : \mathbb{R}^d \times \mathbb{R}^d \to \mathbb{R}^{d \times d}$, transforming $\mu$ into $(\text{Id} + \eta \xi)\#\mu$. Leveraging vector fields to represent infinitesimal changes in probability measures is both natural and practical. However, they did not provide a theoretical justification for the effectiveness of this method.

We resolve this issue and guarantee **(2)** by constructing a *measure-dependent* kernel $K = K_\mu$ based on the Wasserstein Hessian of the objective:

$$K_\mu(x, y) = \int \nabla_\mu^2 F(\mu, x, z) \nabla_\mu^2 F(\mu, z, y) \mu(\mathrm{d}z). \tag{11}$$

The integral operator defined by this kernel coincides with the squared Hessian operator $H_\mu^2$. Specifically, it holds for $f \in L^2(\mu)^d$ that

$$\begin{aligned}
H_\mu^2 f(x) &:= \int \nabla_\mu^2 F(\mu, x, z) \left(\int \nabla_\mu^2 F(\mu, z, y) f(y) \mu(\mathrm{d}y)\right) \mu(\mathrm{d}z) \\
&= \int \left(\int \nabla_\mu^2 F(\mu, x, z) \nabla_\mu^2 F(\mu, z, y) \mu(\mathrm{d}z)\right) f(y) \mu(\mathrm{d}y) \\
&= \int K_\mu(x, y) f(y) \mu(\mathrm{d}y).
\end{aligned}$$

The Hessian-based kernel $K_\mu$ is symmetric, positive semi-definite, and meets the trace-class condition due to the integrability of $\nabla_\mu F(\mu)$. This means that $K_\mu$ satisfies the requirements for a Gaussian process kernel. More importantly, the Gaussian process $\xi$ is 'directed' by $H_\mu$, ensuring that the perturbation is likely to include the direction corresponding to the smallest eigenvalue of $H_\mu$, thereby achieving a reduction in the objective (Proposition 5.4). Details are provided in Appendix D.

### 4.2 Algorithm of PWGF

Based on the above considerations, we propose **perturbed Wasserstein gradient flow (PWGF)** as a method for solving non-convex optimization problems of probability measures. PWGF alternates between perturbing near saddle points using a Gaussian process with the kernel defined in (11) and evolving via WGF when not near saddle points.

We present the specific algorithm in Algorithm 1, including the saddle detection mechanism. In the space of probability measures, the Hessian $H_\mu$ is an operator on $L^2(\mu)^d$, and determining whether its smallest eigenvalue is below a certain threshold is computationally challenging. Drawing on Jin et al. (2017), we propose a practical criterion: Proposition 5.4 says that if a perturbation is introduced at an $(\varepsilon, \delta)$-saddle point, the objective function decreases with high probability after a certain period of WGF. Consequently, by always introducing perturbations at first-order stationary points, we can determine whether a given stationary point is a saddle point based on the decrease on the objective within a fixed time threshold.

The time-discretized version of PWGF is provided in Algorithm 2. To implement PWGF numerically, the optimal probability measure is approximated by the ensemble average of $N$ particles as $\mu = \frac{1}{N}\sum_{j=1}^N \delta_{x_j}$ and the pushforward and descent steps are directly applied to each particle as

$$\text{(perturbation)} \quad x_j \leftarrow x_j + \eta_p \cdot \xi(x_j),$$

$$\text{(gradient descent)} \quad x_j \leftarrow x_j - \eta \nabla F(\mu^{(k)}, x_j).$$

---

**Algorithm 2** PWGF (discrete-time)

---

set hyperparameter $\eta_p = \tilde{O}\left(\delta^{\frac{3}{2}} \wedge \frac{\delta^3}{\varepsilon}\right)$, $\eta = O(1)$, $k_{\text{thres}} = \tilde{O}\left(\frac{1}{\delta}\right)$, and $F_{\text{thres}} = \tilde{O}(\delta^3)$

initialize $\mu^{(0)}$ and $k_p = k_{\text{thres}}$

**for** $k = 0, 1, ...$ **do**

  **if** $\left\|\nabla_\mu F(\mu^{(k)})\right\|_{L^2(\mu^{(k)})} \leq \varepsilon$ and $k - k_{\text{p}} > k_{\text{thres}}$ **then**

    $\xi \sim \text{GP}(0, K_{\mu^{(k)}})$, $\mu^{(k)} \leftarrow (\text{Id} + \eta_p \xi)\#\mu^{(k)}$

    $k_p \leftarrow k$

  **end if**

  **if** $k = k_{\text{p}} + k_{\text{thres}}$ and $F(\mu^{(k_{\text{P}})}) - F(\mu^{(k)}) \leq F_{\text{thres}}$ **then**

    **return** $\mu^{(k_{\text{P}})}$

  **end if**

  $\mu^{(k+1)} \leftarrow (\text{Id} - \eta\nabla_\mu F(\mu^{(k)}))\#\mu^{(k)}$

**end for**

---

## 5 Convergence Analysis

In this section, we provide the theoretical guarantee for the PWGF algorithm. In addition to the assumptions introduced in the previous sections, we impose the following Lipschitz continuity of the Wasserstein gradient and Hessian.

**Assumption 3.** *F satisfies the following smoothness:*

- $\nabla_\mu F$ *is $L_1$-Lipschitz, i.e. for any $\gamma \in \Gamma(\mu, \nu)$,*

$$\int \|\nabla_\mu F(\mu, x) - \nabla_\mu F(\nu, y)\|^2 \gamma(\mathrm{d}x\mathrm{d}y) \leq L_1^2 \int \|x - y\|^2 \gamma(\mathrm{d}x\mathrm{d}y). \tag{12}$$

- $\nabla_\mu^2 F$ *is $L_2$-Lipschitz, i.e. for any $\gamma \in \Gamma(\mu, \nu)$,*

$$\int \|\nabla_\mu^2 F(\mu, x_1, x_2) - \nabla_\mu^2 F(\nu, y_1, y_2)\|^2 \gamma^{\otimes 2}(\mathrm{d}x\mathrm{d}y) \leq L_2^2 \int \|x - y\|^2 \gamma(\mathrm{d}x\mathrm{d}y). \tag{13}$$

- $\nabla\nabla_\mu F$ *is $L_3$-Lipschitz, i.e. for any $\gamma \in \Gamma(\mu, \nu)$,*

$$\gamma-\text{esssup}_{x\in\mathbb{R}^d} \|\nabla\nabla_\mu F(\mu, x) - \nabla\nabla_\mu F(\nu, y)\|^2 \leq L_3^2 \int \|x - y\|^2 \gamma(\mathrm{d}x\mathrm{d}y). \tag{14}$$

*Remark* 5.1. Similar gradient and Hessian Lipschitz continuity assumptions are made in the convergence analysis of perturbed gradient descent methods in Euclidean spaces (Ge et al., 2015; Jin et al., 2017; Li, 2019). Additionally, the first-order Lipschitz assumption is similar to works on convex functionals (Chizat, 2022; Suzuki et al., 2023).

### 5.1 Convergence Results for Continuous-time PWGF

The following is the main theorem of this paper, asserting that PWGF terminates in polynomial time and reaches an $(\varepsilon, \delta)$-stationary point with high probability.

**Theorem 5.2.** *Let $\varepsilon$, $\delta$, $\zeta > 0$ be chosen such that $(L_2 + L_3)\,\varepsilon \leq \delta^2$ and $\varepsilon, \delta \leq \tilde{O}(1)$.[2] Let the initial point be $\mu_0 \in \mathcal{P}_2(\mathbb{R}^d)$, and $\Delta F = F(\mu_0) - \inf_{\mu\in\mathcal{P}_2(\mathbb{R}^d)} F(\mu)$. For hyperparameters $\eta_p = \tilde{O}\left(\delta^{\frac{3}{2}} \wedge \frac{\delta^3}{\varepsilon}\right)$, $T_{\text{thres}} = \tilde{O}\left(\frac{1}{\delta}\right)$, and $F_{\text{thres}} = \tilde{O}(\delta^3)$, PWGF halts after*

$$t = \tilde{O}\left(\Delta F\left(\frac{1}{\varepsilon^2} + \frac{1}{\delta^4}\right)\right)$$

*time steps and reaches an $(\varepsilon, \delta)$-second-order stationary point with probability $1 - \zeta$.*

To prove Theorem 5.2, we present several supporting results. Detailed proofs are deferred to Appendix E. Lemma 5.3 is a fundamental property of WGF, providing a lower bound on the decrease in the objective for non-stationary points.

---

[2]In the analysis of PGD in Euclidean spaces, $\rho\varepsilon \leq \delta^2$ is often assumed, where $\rho$ is the Lipschitz constant of the Hessian (Jin et al., 2017; Li, 2019). We adopt this assumption to the probabilistic measure space setting, where the Hessian Lipschitz constant becomes $\rho = L_2 + L_3$.

**Lemma 5.3.** *For a curve of probability measures $\mu_t$ following the WGF, the following holds:*

$$F(\mu_0) - F(\mu_t) = \int_0^t \|\nabla_\mu F(\mu_\tau)\|_{\mu_\tau}^2 \, d\tau.$$

Proposition 5.4 is the crucial step of our analysis, showing that perturbing an $(\varepsilon, \delta)$-saddle point using a Gaussian process with kernel (11) enables WGF to move along the unstable direction and decrease the objective function. In other words, it allows us to **efficiently escape the saddle point**.

**Proposition 5.4.** *Set $\eta = O(1)$ and let $\varepsilon$, $\delta$, $\eta_p$, $T_{\text{thres}}$, $F_{\text{thres}}$ be chosen as in Theorem 5.2. Suppose $\mu^\dagger \in \mathcal{P}_2^a(\mathbb{R}^d)$ satisfies $\left\|\nabla_\mu F(\mu^\dagger)\right\|_{L^2(\mu^\dagger)} < \varepsilon$ and $\lambda_0 := \lambda_{\min} H_{\mu^\dagger} \leq -\delta$. Generating $\xi \sim \mathrm{GP}(0, k_\mu)$ and setting $\mu_0 = (\mathrm{Id} + \eta_p \xi) \sharp \mu^\dagger$ as the initial point of the WGF, we have with probability $1 - \zeta'$:*

$$F(\mu^\dagger) - F(\mu_{T_{\text{thres}}}) \geq F_{\text{thres}}.$$

Combining these results establishes convergence.

*Proof of Theorem 5.2.* Let $\varepsilon$, $\delta$, $\zeta > 0$ be chosen arbitrarily chosen such that $(L_2 + L_3)\,\varepsilon \leq \delta^2$, and set $\zeta' > 0$ such that $\zeta'$ is polynomial in $\frac{1}{\delta}$ and $\zeta$ up to logarithmic factors.

From Proposition 5.4, perturbations occur at most $m := \lceil \frac{\Delta F}{F_{\text{thres}}} \rceil$ times. Thus, the probability of failure after $m$ perturbations is at most $1 - (1 - \zeta')^m \leq m\zeta'$. Setting $\zeta' = \frac{\zeta}{m}$ ensures that the algorithm reaches an $(\varepsilon, \delta)$-second order stationary point with probability at least $1 - \zeta$.

PWGF consists of a gradient descent phase and an evaluation phase, where the decrease in the objective is assessed after applying a perturbation. Then we define the gradient descent phase as *State 0*, and the evaluation phase as *State 1*. Let $T_0$ denote the total time in *State 0*, where $\|\nabla_\mu F(\mu)\|_{L^2(\mu)} \geq \varepsilon$. By Lemma 5.3, the decrease in the objective during *State 0* is at least $\varepsilon^2 T_0$, implying $T_0 \leq \frac{\Delta F}{\varepsilon^2}$. Moreover, the total time $T_1$ in *State 1* is bounded as $T_1 \leq m T_{\text{thres}} = \frac{\Delta F T_{\text{thres}}}{F_{\text{thres}}} = \tilde{O}\left(\frac{\Delta F}{\delta^4}\right)$. Hence, PWGF halts in $T_0 + T_1 = \tilde{O}\left(\Delta F\left(\frac{1}{\varepsilon^2} + \frac{1}{\delta^4}\right)\right)$. $\square$

## 5.2 Convergence Analysis for Discrete-time PWGF

We also prove convergence to a second-order stationary point for the time-discretized PWGF (Algorithm 2).

**Theorem 5.5.** *Let the initial point be $\mu_0 \in \mathcal{P}_2(\mathbb{R}^d)$ and denote $\Delta F = F(\mu_0) - \inf_{\mu \in \mathcal{P}_2(\mathbb{R}^d)} F(\mu)$. Set $\eta = O(1)$ and let $\varepsilon$, $\delta$, $\eta_p$, $T_{\text{thres}}$, $F_{\text{thres}}$ be chosen as in Theorem 5.2. Then, discrete time PWGF halts after*

$$k = \tilde{O}\left(\Delta F\left(\frac{1}{\varepsilon^2} + \frac{1}{\delta^4}\right)\right)$$

*steps and reaches an $(\varepsilon, \delta)$-second-order stationary point with probability $1 - \zeta$.*

The proof is similar to the continuous-time case; details are deferred to Appendix E.2. Since second-order stationary points are global optima for a strictly benign objective, the convergence of PWGF to a global solution is also guaranteed.

**Corollary 5.6.** *Under the same setting as Theorem 5.5, discrete-time PWGF for $(\varepsilon, \delta, \alpha)$-strictly benign objective $F$ halts after $\tilde{O}\left(\Delta F\left(\frac{1}{\varepsilon^2} + \frac{1}{\delta^4}\right)\right)$ steps and reaches $\alpha$-close to some global optima $\mu^o$ ; $W_2(\mu, \mu^o) \leq \alpha$ with probability $1 - \zeta$.*

## 6 Conclusion

We proposed a new method for non-convex probability optimization, perturbed Wasserstein gradient flows (PWGF), which alternates between perturbing near saddle points using a Hessian-guided Gaussian process and evolving via WGF. We have established that PWGF efficiently achieves second-order optimality with high probability. A potential avenue for future work is to reduce computational cost by using a stochastic approximation of the Hessian as the kernel of the Gaussian process, analogous to stochastic gradients. Another direction is to provide a method for analyzing benignity of non-convex distributional objectives, thereby broadening the range of applications of PWGF.

## Acknowledgments

NY and JK were partially supported by JST CREST (JPMJCR2015). TS was partially supported by JSPS KAKENHI (24K02905) and JST CREST (JPMJCR2115). This research is supported by the National Research Foundation, Singapore, Infocomm Media Development Authority under its Trust Tech Funding Initiative, and the Ministry of Digital Development and Information under the AI Visiting Professorship Programme (award number AIVP-2024-004). Any opinions, findings and conclusions or recommendations expressed in this material are those of the author(s) and do not reflect the views of National Research Foundation, Singapore, Infocomm Media Development Authority, and the Ministry of Digital Development and Information.

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

# A    Applications of Wasserstein Gradient Flow

## A.1    Mean-Field Analysis

Mean-field analysis can be applied to optimization problems where the objective function is expressed as a function of the ensemble average of some underlying functions. We consider the optimization problem :

$$\text{minimize} \quad G\left(\frac{1}{N}\sum_{j=1}^{N} h(x_j)\right) \quad \text{s.t.} \quad x_1, \cdots, x_N \in \mathbb{R}^d, \tag{15}$$

where $h : \mathbb{R}^d \to \mathbb{R}^{d'}$, $G : \mathbb{R}^{d'} \to \mathbb{R}$, and $d' = 1$ for simplicity. The gradient direction of the variable $x_j$ is computed as follows :

$$\nabla_{x_j} G\left(\frac{1}{N}\sum_{j=1}^{N} h(x_j)\right) = \frac{1}{N} G'\left(\frac{1}{N}\sum_{j=1}^{N} h(x_j)\right) \nabla h(x_j). \tag{16}$$

Taking the mean field limit $N \to \infty$ of (15), we lift this problem to the space of measures:

$$\text{minimize} \quad G\left(\int h(x)\mu(\mathrm{d}x)\right) \quad \text{s.t.} \quad \mu \in \mathcal{P}_2(\mathbb{R}^d). \tag{17}$$

The Wasserstein gradient is computed as follows :

$$\nabla_\mu G(\mu) = \nabla \frac{\delta F}{\delta \mu}(\mu, x) = G'\left(\int h(x)\mu(\mathrm{d}x)\right) \nabla h(x). \tag{18}$$

Consequently, from (16) and (18), the update rules for gradient descent applied to the original problem (15) and WGF applied to the lifted problem (17) are equivalent, up to a constant scaling factor. However, properties differ significantly between the two formulations. For example, when $F$ is the identity function, the original problem (15) is not necessarily linear, while the mean-field problem (17) is linear with respect to $\mu$. Similarly, for commonly encountered loss functions with convex losses, (15) is not necessarily convex with respect to the variables $x_1, \ldots, x_N$, but in the mean-field setting (17) becomes convex with respect to $\mu$.

The properties of functions defined on the space of probability measures facilitate the design of WGF-based algorithms with improved performance. The works of Nitanda et al. (2022) and Chizat (2022) have explored the ability of optimization of mean-field Langevin dynamics (MFLD), an approach that augments WGF with Brownian noise, based on the convexity of the objective function. Similarly, Kim & Suzuki (2024) have focused on the benignity of objective functions regarding in-context learning of certain transformer models and, leveraging this insight, have proposed a birth-death modification and also an isotropic perturbation scheme. Our proposed algorithm contributes to this line of research by providing the first convergence guarantees for strictly benign problems on the space of probability measures.

## A.2    Additional Related Works

WGF has important applications in Bayesian inference, where posterior distributions are usually estimated from data via variational inference (VI). VI formulates posterior estimation as an optimization problem of the KL divergence. In particular, particle-based VI is founded on the idea of evolving empirical measures formed by particles using WGF. This idea originates from the work of Jordan et al. (1998), which established a connection between diffusion processes and gradient descent in 2-Wasserstein space with entropy regularization.

Stein variational gradient descent (Liu & Wang, 2016; He et al., 2024) circumvents computational difficulties in calculating descent directions through kernel methods. This approach can be seen as a version of WGF where an integral operator acts on the descent direction (Chewi et al., 2020; Duncan et al., 2023). In this context, derivative methods such as Newton's method on the space of probability measures (Detommaso et al., 2018; Wang & Li, 2020) and accelerated methods (Liu et al., 2018; Taghvaei & Mehta, 2019; Wang & Li, 2022) have also been proposed. Furthermore, optimization of measures also appears in contexts such as online optimization (Guo et al., 2022; Han, 2024) and reinforcement learning (Richemond & Maginnis, 2017; Zhang et al., 2018).

# B  Wasserstein Geometry

## B.1  Wasserstein Space

In this section, we provide basic aspects of Wasserstein geometry and propositions used in this paper. We refer to Ambrosio et al. (2008) for a comprehensive review.

We assume that all curves $\mu_t \in \mathcal{P}_2(\mathbb{R}^d)$ that appear in this section are absolutely continuous and satisfy the continuity equation with respect to a vector field $v_t \in \mathrm{Tan}_{\mu_t}\mathcal{P}_2(\mathbb{R}^d)$. The existence condition for solutions to the continuity equation corresponding to a vector field $v_t$ is provided, for example, by Ambrosio et al. (2008), Section 8.2.

**Definition B.1** (Wasserstein distance)**.** The 2-Wasserstein distance between two points $\mu, \nu \in \mathcal{P}_2(\mathbb{R}^d)$ is defined as

$$W_2(\mu, \nu)^2 := \inf_{\gamma \in \Gamma(\mu, \nu)} \int \|x - y\|^2 \gamma(\mathrm{d}x\mathrm{d}y). \tag{19}$$

Here, $\Gamma(\mu, \nu)$ represents the set of all transport plans:

$$\Gamma(\mu, \nu) = \left\{ \gamma \in \mathcal{P}_2(\mathbb{R}^d \times \mathbb{R}^d) \,\middle|\, p_1 \# \gamma = \mu, \ p_2 \# \gamma = \nu \right\}.$$

Moreover, $\gamma \in \Gamma(\mu, \nu)$ is called the optimal transport plan, if the infmum in (19) is attained by $\gamma$. The set of all optimal transport plans is denoted by $\Gamma_o(\mu, \nu)$. Furthermore, if a measurable map $f : \mathbb{R}^d \to \mathbb{R}^d$ satisfies $(\mathrm{Id} \times f)\sharp\mu \in \Gamma_o(\mu, \nu)$, then $f$ is called the optimal transport map between $\mu$ and $\nu$. In this case, we denote the optimal transport map $f$ as $\mathcal{T}_\mu^\nu$.

$W_2 : \mathcal{P}_2(\mathbb{R}^d) \times \mathcal{P}_2(\mathbb{R}^d) \to \mathbb{R}$ defines a metric structure on $\mathcal{P}_2(\mathbb{R}^d)$. Therefore, we consider $\mathcal{P}_2(\mathbb{R}^d)$ as a metric space equipped with the $W_2$ metric. Convergence in the $W_2$ metric is equivalent to the weak convergence of measures plus uniform integrability of second moments (Ambrosio et al., 2008).

An optimal transport plan is guaranteed to exist for any $\mu, \nu \in \mathcal{P}_2(\mathbb{R}^d)$, as stated in the following proposition.

**Proposition B.2.** *For $\mu, \nu \in \mathcal{P}_2(\mathbb{R}^d)$, there exists an optimal transport plan between $\mu$ and $\nu$. That is, there exists $\gamma \in \Gamma(\mu, \nu)$ such that*

$$W_2(\mu, \nu)^2 = \int \|x - y\|^2 \gamma(\mathrm{d}x\mathrm{d}y).$$

*Proof.* Refer to Ambrosio et al. (2008), Chapter 6. $\qquad\square$

An optimal transport map, not a plan, does not necessarily exist. Specifically, there are cases where even transport maps themselves do not exist. For example, when $d = 1$, $\mu = \delta_0$, and $\nu = \frac{\delta_{-1} + \delta_1}{2}$, no transport map exists, as $T\#\mu \neq \nu$ for any $T : \mathbb{R} \to \mathbb{R}$. However, The following proposition establishes that under certain conditions, the existence and uniqueness of an optimal transport map are guaranteed.

**Proposition B.3** (Brenier's theorem)**.** *For any $\mu \in \mathcal{P}_2^a(\mathbb{R}^d)$ and $\nu \in \mathcal{P}_2(\mathbb{R}^d)$, there exists an optimal transport map $\mathcal{T}_\mu^\nu : \mathbb{R}^d \to \mathbb{R}^d$. Specifically, $\mathcal{T}_\mu^\nu$ satisfies*

$$W_2(\mu, \nu)^2 = \int \|\mathcal{T}_\mu^\nu(x) - x\|^2 \mu(\mathrm{d}x).$$

*Furthermore, the following hold:*

- $\Gamma_o(\mu, \nu) = \{(\mathrm{Id} \times \mathcal{T}_\mu^\nu)\sharp\mu\}$*, that is, the unique optimal transport plan from $\mu$ to $\nu$ is induced by $\mathcal{T}_\mu^\nu$.*

- *The optimal transport map can be expressed as $\mathcal{T}_\mu^\nu = \nabla\varphi$, where $\varphi$ is a convex function defined $\mu$-almost everywhere.*

- *If $\nu \in \mathcal{P}_2^a(\mathbb{R}^d)$ as well, then $\mathcal{T}_\nu^\mu \circ \mathcal{T}_\mu^\nu = \mathrm{Id}$ $\mu$-a.e. and $\mathcal{T}_\mu^\nu \circ \mathcal{T}_\nu^\mu = \mathrm{Id}$ $\nu$-a.e.*

*Proof.* See Ambrosio et al. (2008), Chapter 6. $\qquad\square$

$\mathcal{P}_2(\mathbb{R}^d)$ forms a geodesic metric space due to the existence of optimal transport plans and the pushforward property:

**Proposition B.4.** *Let $\mu, \nu \in \mathcal{P}_2(\mathbb{R}^d)$, and $\gamma \in \Gamma_0(\mu, \nu)$. Then $\mu_t = ((1-t)p_1 + tp_2)\sharp\gamma$ defines a geodesic between $\mu$ and $\nu$, i.e.,*

$$W_2(\mu, \mu_t) = tW_2(\mu, \nu) \quad (\forall t \in [0,1]).$$

*In particular, in case where $\gamma$ is induced by an optimal transport map $\mathcal{T}_\mu^\nu$, $\mu_t = ((1-t)\mathrm{Id} + t\mathcal{T}_\mu^\nu)\#\mu$ and $\mathcal{T}_\mu^{\mu_t} = (1-t)\mathrm{Id} + t\mathcal{T}_\mu^\nu$ hold.*

*Proof.* See Ambrosio et al. (2008), Lemma 7.2.1. □

The following continuity equation describes how a particle distribution $\mu_t$ evolves along a time-dependent vector field $v_t$.

**Definition B.5** (Continuity equation, Ambrosio et al. (2008))**.** Let $\mu_t \in \mathcal{P}_2(\mathbb{R}^d)$ $(t \in I)$ be a curve in Wasserstein space, and let $v_t \in L(\mu_t)^d$ $(t \in I)$ be the corresponding vector field. The curve $\mu_t$ is said to satisfy the continuity equation with respect to the vector field $v_t$ if the distribution equation:

$$\partial_t \mu_t + \nabla \cdot (v_t \mu_t) = 0 \tag{20}$$

holds. Equation (20) means that for all $\varphi \in C_0^\infty(\mathbb{R}^d)$,

$$\frac{\mathrm{d}}{\mathrm{d}t} \int \varphi(x)\mu_t(\mathrm{d}x) = \int \nabla\varphi(x)^\top v_t(x)\mu_t(\mathrm{d}x).$$

The absolute continuity of the curve $\mu_t$ with respect to the Wasserstein distance is equivalent to the satisfaction of the continuity equation for some $v_t \in L^2(\mu_t)$ (Ambrosio et al. (2008) Theorem 8.3.1). In this sense, the continuity equation provides a concept of differentiation consistent with the distance structure induced by the Wasserstein distance.

**Proposition B.6** (Ambrosio et al. (2008), Theorem 8.3.1)**.** *Let $I \subset \mathbb{R}$ be an open interval, and let $\mu_t : I \to \mathcal{P}_2(\mathbb{R}^d)$ be a continuous curve. $\mu_t$ is absolutely continuous, if and if only $\mu_t$ satisfies continuity equation (3) for some vector field $v_t \in L^2(\mu_t)$.*

For the vector field that gives the continuity equation (3),

$$\nabla\cdot((v_t - w_t)\mu_t) = 0 \iff v_t - w_t \in \left\{\nabla\varphi \mid \varphi \in C_0^\infty(\mathbb{R}^d)\right\}^{\perp L^2(\mu_t)} =: X_{\mu_t}$$

is equivalent to the fact that the continuity equation gives the same curve $\mu_t$. Noting the orthogonal decomposition $L^2(\mu_t)^d = X_{\mu_t} \oplus X_{\mu_t}^\perp$, the vector field with the minimal $L^2$ norm among those that give the same curve $\mu_t$ must have no component in the subspace $X_{\mu_t}$, and it follows that $v_t \in X_{\mu_t}^\perp$. From these observations, we define the tangent space representing infinitesimal changes in the space of probability measures $\mathcal{P}_2(\mathbb{R}^d)$ as follows :

**Definition B.7** (Tangent bundle, Ambrosio et al. (2008))**.** We define the tangent space $\mathrm{Tan}_\mu \mathcal{P}_2(\mathbb{R}^d) \subset L^2(\mu)$ at $\mu \in \mathcal{P}_2(\mathbb{R}^d)$ as

$$\mathrm{Tan}_\mu \mathcal{P}_2(\mathbb{R}^d) := X_\mu^\perp = \overline{\{\nabla\varphi \mid \varphi \in C_0^\infty(\mathbb{R}^d)\}}^{L^2(\mu)}.$$

The following proposition, referred to as the Benamou-Brenier formula, demonstrates that the Wasserstein distance is characterized by the minimal action among absolutely continuous curves connecting two given probability measures.

**Proposition B.8** (Benamou-Brenier formula)**.** *For any $\mu, \nu \in \mathcal{P}_2(\mathbb{R}^d)$ and $T > 0$, the following holds:*

$$W_2(\mu, \nu)^2 = \inf \left\{ T \int_0^T \|v_t\|_{L^2(\mu_t)}^2 \, \mathrm{d}t \,\middle|\, \partial_t \mu_t + \nabla \cdot (v_t \mu_t) = 0 \ (t \in (0,T)), \ \mu_0 = \mu, \mu_T = \nu \right\}.$$

*Proof.* From Ambrosio et al. (2008), Theorem 8.3.1,

$$W_2(\mu, \nu)^2 = \inf \left\{ \int_0^1 \|v_t\|_{L^2(\mu_t)}^2 \, \mathrm{d}t \,\middle|\, \partial_t \mu_t + \nabla \cdot (v_t \mu_t) = 0 \ (t \in (0,1)), \ \mu_0 = \mu, \mu_1 = \nu \right\}.$$

By changing the variable $t \mapsto \frac{t}{T}$, the claim follows. □

The following proposition establishes that the infinitesimal behavior of an absolutely continuous curve can be expressed by the pushforward $(\mathrm{Id} + hv_t)\#\mu_t$.

**Proposition B.9.** *Let $\mu_t$ be an absolutely continuous curve satisfying continuous equation with vector field $v_t \in \mathrm{Tan}_{\mu_t}\mathcal{P}_2(\mathbb{R}^d)$. Then,*

$$\lim_{h\to 0} \frac{W_2(\mu_{t+h}, (\mathrm{Id} + hv_t)\#\mu_t)}{h} = 0.$$

*Proof.* See Ambrosio et al. (2008), Proposition 8.4.6. $\square$

The following propositions provide sufficient conditions for a map to be an optimal transport map.

**Proposition B.10** (Santambrogio (2015), Theorem 1.48)**.** *Suppose that $\mu \in \mathcal{P}_2(\mathbb{R}^d)$ and that $\varphi : \mathbb{R}^d \to \mathbb{R}$ is a convex and $\mu$-a.e. differentiable function with $\nabla\varphi \in L^2(\mu)^d$. Then, the map $\nabla\varphi$ provides the optimal transport map from $\mu$ to $(\nabla\varphi)\sharp\mu$.*

*Proof.* Let $\varphi^*$ be a Legendre-Fenchel transformation of $\varphi$. Then, it holds that

$$\varphi(x) + \varphi^*(y) \geq \langle x, y \rangle \quad \forall x, y \in \mathbb{R},$$
$$\varphi(x) + \varphi^*(y) = \langle x, y \rangle \quad (\mathrm{Id} \times \nabla\varphi)\#\mu\text{-a.e. } (x, y).$$

For any transport plan $\gamma \in \Gamma(\mu, \nabla\varphi\#\mu)$, it holds that

$$-2\int \langle x, y \rangle \, \gamma(\mathrm{d}x\mathrm{d}y) \geq -2\int (\varphi(x) + \varphi^*(y))\gamma(\mathrm{d}x\mathrm{d}y)$$

$$= -2\int \varphi(x)\mu(\mathrm{d}x) - \int \varphi^*(\nabla\varphi(x))\mu(\mathrm{d}x)$$

$$= -2\int \langle x, y \rangle \, (\mathrm{Id} \times \nabla\varphi)\#\mu.$$

By adding $\int(\|x\|^2 + \|y\|^2)\gamma(\mathrm{d}x\mathrm{d}y) = \int(\|x\|^2 + \|y\|^2)(\mathrm{Id}\times\nabla\varphi)\#\mu(\mathrm{d}x)$ both sides of the inequality, we obtain

$$\int \|x - y\|^2\gamma(\mathrm{d}x\mathrm{d}y) \geq \int \|x - y\|^2(\mathrm{Id} \times \nabla\varphi)\#\mu(\mathrm{d}x).$$

Then $\nabla\varphi$ is a optimal transport map. $\square$

**Proposition B.11** (Lanzetti et al. (2025), Lemma 2.4)**.** *Let $\mu \in \mathcal{P}_2(\mathbb{R}^d)$ and $\varphi \in C_c^\infty(\mathbb{R}^d)$. Then, there exists $\bar{h} > 0$ such that $\mathrm{Id} + h\nabla\psi$ is an optimal transport map from $\mu$ to $(\mathrm{Id} + h\nabla\psi)\#\mu$ for $h \in [-\bar{h}, \bar{h}]$. Furthermore,*

$$W_2(\mu, (\mathrm{Id} + h\nabla\psi)\#\mu) = h\|\nabla\psi\|_{L^2(\mu)}.$$

*Proof.* From $\psi \in C_c^\infty(\mathbb{R}^d)$, we have $\sup_{x\in\mathbb{R}^d}\|\nabla\psi\|(x) < \infty$, $\sup_{x\in\mathbb{R}^d}\|\nabla^2\psi(x)\| < \infty$. Then, by taking $\bar{h} = \sup_{x\in\mathbb{R}^d}\|\nabla^2\psi(x)\|$, we verify that $\frac{1}{2}\|x\|^2 + h\psi(x)$ is convex and $\nabla\left(\frac{1}{2}\|x\|^2 + h\psi(x)\right) = (\mathrm{Id} + h\nabla\psi)(x)$ is $L^2(\mu)$ integrable for all $h \in [-\bar{h}.\bar{h}]$. Noting that $\nabla\left(\frac{1}{2}\|x\|^2 + h\psi(x)\right) = (\mathrm{Id} + h\nabla\psi)(x)$ and applying Proposition B.10, we obtain that $\mathrm{Id} + h\nabla\psi$ yields an optimal transport map. Consequently, the Wasserstein distance between $\mu$ and $(\mathrm{Id} + h\nabla\psi)\#\mu$ is computed as:

$$W_2(\mu, (\mathrm{Id} + h\nabla\psi)\#\mu) = \left(\int \|x + h\nabla\psi(x) - x\|^2\mu(\mathrm{d}x)\right)^{\frac{1}{2}} = h\|\nabla\psi\|_{L^2(\mu)}.$$

$\square$

**Corollary B.12.** *Let $\mu \in \mathcal{P}_2(\mathbb{R}^d)$ and $\varphi \in C^2(\mathbb{R}^d)$ satisfy $\|\nabla\varphi\|_{L^2(\mu)} < \infty$, $\mu-\mathrm{esssup}_{x\in\mathbb{R}^d}\|\nabla^2\varphi(x)\| < \infty$. Then, there exists $\bar{h} > 0$ such that $\mathrm{Id} + h\nabla\varphi$ is an optimal transport map from $\mu$ to $(\mathrm{Id} + h\nabla\varphi)\#\mu$ for $h \in [-\bar{h}, \bar{h}]$. Furthermore, $W_2(\mu, (\mathrm{Id} + h\nabla\varphi)\#\mu) = h\|\nabla\varphi\|_{L^2(\mu)}$ holds.*

## B.2 Wasserstein Gradient

This section provides additional discussion on the relationship between the Wasserstein gradient and the first variation.

**Definition B.13** (Wasserstein gradient, Lanzetti et al. (2025); Bonnet (2019))**.** The Wasserstein gradient $\nabla_\mu F : \mathcal{P}_2(\mathbb{R}^d) \times \mathbb{R}^d \to \mathbb{R}^d$ at $\mu \in \mathcal{P}_2(\mathbb{R}^d)$ is defined as a vector-field such that for any $\nu \in \mathcal{P}_2(\mathbb{R}^d)$ and $\gamma \in \Gamma_0(\mu, \nu)$, it holds that

$$F(\nu) - F(\mu) = \int \nabla_\mu F(\mu, x)^\top (y - x) \gamma(\mathrm{d}x\mathrm{d}y) + O\big(W_2(\mu, \nu)^2\big). \tag{21}$$

In particular, there exists a unique element that satisfies $\nabla_\mu F(\mu) \in \mathrm{Tan}_\mu \mathcal{P}_2(\mathbb{R}^d)$ (Lanzetti et al. (2025) Proposition 2.5), and we take this as the Wasserstein gradient.

The first variation $\frac{\delta F}{\delta \mu}$ appearing in the WGF equation (1) is defined as follows.

**Definition B.14** (First variation)**.** The first variation $\frac{\delta F}{\delta \mu} : \mathcal{P}_2(\mathbb{R}^d) \times \mathbb{R}^d \to \mathbb{R}$ is defined as a functional satisfying for any $\nu \in \mathcal{P}_2(\mathbb{R}^d)$,

$$\frac{\mathrm{d}}{\mathrm{d}h}\bigg|_{h=0} F(\mu + h(\nu - \mu)) = \int \frac{\delta F}{\delta \mu}(\mu, x)(\mu - \nu)(\mathrm{d}x). \tag{22}$$

The first variation, if it exists, is unique up to a constant difference.

Recall the definition of the total variation:

**Definition B.15** (Total variation)**.** Suppose that $\mu, \nu \in \mathcal{P}_2(\mathbb{R}^d)$. The total variation between $\mu$ and $\nu$ is:

$$\mathrm{TV}(\mu, \nu) := \sup_{B \in \mathcal{B}(\mathbb{R}^d)} |\mu(B) - \nu(B)| = \frac{1}{2} \sup_{\mathcal{C} \subset \mathcal{B}(\mathbb{R}^d), \cup\mathcal{C}=\mathbb{R}^d} \sum_{B \in \mathcal{C}} |\mu(B) - \nu(B)|.$$

As is evident from the definition, the mixture $(1 - h)\mu + h\nu$, defines the constant-speed geodesic between $\mu, \nu$ in the sense of total variation. Thus, the first variation can be interpreted as the coefficient of differentiation along the geodesic with respect to the total variation distance. On the other hand, as observed in the previous section, the geodesic of $W_2$ is represented as $((1 - h)p_1 + hp_2)\#\gamma$ for $\gamma \in \Gamma_0(\mu, \nu)$. Therefore, the Wasserstein gradient can be understood as the coefficient of differentiation along the geodesic with respect to the Wasserstein distance.[3]

Through a naive but mathematically non-rigorous calculation, we have

$$
\begin{aligned}
F(\nu) - F(\mu) &\approx \int \frac{\delta F}{\delta \mu}(\mu, x)(\nu - \mu)(\mathrm{d}x) \\
&= \int \left( \frac{\delta F}{\delta \mu}(\mu, y) - \frac{\delta F}{\delta \mu}(\mu, x) \right) \gamma(\mathrm{d}x\mathrm{d}y) \\
&\approx \int \left( \left\langle \nabla \frac{\delta F}{\delta \mu}(\mu, x), y - x \right\rangle + \frac{1}{2} \left\langle y - x, \nabla^2 \frac{\delta F}{\delta \mu}(\mu, x)(y - x) \right\rangle \right) \gamma(\mathrm{d}x\mathrm{d}y) \\
&= \int \left\langle \nabla \frac{\delta F}{\delta \mu}(\mu, x), y - x \right\rangle \gamma(\mathrm{d}x\mathrm{d}y) + O\big(W_2(\mu, \nu)^2\big).
\end{aligned}
$$

This points to $\nabla_\mu F = \nabla \frac{\delta F}{\delta \mu}(\mu)$. This equation does not hold without certain conditions, but it is valid and implicitly assumed to hold in many practical cases. Kent et al. (2021) make a brief mention of this frustration. The equivalence $\nabla_\mu \cdot = \nabla \frac{\delta \cdot}{\delta \mu}$ specifically provides the following correspondence.

$$\nabla_\mu F(\mu, x) = \nabla \frac{\delta F}{\delta \mu}(\mu, x),$$

$$\nabla_\mu^2 F(\mu, x, y) = \nabla_x \nabla_y \frac{\delta^2 F}{\delta \mu^2}(\mu, x, y),$$

$$\nabla \nabla_\mu F(\mu, x) = \nabla^2 \frac{\delta F}{\delta \mu}(\mu, x).$$

---

[3]There is no strict dominance or subordination between total variation and Wasserstein distance in $\mathbb{R}^d$.

# C   Optimality Conditions for Functionals on Probability Space

In this section, we discuss the details of the optimality conditions for functionals on probability measures based on the Wasserstein gradient. The first-order optimality conditions have been studied in Bonnet (2019) and Lanzetti et al. (2025), and we begin by reviewing these works. We then extend these results to second-order conditions.

## C.1   First-order Condition

The first order condition for probability measure optimization is studied by Lanzetti et al. (2025), including constrained optimization problem. Here, we review the results of the first-order condition for unconstrained problem. In the next section, we extend these results and obtain the second-order condition.

The following proposition shows that Equation (21) holds even when $\gamma$ is not optimal.

**Proposition C.1.** *Let $F : \mathcal{P}_2(\mathbb{R}^d) \to \mathbb{R}$ be a functional on probability space, and differentiable at $\mu$. Then, for any transport plan $\gamma \in \Gamma(\mu, \nu)$ (not necessarily an optimal one),*

$$F(\nu) - F(\mu) = \int \nabla_\mu F(\mu, x)^\top (y - x) \gamma(\mathrm{d}x\mathrm{d}y) + O\left( \int \|x - y\|^2 \gamma(\mathrm{d}x\mathrm{d}y) \right).$$

*Proof.* See Lanzetti et al. (2025). $\qquad\qquad\square$

The following two propositions are useful for computing changes in the objective function. The next proposition provides a first-order Taylor expansion for infinitesimal perturbations induced by a vector field.

**Proposition C.2.** *For a sufficiently smooth functional $F : \mathcal{P}_2(\mathbb{R}^d) \to \mathbb{R}$ and a vector field $v \in L^2(\mu)^d$ ($\mu \in \mathcal{P}_2(\mathbb{R}^d)$), it holds that*

$$F((\mathrm{Id} + hv)\#\mu) - F(\mu) = h \langle \nabla_\mu F(\mu), v \rangle_{L^2(\mu)} + O(h^2).$$

*Proof.* Consider the situation of Proposition C.1 where $\nu = (\mathrm{Id} + hv)\#\mu$, $\gamma = (\mathrm{Id} \times (\mathrm{Id} + hv))\#\mu$. Then,

$$\int \nabla_\mu \langle F(\mu, x), y - x \rangle \gamma(\mathrm{d}x\mathrm{d}y) = \int \langle \nabla_\mu F(\mu, x), (x + hv(x) - x \rangle \mu(\mathrm{d}x)$$
$$= h \langle \nabla_\mu F(\mu), v \rangle_{L^2(\mu)},$$

and also

$$\int \|x - y\|^2 \gamma(\mathrm{d}x\mathrm{d}y) = \int \|(x + hv(x)) - x\|^2 \mu(\mathrm{d}x) = h^2 \|v\|^2_{L^2(\mu)}$$

holds. So, by Proposition C.1, the claim follows. $\qquad\qquad\square$

The following proposition provides a formula for differentiating the objective along an absolutely continuous curve. This corresponds to the chain rule for differentiable curves in Euclidean space.

**Proposition C.3** (Chain rule). *For a sufficiently smooth functional $F : \mathcal{P}_2(\mathbb{R}^d) \to \mathbb{R}$ and absolutely continuous curve $\mu_t$ satisfying a continuous equation $\partial_t \mu_t + \nabla \cdot (v_t \mu_t) = 0$, $t \mapsto F(\mu_t)$ is differentiable, and the following holds:*

$$\frac{\mathrm{d}}{\mathrm{d}t} F(\mu_t) = \langle \nabla_\mu F(\mu_t), v_t \rangle_{L^2(\mu_t)}.$$

*Proof.* By applying the definition of Wasserstein gradient (Definition B.13) for $\mu = \mu_{t+h}$, $\nu = (\mathrm{Id} + hv_t)\#\mu_t$ and $\gamma \in \Gamma_0(\mu_{t+h}, (\mathrm{Id} + hv_t)\#\mu_t)$, it holds that

$$\left| \frac{F(\mu_{t+h}) - F((\mathrm{Id} + hv_t)\#\mu_t)}{h} \right| \leq \left| \frac{1}{h} \int \langle \nabla_\mu F(\mu_t, x), y - x \rangle \gamma(\mathrm{d}x\mathrm{d}y) \right| + o(1)$$
$$\leq \|\nabla_\mu F(\mu_t)\|_{L^2(\mu_t)} \frac{W_2(\mu_{t+h}, (\mathrm{Id} + hv_t)\#\mu_t)}{h} + o(1)$$
$$\to 0,$$

as $h \to 0$, where Cauchy Schwarz inequality is used in second line and Proposition B.9 is used in third line. Then by Proposition C.2, we have

$$\frac{\mathrm{d}}{\mathrm{d}t}F(\mu_t) = \lim_{h \to 0} \frac{F(\mu_{t+h}) - F(\mu_t)}{h}$$
$$= \lim_{h \to 0} \frac{F((\mathrm{Id} + hv_t)\#\mu_t) - F(\mu_t)}{h}$$
$$= \langle \nabla_\mu F(\mu_t), v_t \rangle_{L^2(\mu_t)}.$$

$\square$

The above proposition shows that the Wasserstein gradient plays a crucial role for first-order perturbations. In particular, if $\nabla_\mu F(\mu) = 0$, the objective function does not change under any first-order perturbation. This suggests that it is reasonable to define first-order stationary points as points satisfying $\nabla_\mu F(\mu) = 0$. Furthermore, supporting this observation, Lanzetti et al. (2025) established the following proposition.

**Proposition C.4** (First-order necessary condition). *Let $\mu^* \in \mathcal{P}_2(\mathbb{R}^d)$ be a local minimizer of a differentiable functional $F : \mathcal{P}_2(\mathbb{R}^d) \to \mathbb{R}$ i.e. it holds that there exists a constant $r > 0$ such that*

$$W_2(\mu, \mu^*) < r \implies F(\mu) \leq F(\mu^*).$$

*Then the Wasserstein gradient of $F$ vanishes at $\mu^*$:*

$$\nabla_\mu F(\mu^*, x) = 0 \quad \mu^*\text{-a.e.}x,$$

*i.e. $\nabla_\mu F(\mu^*) = 0$ in $L^2(\mu^*)^d$.*

*Proof.* See Lanzetti et al. (2025), Theorem 3.1. $\square$

**Proposition C.5** (First-order sufficient condition). *Suppose that $F$ is differentiable and $\alpha$-geodesically convex with $\alpha \geq 0$, i.e. it holds that*

$$F(\nu) - F(\mu) \geq \int \langle \nabla_\mu F(\mu, x), y - x \rangle \gamma(\mathrm{d}x\mathrm{d}y) + \frac{\alpha}{2} W_2(\mu, \nu)^2 \quad \forall \gamma \in \Gamma_0(\mu, \nu).$$

*Then, $\nabla_\mu F(\mu^*) = 0$ $\mu$-a.e. implies that $\mu^*$ is global minimizer of F, i.e. $F(\mu) \geq F(\mu^*)$ holds for any $\mu \in \mathcal{P}_2(\mathbb{R}^d)$.*

*Proof.* See Lanzetti et al. (2025), Theorem 3.3. $\square$

### C.2 Second-order Condition

Building on the results from the previous section, we discuss second-order optimality in measure optimization. To obtain second-order terms, we first prove the following lemma. The key point is that we obtain the derivative $F(\mu_t)$ not only at $t = 0$, but also for $0 < t < 1$. By obtaining this, we can compute the second-order coefficient.[4]

**Lemma C.6.** *Let $F$ be a sufficiently smooth functional, $\mu, \nu \in \mathcal{P}_2(\mathbb{R}^d)$, and $\mu_h$ be a constant geodesic induced by $\gamma \in \Gamma(\mu, \nu)$. Then, for any $0 \leq t < 1$,*

$$\frac{\mathrm{d}}{\mathrm{d}t}F(\mu_t) = \int \langle \nabla_\mu F(\mu_t, (1-t)x + ty), y - x \rangle \gamma(\mathrm{d}x\mathrm{d}y).$$

*Proof.* By Proposition C.1, we obtain the following statement (*): let $\mu, \nu \in \mathcal{P}_2(\mathbb{R}^d)$, $\gamma \in \Gamma(\mu, \nu)$ and let $\mu_t := ((1-t)p_1 + tp_2)\#\gamma$, then

$$\frac{\mathrm{d}}{\mathrm{d}t}\Big|_{t=0} F(\mu_t) = \int \langle \nabla_\mu F(\mu, x), y - x \rangle \gamma(\mathrm{d}x\mathrm{d}y).$$

---

[4]Similar to this proposition, higher-order terms (third-order and beyond) can be computed in the same manner. It is conjectured that higher-order terms can also be expressed as the action of $\nabla_\mu$ or $\nabla$ on $F$.

Since we suppose $t < 1$,

$$\frac{\mathrm{d}}{\mathrm{d}t} F(\mu_t) = \lim_{s \to 0} \frac{F(\mu_{t+s}) - F(\mu_t)}{s}$$

$$= \frac{1}{1-t} \lim_{s \to 0} \frac{F(\mu_{t+(1-t)s}) - F(\mu_t)}{s}$$

$$= \frac{1}{1-t} \frac{\mathrm{d}}{\mathrm{d}s} \Big|_{s=0} F(\mu_{t+(1-t)s}).$$

Here, it holds that

$$\mu_{t+(1-t)s} = ((1 - (t + (1-t)s))p_1 + (t + (1-t)s)p_2)\#\gamma$$

$$= ((1-s)((1-t)p_1 + tp_2) + sp_2)\#\gamma$$

$$= ((1-s)p_1 + sp_2)\#(((1-t)p_1 + tp_2) \times p_2)\#\gamma.$$

Then, by $(*)$ for $\mu \leftarrow \mu_t$, $\nu \leftarrow \nu, \gamma \leftarrow ((1-t)p_1 + tp_2 \times \mathrm{Id})\#\gamma$, we have

$$\frac{\mathrm{d}}{\mathrm{d}t} F(\mu_t) = \frac{1}{1-t} \lim_{s \to 0} \frac{F(\mu_{t+(1-t)s}) - F(\mu_t)}{s}$$

$$= \frac{1}{1-t} \int \langle \nabla_\mu F(\mu_t, x), y - x \rangle \, ((1-t)p_1 + tp_2 \times \mathrm{Id})\#\gamma(\mathrm{d}x\mathrm{d}y)$$

$$= \frac{1}{1-t} \int \langle \nabla_\mu F(\mu_t, (1-t)x + ty), y - ((1-t)x + ty) \rangle \, \gamma(\mathrm{d}x\mathrm{d}y)$$

$$= \int \langle \nabla_\mu F(\mu_t, (1-t)x + ty), y - x \rangle \, \gamma(\mathrm{d}x\mathrm{d}y).$$

$\square$

**Proposition C.7.** *Suppose* $F : \mathcal{P}_2(\mathbb{R}^d)$ *is sufficiently smooth. Let* $\mu, \nu \in \mathcal{P}_2(\mathbb{R}^d)$, *and let* $\mu_h$ *be a constant geodesic induced by* $\gamma \in \Gamma_0(\mu, \nu)$. *Then, as* $h \to 0$,

$$F(\mu_h) - F(\mu) = h \int \langle \nabla_\mu F(\mu, x), y - x \rangle \, \gamma(\mathrm{d}x\mathrm{d}y)$$

$$+ \frac{h^2}{2} \int \langle y_1 - x_1, \nabla_\mu^2 F(\mu, x_1, x_2)(y_2 - x_2) \rangle \, \gamma(\mathrm{d}x_1\mathrm{d}y_1)\gamma(\mathrm{d}x_2\mathrm{d}y_2)$$

$$+ \frac{h^2}{2} \int \langle y - x, \nabla\nabla_\mu F(\mu, x)(y - x) \rangle \, \gamma(\mathrm{d}x\mathrm{d}y) + o(h^2).$$

*Proof.* By Lemma C.6,

$$\frac{\mathrm{d}^2}{\mathrm{d}h^2} \Big|_{h=0} F(\mu_h) = \frac{\mathrm{d}}{\mathrm{d}h} \Big|_{h=0} \int \langle \nabla_\mu F(\mu_h, (1-h)x + hy), y - x \rangle \, \gamma(\mathrm{d}x\mathrm{d}y)$$

$$= \int \left\langle \frac{\mathrm{d}}{\mathrm{d}h} \Big|_{h=0} \nabla_\mu F(\mu_h, (1-h)x + hy), y - x \right\rangle \gamma(\mathrm{d}x\mathrm{d}y)$$

$$= \int \left\langle \int \nabla_\mu^2 F(\mu, x_1, x_2)(y_1 - x_1)\gamma(\mathrm{d}x_1\mathrm{d}y_1), y_2 - x_2 \right\rangle \gamma \mathrm{d}x_2\mathrm{d}y_2$$

$$+ \int \langle \nabla\nabla_\mu F(\mu, x)(y - x), y - x \rangle \, \gamma(\mathrm{d}x\mathrm{d}y)$$

$$= \iint \langle y_1 - x_1, \nabla_\mu^2 F(\mu, x_1, x_2)(y_2 - x_2) \rangle \, \gamma^{\otimes 2}(\mathrm{d}x_1\mathrm{d}y_1\mathrm{d}x_2\mathrm{d}y_2)$$

$$+ \int \langle \nabla\nabla_\mu F(\mu, x)(y - x), y - x \rangle \, \gamma(\mathrm{d}x\mathrm{d}y).$$

$\square$

*Remark* C.8. Except for the term $\nabla\nabla_\mu F$, this expression can be interpreted in a manner similar to Taylor expansion in Euclidean space. The term $\nabla\nabla_\mu F$ arises from the change in the metric $\mu$ of

the tangent space. This phenomenon is similar to what occurs on Riemannian manifolds, where the metric of the tangent spaces is not necessarily constant. It is worth noting that a similar calculation is performed in Appendix 5 of Bonnet (2019). As stated in the main text, at first-order stationary points, we have $\nabla_\mu F = 0$, which implies $\nabla \nabla_\mu F = 0$ Therefore, second-order optimality at stationary points can be understood in terms of the integral operator property of $\nabla^2_\mu F$.

By setting $\nu = (\mathrm{Id} + hv)\#\mu$ in Proposition C.7, we obtain the following:

**Proposition C.9.** *Suppose $F : \mathcal{P}_2(\mathbb{R}^d) \to \mathbb{R}$ is sufficiently smooth. Then, it holds that for all $v \in L^2(\mu)^d$,*

$$F((\mathrm{Id} + hv)\#\mu) - F(\mu) = h \langle \nabla_\mu F(\mu), v \rangle_{L^2(\mu)} + \frac{h^2}{2} \langle v, (H_\mu + H'_\mu)v \rangle_{L^2(\mu)} + o(h^2) \quad (h \to 0).$$

For the next lemma, we denote the interior and boundary of a set $A \subset \mathbb{R}^d$ as $A^o$ and $\partial A$, respectively.

**Lemma C.10.** *Suppose that $\mu \in \mathcal{P}_2^a(\mathbb{R}^d)$ and that $f : \mathbb{R}^d \to \mathbb{R}$ is of class $C^1(\mathbb{R}^d)$. Then, it holds that*

$$f = 0 \quad \mu\text{-a.e.} \implies \nabla f = 0 \quad \mu\text{-a.e.}$$

*Proof.* Let $B \subset \mathbb{R}^d$ be a closed set $B = \{x \in \mathbb{R}^d \mid f(x) = 0\}$. For all $x \in B^o \subset B$, there exists $r > 0$ such that

$$\|y - x\| < r \implies y \in B^o \implies f(y) = 0.$$

Then we have $\nabla f(x) = 0$. Thus,

$$\begin{aligned}
\mu(\{x \in \mathbb{R}^d | \nabla f(x) = 0\}) &\geq \mu(B^o) \\
&= \mu(B) - \mu(\partial B) \\
&= 1 - \mu(\partial B).
\end{aligned}$$

Since $\mu$ is absolutely continuous with respect to Lebesgue measure and $B$ is a Jordan measurable set, $\mu(\partial B) = 0$ holds. Hence, $\nabla f(x) = 0$ $\mu$-a.e. $\qquad\square$

**Proposition C.11** (Second-order necessary condition, restatement of Proposition 3.3). *Let $F : \mathcal{P}_2(\mathbb{R}^d) \to \mathbb{R}$ be sufficiently smooth. If $\mu^* \in \mathcal{P}_2^a(\mathbb{R}^d)$ be a local minimum of F, i.e. it holds that there exists a constant $r > 0$ such that*

$$W_2(\mu, \mu^*) < r \implies F(\mu) \leq F(\mu^*).$$

*Then it holds that $H_{\mu^*} \succeq O$.*

*Proof.* According to Proposition C.4, $\nabla_\mu F(\mu^*) = 0$ $\mu$-a.e. Then it follows from Lemma C.10 that $\nabla \nabla_\mu F(\mu^*) = 0$ a.e.

For any vector field $v \in L^2(\mu^*)$, there exists a constant $\bar{h} > 0$ such that $h \leq \bar{h} \implies W_2(\mu^*, (\mathrm{Id} + hv)\#\mu^*) = h\|v\|_{L^2(\mu^*)} \leq r$. By applying Proposition C.9, we have

$$\langle v, H_{\mu^*} v \rangle_{L^2(\mu^*)} = F((\mathrm{Id} + hv)\#\mu^*) - F(\mu^*) + o(1) \geq o(1).$$

By letting $h \to 0$, we have $\langle v, H_{\mu^*} v \rangle_{L^2(\mu^*)} \geq 0$ i.e. $H_{\mu^*} \succeq O$. $\qquad\square$

# D  Kernels and Gaussian Processes

The purpose of this section is to present a series of propositions regarding positive semi-definite kernels, the integral operators they define, and Gaussian processes. Additionally, we aim to derive an inequality that evaluates the tail probability of the $L^2(\mu)$ norm of a Gaussian process.

In the following propositions, the kernel function $K : \mathbb{R}^d \times \mathbb{R}^d \to \mathbb{R}^{d \times d}$ of interest corresponds to the integral kernel $K_\mu$ of the squared Hessian defined in (11). However, we first examine the properties of general kernels and Gaussian processes, applying these results to $K_\mu$ defined in (11) at the end of this section.

In this paper, we consider multivariate Gaussian processes. Kim & Suzuki (2024) were the first to propose introducing random perturbations to probability measures using a multivariate Gaussian process. Another application of multivariate Gaussian processes is modeling vector fields on Riemannian manifolds (Hutchinson et al., 2021). For detailed definition and properties, please refer to Chen et al. (2023).

**Definition D.1** (Multivariate Gaussian process). The vector-valued function $\xi : \mathbb{R}^d \to \mathbb{R}^d$ is said to follow a multivariate Gaussian process if any finite collection of variables $\xi(x_1), \cdots, \xi(x_N)$ are jointly normally distributed. This process is determined by the vector-valued mean function $m : \mathbb{R}^d \to \mathbb{R}^d$ and the matrix-valued covariance function $K : \mathbb{R}^d \times \mathbb{R}^d \to \mathbb{R}^{d \times d}$:

$$m(x) = \mathbb{E}[\xi(x)] \quad (x \in \mathbb{R}^d),$$
$$K(x, \tilde{x}) = \mathbb{E}\big[(\xi(x) - m(x))(\xi(x) - m(x))^\top\big] \quad (x, \tilde{x} \in \mathbb{R}^d).$$

In this case, we denote $\xi \sim \mathrm{GP}(m, K)$.

**Proposition D.2.** Suppose $K : \mathbb{R}^d \times \mathbb{R}^d \to \mathbb{R}^{d \times d}$ satisfies $\int \|K(x, y)\| \, \mu^{\otimes 2}(\mathrm{d}x \mathrm{d}y) < \infty$ and $K(x, y)^\top = K(y, x)$ for all $x, y \in \mathbb{R}^d$. Then, there exists a sequence $\{\kappa_n\}_{n \geq 1} \subset \mathbb{R}$, which is finite or converges to $0$, satisfies $\|T_K\|_{\mathrm{Tr}, L^2(\mu)} = \sum_{n \geq 1} |\kappa_n| < \infty$, and is non-increasing. Furthermore, there exists an orthonormal basis $\{\psi_n\}_{n \geq 1} \subset L^2(\mathbb{R}^d)^d$ such that

$$K(x, y) = \sum_{n \geq 1} \kappa_n \psi_n(x) \psi_n(y)^\top,$$

where the infinite sum converges in the $L^2(\mu)^{d \times d}$ norm.

*Proof.* This follows from the eigenvalue expansion theorem for compact self-adjoint operators on Hilbert spaces. The trace-class property $\sum_{n \geq 1} |\kappa_n| < \infty$ follows from $T_k$ being trace-class. $\square$

**Proposition D.3.** Let $\mu \in \mathcal{P}_2(\mathbb{R}^d)$ and $\psi \in L^2(\mu)^d$. Suppose $K : \mathbb{R}^d \times \mathbb{R}^d \to \mathbb{R}^{d \times d}$ is a positive semi-definite kernel satisfying $\int \|K(x, y)\| \, \mu^{\otimes 2}(\mathrm{d}x \mathrm{d}y) < \infty$. Then, $\xi \sim \mathrm{GP}(0, K)$ satisfies $\xi \in L^2(\mu)^d$ almost surely.

*Proof.* Using the integrability of $k$ and equivalence of matrix norms:

$$\mathbb{E}[\|\xi\|_\mu^2] = \int \mathbb{E}[\xi(x)^\top \xi(x)] \, \mu(\mathrm{d}x) = \int \mathrm{tr}(K(x, x)) \, \mu(\mathrm{d}x) < \infty.$$

Setting $A_n = \{\|\xi\|_\mu^2 \geq n\}$ for $n \in \mathbb{N}$, it holds that

$$P(\|\xi\|_\mu^2 < \infty) = 1 - P\left(\bigcap_{n \geq 1} A_n\right) = 1 - \lim_{n \to \infty} P(A_n) \geq 1 - \limsup_{n \to \infty} \frac{1}{n} \mathbb{E}[\|\xi\|_\mu^2] = 1.$$

Thus, $\xi \in L^2(\mu)^d$ almost surely. $\square$

**Proposition D.4.** Let $\mu \in \mathcal{P}_2(\mathbb{R}^d)$ and $\psi \in L^2(\mu)^d$. Suppose $K : \mathbb{R}^d \times \mathbb{R}^d \to \mathbb{R}^{d \times d}$ is a positive semi-definite kernel satisfying $\int \|K(x, y)\| \, \mu^{\otimes 2}(\mathrm{d}x \mathrm{d}y) < \infty$. Then, for any $\xi \sim \mathrm{GP}(0, K)$, we have $\langle \psi, \xi \rangle_{L^2(\mu)} \sim \mathcal{N}(0, \langle \psi, T_K \psi \rangle_{L^2(\mu)})$.

*Proof.* The proof follows Kim & Suzuki (2024), Lemma E.9. First, we show that $\langle \psi, \xi \rangle_{L^2(\mu)}$ is normally distributed.

We define the closed subspace of square-integrable real-valued random variables by $E = \overline{\text{span}}\{\psi(x)^\top \xi(x) \mid x \in \mathbb{R}^d\}$. For any $Z \in E^\perp$, the following holds:

$$\mathbb{E}[Z\langle \psi, \xi \rangle_{L^2(\mu)}] = \int \mathbb{E}[Z\psi(x)^\top \xi(x)]\, \mu(\mathrm{d}x) = 0.$$

Thus, $\langle \psi, \xi \rangle_{L^2(\mu)} \in (E^\perp)^\perp = E$ holds, meaning that $\langle \psi, \xi \rangle_{L^2(\mu)}$ is the $L^2(P)$ limit, and therefore, the law convergence limit of of normally distributed random variables. As will be shown later, the mean of $\langle \psi, \xi \rangle_{L^2(\mu)}$ is 0, and its variance is $\langle \psi, T_K \psi \rangle_{L^2(\mu)}$. Therefore, the characteristic function converges pointwise to the characteristic function of a normal distribution, implying that $\langle \psi, \xi \rangle_{L^2(\mu)}$ follows a normal distribution.

Moreover, the mean and variance are computed as:

$$\mathbb{E}[\langle \psi, \xi \rangle_{L^2(\mu)}] = \int \psi(x)^\top \mathbb{E}[\xi(x)]\, \mu(\mathrm{d}x) = 0,$$

$$\mathbb{E}[\langle \psi, \xi \rangle^2_{L^2(\mu)}] = \int \psi(x)^\top \mathbb{E}[\xi(x)\xi(y)^\top]\psi(y)\, \mu^{\otimes 2}(\mathrm{d}x\mathrm{d}y)$$

$$= \int \psi(x)^\top T_K \psi(x)\, \mu(\mathrm{d}x) = \langle \psi, T_K \psi \rangle_{L^2(\mu)}.$$

Here, Fubini's theorem and the definition of Gaussian processes are used. $\qquad\square$

**Proposition D.5** (Karhunen–Loève expansion). *Suppose $K : \mathbb{R}^d \times \mathbb{R}^d \to \mathbb{R}^{d\times d}$ be a positive semidefinite kernel and satisfy $\int \|K(x, y)\| \mu^{\otimes 2}(\mathrm{d}x\mathrm{d}y) < \infty$. Let $\{\kappa_n\}_{n\geq 1} \subset \mathbb{R}_{\geq 0}$ and $\{\psi_n\}_{n\geq 1} \subset L^2(\mu)^d$ be as in Lemma D.2. The sequence of the random variables $X_n := \langle \psi_n, \xi \rangle_{L^2(\mu)} \sim \mathcal{N}(0, \kappa_n)$ is mutually independent. Furthermore, the Gaussian process $\xi \sim \mathrm{GP}(0, K)$ is represented as follows:*

$$\xi(x) = \sum_{n\geq 1} X_n \psi_n(x),$$

*where the right-hand infinite sum means convergence in $L^2(P \otimes \mu)$.*

*Proof.* First, we show that the sequence of the random variables $X_n = \langle \psi_n, \xi \rangle_{L^2(\mu)}$ $(n \geq 1)$ is mutually independent.

$$\mathrm{E}[X_n X_m] = \mathrm{E}\left[\int \psi_n(x)^\top \xi(x)\mu(\mathrm{d}x) \int \xi(y)^\top \psi_m(y)\mu(\mathrm{d}y)\right]$$

$$= \int \psi_n(x)^\top \mathrm{E}[\xi(x)\xi(y)^\top]\psi_m(x)\mu \otimes \mu(\mathrm{d}x\mathrm{d}y)$$

$$= \int \psi_n(x)^\top \left(\int K(x, y)\psi_m(y)\mu(\mathrm{d}y)\right)\mu(\mathrm{d}x)$$

$$= \int \psi_n(x)^\top (\kappa_m \psi_m(x))\mu(\mathrm{d}x)$$

$$= \kappa_m \delta_{n,m},$$

where Fubini's theorem is used in the second and fourth line. The fifth line follows from the fact that $\psi_m$ is the eigenvector of the integral operator $T_K$.

Thus $n \neq m$ implies the covariance of $X_n$ and $X_m$ is equal to 0. This implies that $X_n$ and $X_m$ are mutually independent, because they follow the normal distribution. Hence we obtain the mutual independency of the sequence $X_n = \langle \psi_n, \xi \rangle_{L^2(\mu)}$ $(n \geq 1)$.

The Karhunen-Loève expansion follows from

$$\mathrm{E}\left[\left\|\xi - \sum_{n=1}^{N} X_n \psi_n\right\|_{L^2(\mu)}^2\right] = \mathrm{E}\left[\|\xi\|_{L^2(\mu)}^2\right] - 2\sum_{n=1}^{N} \mathrm{E}\left[X_n \langle \xi, \psi_n \rangle_{L^2(\mu)}\right]$$

$$+ \sum_{n=1}^{N}\sum_{m=1}^{N} \langle \psi_n, \psi_m \rangle_{L^2(\mu)} \mathrm{E}[X_n X_m]$$

$$= \mathrm{E}\left[\|\xi\|_{L^2(\mu)}^2\right] - \sum_{n=1}^{N} \mathrm{E}\left[X_n^2\right]$$

$$= \sum_{n=N+1}^{\infty} \kappa_n \xrightarrow{N\to\infty} 0.$$

$\square$

**Proposition D.6** (extended Markov's inequality). *For any random variable $X$ and non-decreasing positive-valued measurable function $\varphi : \mathbb{R} \to \mathbb{R}_{>0}$, the following holds :*

$$P(X \geq M) \leq \frac{\mathrm{E}[\varphi(X)]}{\varphi(M)} \quad (\forall M \in \mathbb{R}).$$

*Proof.* By applying Markov's inequality, we have

$$P(X \geq M) \leq \Pr(\varphi(X) \geq \varphi(M)) \leq \frac{\mathrm{E}[\varphi(X)]}{\varphi(M)} \quad (\forall M \in \mathbb{R}).$$

$\square$

With the above preparations, we obtain an upper bound for the tail probability of the $L^2$ norm of the Gaussian process.

**Proposition D.7.** *Suppose $K : \mathbb{R}^d \times \mathbb{R}^d \to \mathbb{R}^{d\times d}$ be a positive semi-definite symmetric kernel and satisfy $\int \|K(x,y)\| \mu^{\otimes 2}(\mathrm{d}x\mathrm{d}y) < \infty$ holds. Let $\{\kappa_n\}_{n\geq 1} \subset \mathbb{R}_{\geq 0}$ and $\{\psi_n\}_{n\geq 1} \subset L^2(\mu)^d$ be as in Proposition D.2. The Gaussian process $\xi \sim \mathrm{GP}(0, K)$ with kernel function $K$ satisfies the following:*

$$P\left(\|\xi\|_{L^2(\mu)} \geq M\right) \leq \exp\left(-\frac{e-1}{2e\kappa_1}M^2 + \frac{\sum_{n\geq 1}\kappa_n}{2\kappa_1}\right) \quad \left(\forall M > \|T_K\|_{\mathrm{Tr},L^2(\mu)}^{\frac{1}{2}}\right).$$

*Proof.* First, we calculate the characteristic function of $\|\xi\|_{L^2(\mu)}^2$.

Since $\xi \sim \mathrm{GP}(0, K)$, by Lemma D.5, we can express $\|\xi\|_{L^2(\mu)}^2 = \sum_{n\geq 1} X_n^2 =: Y$ $(X_n \overset{\mathrm{i.d.}}{\sim} \mathcal{N}(0, \kappa_n) \, \forall n \geq 1)$. Let $Y_N = \sum_{n=1}^{N} X_n^2$. Then for any $s \in \mathbb{C}$, we have

$$\mathrm{E}[e^{sX_n^2}] = \frac{1}{\sqrt{1 - 2\kappa_n s}},$$

and so, by the independence of $\{X_n\}_{n\geq 1}$,

$$\mathrm{E}[e^{sY_N}] = \prod_{n=1}^{N} \frac{1}{\sqrt{1 - 2\kappa_n s}}.$$

In particular, setting $s = \mathrm{i}u$ $(u \in \mathbb{R})$, the characteristic function of $Y_N$ is

$$\log \mathrm{E}[e^{\mathrm{i}uY_N}] = -\frac{1}{2}\sum_{n=1}^{N} \log(1 - 2\mathrm{i}\kappa_n u). \tag{23}$$

Let $\theta_n = \mathrm{Arg}(1 - 2\mathrm{i}\kappa_n u)$ $(-\frac{\pi}{2} < \theta_n < \frac{\pi}{2})$. Then for the real and imaginary parts,

$$\left| \mathrm{Re} \sum_{n=1}^{N} (\log(1 - 2\mathrm{i}\kappa_n u)) \right| \leq \frac{1}{2} \sum_{n=1}^{N} \log\left(1 + 4\kappa_n^2 u^2\right) \leq \frac{1}{2} \sum_{n=1}^{N} 4\kappa_n{}^2 u^2$$

$$\leq 2u^2 \sum_{n\geq 1} \kappa_n{}^2 \leq 2u^2 \left( \sum_{n\geq 1} \kappa_n \right)^2 < \infty,$$

$$\left| \mathrm{Im} \sum_{n=1}^{N} (\log(1 - 2\mathrm{i}\kappa_n u)) \right| \leq \sum_{n=1}^{N} |\theta_n| \leq \sum_{n=1}^{N} |\tan\theta_n| = \sum_{n=1}^{N} 2\kappa_n |u| \leq 2|u| \sum_{n\geq 1} \kappa_n < \infty.$$

Thus, the characteristic function of $Y_N$ converges pointwise as $N \to \infty$. By Levy's continuity theorem, $Y_N$ weakly converges to $Y = \lim_{N\to\infty} Y_N = \sum_{n\geq 1} X_n{}^2$, and its characteristic function is $\log \mathrm{E}[e^{\mathrm{i}uY}] = -\frac{1}{2} \sum_{n\geq 1} \log(1 - 2\mathrm{i}\kappa_n u)$.
Now, we will proceed to

- Show that for some $t > 0$, replacing $\mathrm{i}u$ with $t$ makes the series on the right-hand side of (23) converge. This allows us to show the existence of the moment generating function $\mathrm{E}[e^{tY}]$ for such $t$.

- Use Lemma D.6 with $\varphi(x) \leftarrow e^{tx}$, $X \leftarrow Y$ to obtain the upper bound for the tail probability.

- Adjust $t > 0$ (within the range where the moment generating function exists) to obtain the best possible upper bound for the tail probability.

The function $t \mapsto -\frac{1}{2} \log(1 - 2\kappa_n t)$ is convex, and the tangent at $t = 0$ is $t \mapsto \kappa_n t$. Thus for $a > 1$, the equation for $t$;

$$a\kappa_n t = -\frac{1}{2} \log(1 - 2\kappa_n t)$$

has a solution $t = t_n > 0$ for $t > 0$, and it holds that

$$0 \leq t \leq t_n \implies -\frac{1}{2} \log(1 - 2\kappa_n t) \leq a\kappa_n t.$$

Since $\kappa_n t_n$ is constant for $n \geq 1$ and $\{\kappa_n\}_{n\geq 1}$ is non-increasing, the sequence $\{t_n\}_{n\geq 1}$ is non-decreasing. Hence, for any $n \geq 1$, $t \in [0, t_1]$, we have $-\frac{1}{2} \log(1 - 2\kappa_n t) \leq a\kappa_n t$, so

$$\log \mathrm{E}[e^{t\|\xi\|_\mu^2}] \leq at \sum_{n\geq 1} \kappa_n < \infty.$$

Therefore, for $t \in [0, t_1]$, the moment generating function $\mathrm{E}[e^{t\|\xi\|_\mu^2}]$ exists, and $\mathrm{E}[e^{t\|\xi\|_\mu^2}] \leq e^{at \sum_{n\geq 1} \kappa_n}$. Hence, for the upper bound of the tail probability of the Gaussian process norm, we have

$$P\left( \|\xi\|_\mu \geq M \right) \leq \frac{\mathrm{E}[e^{t\|\xi\|_\mu^2}]}{e^{tM^2}}$$

$$\leq \exp\left( -tM^2 + at \sum_{n\geq 1} \kappa_n \right).$$

By optimizing the right-hand side of this expression with respect to $t$, $a$, we obtain the best upper bound. For $a > 1$, $at\kappa_1 = -\frac{1}{2} \log(1 - 2\kappa_1 t)$, considering large $M$, we solve

$$\text{minimize } f(t) = -tM^2 + at \sum_{n\geq 1} \kappa_n, \tag{24}$$

$$\text{s.t. } t > 0, \quad a = -\frac{1}{2\kappa_1 t} \log(1 - 2\kappa_1 t) > 1. \tag{25}$$

The derivative of (24) is

$$\frac{\mathrm{d}}{\mathrm{d}t}f(t) = -M^2 + \frac{\sum_{n\geq 1}\kappa_n}{1 - 2\kappa_1 t} = 0, \tag{26}$$

which gives

$$t := t^* = \frac{1}{2\kappa_1}\left(1 - \frac{\sum_{n\geq 1}\kappa_n}{M^2}\right). \tag{27}$$

If $M^2 > \sum_{n\geq 1}\kappa_n$, then $t^* > 0, a > 1$, satisfying the condition (25). The upper bound becomes

$$P\Big(\|\xi\|_{L^2(\mu)} \geq M\Big) \leq \exp f(t^*)$$

$$= \exp\left(-t^* M^2 - \frac{\sum_{n\geq 1}\kappa_n}{2\kappa_1}\log(1 - 2\kappa_1 t^*)\right)$$

$$= \exp\left(-\frac{M^2}{2\kappa_1} + \frac{\sum_{n\geq 1}\kappa_n}{2\kappa_1} + \frac{\sum_{n\geq 1}\kappa_n}{2\kappa_1}\log\frac{M^2}{\sum_{n\geq 1}\kappa_n}\right),$$

where in the third line we used (26), (27). Finally, applying the inequality $-x + \log x \leq -\frac{e-1}{e}x$ at $x = \frac{M^2}{\sum_{n\geq 1}\kappa_n}$, we obtain

$$f(t^*) \leq \frac{\sum_{n\geq 1}\kappa_n}{2\kappa_1}\left(1 - \frac{e-1}{e}\frac{M^2}{\sum_{n\geq 1}\kappa_n}\right).$$

So, we conclude

$$P\Big(\|\xi\|_{L^2(\mu)} \geq M\Big) \leq \exp\left(-\frac{e-1}{2e\kappa_1}M^2 + \frac{\sum_{n\geq 1}\kappa_n}{2\kappa_1}\right).$$

$\square$

We apply the results from the previous section.

**Lemma D.8.** *Let $K_\mu$ be the Hessian-based kernel introduced in (11). Then, the Gaussian process $\xi \sim \mathrm{GP}(0, K_\mu)$ satisfies the following :*

- $\langle \psi_n, \xi \rangle_{L^2(\mu^\dagger)} \sim \mathcal{N}(0, \lambda_n^2)$.

- *P-almost surely, it occurs that $\xi \in \mathcal{R}(H_\mu) \subset \mathrm{Tan}_\mu \mathcal{P}_2(\mathbb{R}^d)$. In particular, from the assumptions, $\mathrm{Id} + \eta_p \xi$ gives the optimal transport map from $\mu$ to $(\mathrm{Id} + \eta_p \xi)\#\mu$ for sufficiently small $\eta_p > 0$ P-a.s.*

- *For any constant $M > R_2 \geq \left(\sum_{n\geq 1}\lambda_n^2\right)^{\frac{1}{2}}$, the following holds :*

$$P\Big(\|\xi\|_{L^2(\mu)} \geq M\Big) \leq \exp\left(-\frac{e-1}{2e\lambda_1^2}M^2 + \frac{\sum_{n\geq 1}\lambda_n^2}{2\lambda_1^2}\right).$$

*Proof.* Given the assumption $\int \|\nabla_\mu^2 F(\mu, x, y)\|\mu^{\otimes 2}(\mathrm{d}x\mathrm{d}y) < \infty$, Lemma D.2 can be applied to $K = \nabla_\mu^2 F(\mu)$. Specifically, there exists a sequence of real numbers $\{\lambda_n\}_{n\geq 1} \subset \mathbb{R} \setminus \{0\}$ satisfying $\|H_\mu\|_{\mathrm{HS},L^2(\mu)}^2 = \sum_{n\geq 1}\lambda_n^2 < \infty$ and an orthonormal basis $\{\psi_n\}_{n\geq 1}$ of $L^2(\mu)$, such that

$$\nabla_\mu^2 F(\mu, x, y) = \nabla_1 \nabla_2 \frac{\delta^2 F}{\delta\mu^2}(x, y) = \sum_{n\geq 1}\lambda_n \psi_n(x)\psi_n(y)^\top, \quad \mu^{\otimes 2}\text{-a.e. } (x, y). \tag{28}$$

Regarding the kernel of the squared Hessian $K_\mu$,

$$K_\mu(x, y) = \sum_{n\geq 1}\lambda_n^2 \psi_n(x)\psi_n(y)^\top, \quad \mu^{\otimes 2}\text{-a.e. } (x, y).$$

Here, $K_\mu$ satisfies the assumptions of Lemma D.5, corresponding to the case where $\kappa_n = \lambda_n^2$ in that lemma. Then, from Proposition D.4, $\langle \psi_n, \xi \rangle_{L^2(\mu)} \sim \mathcal{N}(0, \lambda_n^2)$ holds. Finally, it follows from Proposition D.7 that for any $M \geq R_2 \geq \|H_\mu\|_{\mathrm{HS}, L^2(\mu)} = \left\|H_\mu^2\right\|_{\mathrm{Tr}, L^2(\mu)} = \left(\sum_{n \geq 1} {\lambda_n}^2\right)^{\frac{1}{2}}$,

$$P\left(\|\xi\|_{L^2(\mu)} \geq M\right) \leq \exp\left(-\frac{e-1}{2e{\lambda_1}^2}M^2 + \frac{\sum_{n \geq 1} {\lambda_n}^2}{2{\lambda_1}^2}\right).$$

$\square$

# E Proofs for Convergence Analysis

The lemmas for the main theorem are re-stated and the proofs are provided here.

## E.1 Continuous-time Convergence Analysis

### E.1.1 Proof Sketch for Lemmas

The proof of Lemma 5.3 is relatively straightforward using the chain rule. In contrast, the proof of Lemma 5.4 is lengthy and relies on two sub-lemmas inspired by the analysis of SSRGD (Li, 2019) in Euclidean spaces.

**Lemma E.1: small increase by perturbation.** This lemma ensures that the increase in the objective function due to perturbation is upper bounded by $F_{\mathrm{thres}}$ with high probability.

**Lemma E.2: large decrease by WGF.** Put simply, this lemma asserts that a small deviation in the direction of the eigenvector corresponding to the smallest eigenvalue of the Hessian results in exponential decrease in the objective under WGF. This result is derived from the fact that the dynamics of slightly deviated two points evolve approximately under $H_{\mu^\dagger}$, causing their "distance" to grow exponentially over time.

By combining these results, namely that the objective function does not increase significantly and decreases substantially after perturbation, the lemma is proven.

### E.1.2 Proof of Lemma 5.3

**Lemma 5.3.** *For a curve of probability measures $\mu_t$ following the WGF, the following holds:*

$$F(\mu_0) - F(\mu_t) = \int_0^t \|\nabla_\mu F(\mu_\tau)\|_{\mu_\tau}^2 \, \mathrm{d}\tau.$$

*Proof.* We have

$$F(\mu_0) - F(\mu_t) = \int_0^t -\frac{\mathrm{d}}{\mathrm{d}\tau} F(\mu_\tau) \mathrm{d}\tau = \int_0^t \|\nabla_\mu F(\mu_\tau)\|_{L^2(\mu_\tau)}^2 \mathrm{d}\tau,$$

due to the chain rule. □

### E.1.3 Proof of Proposition 5.4

**Proposition 5.4.** *Set $\eta = O(1)$ and let $\varepsilon$, $\delta$, $\eta_p$, $T_{\mathrm{thres}}$, $F_{\mathrm{thres}}$ be chosen as in Theorem 5.2. Suppose $\mu^\dagger \in \mathcal{P}_2^a(\mathbb{R}^d)$ satisfies $\left\|\nabla_\mu F(\mu^\dagger)\right\|_{L^2(\mu^\dagger)} < \varepsilon$ and $\lambda_0 := \lambda_{\min} H_{\mu^\dagger} \le -\delta$. Generating $\xi \sim \mathrm{GP}(0, k_\mu)$ and setting $\mu_0 = (\mathrm{Id} + \eta_p \xi)\sharp\mu^\dagger$ as the initial point of the WGF, we have with probability $1 - \zeta'$:*

$$F(\mu^\dagger) - F(\mu_{T_{\mathrm{thres}}}) \ge F_{\mathrm{thres}}.$$

Intuitively, Proposition 5.4 indicates that when a perturbation is applied near the saddle point, the objective function decreases over a period of time $T_{\mathrm{thres}}$ with high probability, which corresponds to escaping the saddle point. As discussed above, the approach is based on the argument by Li (2019) and utilizes Lemma E.1 and Lemma E.2. These two lemmas postulate that the $L^2$-norm $\|\xi\|_{L^2(\mu)}$ of the Gaussian process $\xi$ is uniformly bounded. This corresponds to the condition that perturbations in Euclidean space are sampled from spheres with a fixed radius, thus the perturbation size was uniformly bounded. In infinite-dimensional spaces, such uniform sampling cannot be used, which is one of the reasons why the Gaussian process is employed. In this case, the norm of the Gaussian process can take arbitrarily large values even though with low probability. Therefore, it is necessary to exploit tail probability estimates of the norm of the Gaussian process, as in the following.

**Lemma D.8.** *Let $K_\mu$ be the Hessian-based kernel introduced in (11). Then, the Gaussian process $\xi \sim \mathrm{GP}(0, K_\mu)$ satisfies the following :*

- $\langle \psi_n, \xi \rangle_{L^2(\mu^\dagger)} \sim \mathcal{N}(0, {\lambda_n}^2)$.

- *P-almost surely, it occurs that $\xi \in \mathcal{R}(H_\mu) \subset \mathrm{Tan}_\mu \mathcal{P}_2(\mathbb{R}^d)$. In particular, from the assumptions, $\mathrm{Id} + \eta_p \xi$ gives the optimal transport map from $\mu$ to $(\mathrm{Id} + \eta_p \xi) \# \mu$ for sufficiently small $\eta_p > 0$ P-a.s.*

- *For any constant $M > R_2 \geq \left( \sum_{n \geq 1} {\lambda_n}^2 \right)^{\frac{1}{2}}$, the following holds :*

$$P\Big( \|\xi\|_{L^2(\mu)} \geq M \Big) \leq \exp\left( -\frac{e-1}{2e{\lambda_1}^2} M^2 + \frac{\sum_{n \geq 1} {\lambda_n}^2}{2{\lambda_1}^2} \right).$$

Let [5]

$$M = \left( \frac{e{R_1}^2}{e-1} \left( 1 + 2\log \frac{2}{\zeta'} \right) \right)^{\frac{1}{2}} \vee 2R_1 \left( \log \frac{4\sqrt{2}}{\zeta'} \right)^{\frac{1}{2}} = \tilde{O}(1). \tag{29}$$

From Assumption 1,

$${R_1}^2 \geq \int \left\| \nabla_\mu^2 F(\mu, x, y) \right\|_{\mathrm{F}}^2 \mu^{\otimes 2}(\mathrm{d}x\mathrm{d}y) = \sum_{n \geq 1} {\lambda_n}^2 \geq {\lambda_1}^2,$$

so that

$$\begin{aligned}
M^2 &\geq \frac{e{R_1}^2}{e-1} \left( 1 + 2\log \frac{2}{\zeta'} \right) \\
&\geq \frac{e}{e-1} \sum_{n \geq 1} {\lambda_n}^2 + \frac{2e}{e-1} {\lambda_1}^2 \log \frac{2}{\zeta'}
\end{aligned}$$

holds. Then it follows that

$$\exp\left( -\frac{e-1}{2e{\lambda_1}^2} M^2 + \frac{\sum_{n \geq 1} {\lambda_n}^2}{2{\lambda_1}^2} \right) \leq \frac{\zeta'}{2}.$$

Therefore, from Lemma D.8, $\|\xi\|_{L^2(\mu)} \leq \tilde{M} = \tilde{O}(1)$ occurs with probability at least $1 - \frac{\zeta'}{2}$.

We now take the hyperparameters $\eta_p$, $F_{\mathrm{thres}}$, $T_{\mathrm{thres}}$ as follows:

$$\begin{aligned}
T_{\mathrm{thres}} &= \frac{2}{\delta} \log \frac{16 L_1^{\frac{1}{2}} M}{\sqrt{e} \delta^{\frac{1}{2}} r} = \tilde{O}\left( \frac{1}{\delta} \right) \\
&= O\left( \frac{1}{\delta} \log \frac{\log \frac{1}{\zeta'}}{\delta^{\frac{1}{2}} \zeta'} \right), \\
\eta_p &= \frac{2 F_{\mathrm{thres}}}{M(\varepsilon + \sqrt{\varepsilon^2 + 2 L_1 F_{\mathrm{thres}}})} = \tilde{O}\left( \frac{\delta^3}{\varepsilon} \wedge \delta^{\frac{3}{2}} \right) \tag{30} \\
&= O\left( \delta^3 \varepsilon^{-1} \left( \log \frac{1}{\zeta'} \right)^{-1} \left( \log \frac{\log \frac{1}{\zeta'}}{\delta^{\frac{1}{2}} \zeta'} \right)^{-3} \wedge \delta^{\frac{3}{2}} \left( \log \frac{1}{\zeta'} \right)^{-1} \left( \log \frac{\log \frac{1}{\zeta'}}{\delta^{\frac{1}{2}} \zeta'} \right)^{-\frac{3}{2}} \right), \\
F_{\mathrm{thres}} &= \frac{T_{\mathrm{thres}}^{-3}}{18(L_2 + L_3)^2} \log^2 \frac{3}{2} = \tilde{O}(\delta^3) \\
&= O\left( \delta^3 \left( \log \frac{\log \frac{1}{\zeta'}}{\delta^{\frac{1}{2}} \zeta'} \right)^{-3} \right).
\end{aligned}$$

It should be noted that $F_{\mathrm{thres}} = \eta_p M \varepsilon + \frac{{L_1}^2}{2} {\eta_p}^2 M^2$ holds.

---

[5] The right-hand side implies the upper bound $\sqrt{2} \exp\left( -\frac{M^2}{4{\lambda_0}^2} \right) \leq \frac{\zeta'}{4}$, where $\lambda_0$ is the smallest eigenvalue of $H_\mu$. This result is utilized later in the proof of Proposition 5.4.

**Lemma E.1.** *Consider the situation of Proposition 5.4, and let $\mu^\dagger$ be an $(\varepsilon, \delta)$-saddle point, let the initial point be $\mu_0 := (\mathrm{Id} + \eta_p \xi)\#\mu^\dagger$, and the hyperparameters $\eta_p, F_{\mathrm{thres}}$ be taken as in (30). If the $L^2$-norm of the Gaussian process $\xi$ satisfies $\|\xi\|_{L^2(\mu)} \le M$, then it holds that*

$$F(\mu_0) - F(\mu^\dagger) \le F_{\mathrm{thres}}.$$

*Proof.* Let $\nu_h = (\mathrm{Id} + h\eta_p \xi)\sharp\mu^\dagger$ for $h \in [0,1]$, joining $\mu^\dagger$ and $\mu_0 = (\mathrm{Id} + \eta_p \xi)\#\mu^\dagger$. From Lemma C.6,

$$
\begin{aligned}
\frac{\mathrm{d}}{\mathrm{d}h}F(\mu_h) &= \eta_p \left\langle \nabla_\mu F(\nu_h) \circ (\mathrm{Id} + h\eta_p \xi), \xi \right\rangle_{L^2(\mu^\dagger)} \\
&\le \eta_p \|\nabla_\mu F(\nu_h) \circ (\mathrm{Id} + h\eta_p \xi)\|_{L^2(\mu^\dagger)} \|\xi\|_{L^2(\mu^\dagger)} \\
&\le \eta_p \Big( \|\nabla_\mu F(\mu^\dagger)\|_{L^2(\mu^\dagger)} + \|\nabla_\mu F(\nu_h) \circ (\mathrm{Id} + h\eta_p \xi) - \nabla_\mu F(\mu^\dagger)\|_{L^2(\mu^\dagger)} \Big) \|\xi\|_{L^2(\mu^\dagger)} \\
&\le \eta_p \left( \varepsilon + L_1 \left( \int \|x - y\|^2 (\mathrm{Id} \times (\mathrm{Id} + h\eta_p \xi))\#\mu^\dagger(\mathrm{d}x\mathrm{d}y) \right)^{\frac{1}{2}} \right) \|\xi\|_{L^2(\mu^\dagger)} \\
&= \eta_p \|\xi\|_{L^2(\mu^\dagger)}\varepsilon + L_1 \eta_p^2 \|\xi\|_{L^2(\mu^\dagger)}^2 h.
\end{aligned}
$$

Here, the Cauchy-Schwarz inequality was used in the second line, the conditions $\|\nabla_\mu F(\mu)\|_{L^2(\mu)} < \varepsilon$ and the gradient's Lipschitz continuity were applied in the fourth line, and Proposition B.4 was invoked in the fifth line. Therefore,

$$
\begin{aligned}
F(\mu_0) - F(\mu^\dagger) &= \int_0^1 \frac{\mathrm{d}}{\mathrm{d}h} F(\mu_h) \, \mathrm{d}h \\
&\le \eta_p \|\xi\|_{L^2(\mu^\dagger)}\varepsilon + \frac{L_1}{2}\eta_p^2 \|\xi\|_{L^2(\mu^\dagger)}^2 \\
&\le \eta_p M\varepsilon + \frac{L_1}{2}\eta_p^2 M^2 \\
&= F_{\mathrm{thres}}.
\end{aligned}
$$

The third line follows from $\|\xi\|_{L^2(\mu^\dagger)} \le M$ and the fourth line follows from the choice of $\eta_p$ in (30). $\qquad\square$

**Lemma E.2.** *Consider the situation of Proposition 5.4. That is, let $\mu^\dagger$ be an $(\varepsilon, \delta)$-saddle point, $\mu_0 := (\mathrm{Id} + \eta_p \xi)\#\mu^\dagger$, and the hyperparameters $\eta_p, F_{\mathrm{thres}}$ be taken as (30). Furthermore, we set $\tilde{\mu}_0 = (\mathrm{Id} + \eta_p \tilde{\xi})\sharp\mu^\dagger$, where $\tilde{\xi} = \xi + r\psi_0$, $r \in \mathbb{R}$ is a constant, and $\psi_0$ is the eigenvector of the Hessian operator $H_\mu$ corresponding to the smallest eigenvalue $\lambda_0$.*

*Letting $\mu_t$, $\tilde{\mu}_t$ be WGF initialized at $\mu_0$, $\tilde{\mu}_0$ respectively, then $\|\xi\|_{L^2(\mu^\dagger)} \le M$ and $\frac{\sqrt{2\pi}|\lambda_0|\zeta'}{8} \le |r| \le 2M$ implies that there exists $t \in [0, T_{\mathrm{thres}}]$ satisfying*

$$(F(\mu_0) - F(\mu_t)) \vee (F(\tilde{\mu}_0) - F(\tilde{\mu}_t)) \ge 2F_{\mathrm{thres}}.$$

*Proof.* We prove this lemma by contradiction. We assume that for any $t \in [0, T_{\mathrm{thres}}]$, it holds that

$$(F(\mu_0) - F(\mu_t)) \vee (F(\tilde{\mu}_0) - F(\tilde{\mu}_t)) < 2F_{\mathrm{thres}}.$$

We denote the characteristics of the vector fields $-\nabla_\mu F(\mu_t)$ and $-\nabla_\mu F(\tilde{\mu}_t)$ as $X_t, \tilde{X}_t$ respectively. That is, $\mu_t = X_t\#\mu_0, \tilde{\mu}_t = \tilde{X}_t\#\tilde{\mu}_0$, and

$$\frac{\mathrm{d}}{\mathrm{d}t}X_t = -\nabla_\mu F(\mu_t) \circ X_t, \quad \frac{\mathrm{d}}{\mathrm{d}t}\tilde{X}_t = -\nabla_\mu F(\tilde{\mu}_t) \circ \tilde{X}_t.$$

It is worth noting here that

$$\|X_t - \mathrm{Id}\|_{L^2(\mu_0)} = \left\| \int_0^t (-\nabla_\mu F(\mu_\tau) \circ X_\tau) \mathrm{d}\tau \right\|_{L^2(\mu_0)}$$

$$\leq \int_0^t \|\nabla_\mu F(\mu_\tau)\|_{L^2(\mu_\tau)} \mathrm{d}\tau$$

$$\leq t^{\frac{1}{2}} \left( \int_0^t \|\nabla_\mu F(\mu_\tau)\|_{L^2(\mu_\tau)}^2 \mathrm{d}\tau \right)^{\frac{1}{2}}$$

$$\leq (T_{\mathrm{thres}}(F(\mu_0) - F(\mu_t)))^{\frac{1}{2}},$$

where the first line follows from the fact that $X_t$ is the characteristic of $-\nabla_\mu F(\mu_t)$, the second from the properties of the Bochner integral, the third from Jensen's inequality, and the fourth from Proposition 5.3. Similarly, $\|\tilde{X}_t - \mathrm{Id}\|_{L^2(\tilde{\mu}_0)} \leq (T_{\mathrm{thres}} F(\tilde{\mu}_0) - F(\tilde{\mu}_t))$ holds. Furthurmore, we set $Y_t := X_t \circ (\mathrm{Id} + \eta_p \xi)$ and $\tilde{Y}_t = \tilde{X}_t \circ (\mathrm{Id} + \eta_p \xi)$, which satisfy

$$\|Y_t - \mathrm{Id}\|_{L^2(\mu^\dagger)} \leq \|X_t - (\mathrm{Id} + \eta_p \xi)\|_{L^2(\mu_0)} + \eta_p \|\xi\|_{L^2(\mu^\dagger)}$$

$$\leq (T_{\mathrm{thres}}(F(\mu_0) - F(\mu_t)))^{\frac{1}{2}} + \eta_p \|\xi\|_{L^2(\mu^\dagger)}$$

$$\leq \sqrt{2} T_{\mathrm{thres}}^{\frac{1}{2}} F_{\mathrm{thres}}^{\frac{1}{2}} + \eta_p M, \tag{31}$$

and

$$\|\tilde{Y}_t - \mathrm{Id}\|_{L^2(\mu^\dagger)} \leq (T_{\mathrm{thres}}(F(\tilde{\mu}_0) - F(\tilde{\mu}_t)))^{\frac{1}{2}} + \eta_p \left\| \tilde{\xi} \right\|_{L^2(\mu^\dagger)}$$

$$\leq (T_{\mathrm{thres}}(F(\tilde{\mu}_0) - F(\tilde{\mu}_t)))^{\frac{1}{2}} + \eta_p \|\xi\|_{L^2(\mu^\dagger)} + \eta_p r$$

$$\leq \sqrt{2} T_{\mathrm{thres}}^{\frac{1}{2}} F_{\mathrm{thres}}^{\frac{1}{2}} + \eta_p M + \eta_p r. \tag{32}$$

We analyze the vector $w_t := \tilde{Y}_t - Y_t$. The goal is to obtain a contradiction by confirming that $\|w_t\|_{L^2(\mu^\dagger)}$ becomes large. To achieve this, we investigate how $w_t$ evolves with respect to time:

$$\frac{\mathrm{d}}{\mathrm{d}t} w_t = \frac{\mathrm{d}}{\mathrm{d}t} \tilde{Y}_t - \frac{\mathrm{d}}{\mathrm{d}t} Y_t$$

$$= \frac{\mathrm{d}}{\mathrm{d}t} \tilde{X}_t \circ (\mathrm{Id} + \eta_p \tilde{\xi}) - \frac{\mathrm{d}}{\mathrm{d}t} X_t \circ (\mathrm{Id} + \eta_p \xi)$$

$$= -\nabla_\mu F(\tilde{\mu}_t) \circ \tilde{X}_t \circ (\mathrm{Id} + \eta_p \tilde{\xi}) + \nabla_\mu F(\mu_t) \circ X_t \circ (\mathrm{Id} + \eta_p \xi)$$

$$= -\int_0^1 \frac{\mathrm{d}}{\mathrm{d}h} (\nabla_\mu F(\nu_h) \circ Y_h) \mathrm{d}h, \tag{33}$$

where $\nu_h$ is a curve connecting $\mu_t$ and $\tilde{\mu}_t$; $\nu_h := ((1-h)Y_t + h\tilde{Y}_t)\sharp\mu^\dagger$ ($h \in [0,1]$), $\nu_0 = \mu_t$ and $\nu_1 = \tilde{\mu}_t$ hold. Moreover, its direction vector is always $w_t = \tilde{Y}_t - Y_t$, so the integrand of (33) is obtained by operating $w_t$. Specifically,

$$\frac{\mathrm{d}}{\mathrm{d}h} (\nabla_\mu F(\nu_h) \circ Y_h)(x)$$

$$= \int \nabla_\mu^2 F(\nu_h, Y_h(x), Y_h(y)) w_t(y) \mu^\dagger(\mathrm{d}y) + \nabla \nabla_\mu F(\nu_h, Y_h(x)) w_t(x)$$

holds. Let

$$\frac{\mathrm{d}}{\mathrm{d}t} w_t = -H_{\mu^\dagger} w_t - \left( \int_0^1 \Delta_{t,h} \mathrm{d}h \right) w_t = -H_{\mu^\dagger} w_t - \Delta_t w_t, \tag{34}$$

where $\Delta_{t,h}$ is an operator on $L^2(\mu^\dagger)^d$, which satisfies

$$
\begin{aligned}
\Delta_{t,h}&f(x)\\
&= \int \left(\nabla_\mu^2 F(\nu_h, Y_h(x), Y_h(y)) - \nabla_\mu^2 F(\mu^\dagger, x, y)\right) f(y)\mu^\dagger(\mathrm{d}y)\\
&\quad + \nabla\nabla_\mu F(\nu_h, Y_h(x)) f(x)\\
&= \int \left(\nabla_\mu^2 F(\nu_h, Y_h(x), Y_h(y)) - \nabla_\mu^2 F(\mu^\dagger, x, y)\right) f(y)\mu^\dagger(\mathrm{d}y)\\
&\quad + \left(\nabla\nabla_\mu F(\nu_h, Y_h(x)) - \nabla\nabla_\mu F(\mu^\dagger, x)\right) f(x)\\
&\quad + \nabla\nabla_\mu F(\mu^\dagger, x) f(x). \quad (35)
\end{aligned}
$$

Moreover, the operator $\Delta_t$ is defined as $\Delta_t = \int_0^1 \Delta_{t,h}\mathrm{d}h$, the norm of which is bounded as $\tilde{O}\left(T_{\mathrm{thres}}^{\frac{1}{2}} F_{\mathrm{thres}}^{\frac{1}{2}} + \eta_p + \varepsilon\right)$ from (31), (32), and (35). It should be noted that from (34), $w_t$ evolves according to $-H_{\mu^\dagger}$ unless $\|\Delta_t\|_{L^2(\mu^\dagger)}$ is small. Since

$$
\frac{\mathrm{d}}{\mathrm{d}t}\left(e^{tH_{\mu^\dagger}} w_t\right) = e^{tH_{\mu^\dagger}}\left(\frac{\mathrm{d}}{\mathrm{d}t}w_t + H_{\mu^\dagger} w_t\right) = -e^{tH_{\mu^\dagger}}\Delta_t w_t
$$

holds from (34), by integrating we have

$$
e^{tH_{\mu^\dagger}} w_t - w_0 = -\int_0^t e^{\tau H_{\mu^\dagger}}\Delta_\tau w_\tau \mathrm{d}\tau,
$$

and so

$$
w_t = e^{-tH_{\mu^\dagger}} w_0 - \int_0^t e^{(\tau-t)H_{\mu^\dagger}}\Delta_\tau w_\tau \mathrm{d}\tau = e^{-\lambda_0 t}\eta_p r\psi_0 - \int_0^t e^{(\tau-t)H_{\mu^\dagger}}\Delta_\tau w_\tau \mathrm{d}\tau,
$$

where we used the equation $w_0 = \tilde{Y}_0 - Y_0 = \eta_p(\tilde{\xi} - \xi) = \eta_p r\psi_0$. Then, it holds that

$$
\begin{aligned}
\left|\|w_t\|_{L^2(\mu^\dagger)} - e^{-\lambda_0 t}\eta_p r\right| &= \left|\|w_t\|_{L^2(\mu^\dagger)} - \left\|e^{-\lambda_0 t}\eta_p r\psi_0\right\|_{L^2(\mu^\dagger)}\right|\\
&\leq \left\|w_t - e^{-\lambda_0 t}\eta_p r\psi_0\right\|_{L^2(\mu^\dagger)}\\
&\leq \int_0^t \left\|e^{(\tau-t)H_{\mu^\dagger}}\right\|_{L^2(\mu^\dagger)}\|\Delta_\tau\|_{L^2(\mu^\dagger)}\|w_\tau\|_{L^2(\mu^\dagger)}\mathrm{d}\tau\\
&\leq \int_0^t e^{\lambda_0(\tau-t)}\|\Delta_\tau\|_{L^2(\mu^\dagger)}\|w_\tau\|_{L^2(\mu^\dagger)}\mathrm{d}\tau. \quad (36)
\end{aligned}
$$

Here we use $\|e^{(\tau-t)H_{\mu^\dagger}}\|_{L^2(\mu^\dagger)} = e^{\lambda_0(\tau-t)}$, which is implied by $\lambda_0 = \lambda_{\min}(H_{\mu^\dagger})$ and $\tau - t \leq 0$. From (35) and the inequality $|H'_{\mu^\dagger}\|_{L^2(\mu^\dagger)} \leq R_2\|\nabla_\mu F(\mu^\dagger)\|_{L^2(\mu^\dagger)} \leq R_2\varepsilon$, it follows from (31) and (32) that

$$
\begin{aligned}
\|\Delta_{t,h}\|_{L^2(\mu^\dagger)} &\leq (L_2 + L_3)\|Y_h - \mathrm{Id}\|_{L^2(\mu^\dagger)} + R_2\varepsilon\\
&= (1-h)(L_2 + L_3)\|Y_t - \mathrm{Id}\|_{L^2(\mu^\dagger)}\\
&\quad + h(L_2 + L_3)\left\|\tilde{Y}_t - \mathrm{Id}\right\|_{L^2(\mu^\dagger)} + R_2\varepsilon\\
&\leq (L_2 + L_3)\left(\sqrt{2}T_{\mathrm{thres}}^{\frac{1}{2}}F_{\mathrm{thres}}^{\frac{1}{2}} + \eta_p M + h\eta_p r\right) + R_2\varepsilon.
\end{aligned}
$$

Therefore, it holds that

$$
\begin{aligned}
\|\Delta_t\|_{L^2(\mu^\dagger)} &\leq \int_0^1 \|\Delta_{t,h}\|_{L^2(\mu^\dagger)}\mathrm{d}h\\
&\leq (L_2 + L_3)\left(\sqrt{2}T_{\mathrm{thres}}^{\frac{1}{2}}F_{\mathrm{thres}}^{\frac{1}{2}} + \eta_p M + \frac{1}{2}\eta_p r\right) + R_2\varepsilon.\\
&\leq (L_2 + L_3)\left(\sqrt{2}T_{\mathrm{thres}}^{\frac{1}{2}}F_{\mathrm{thres}}^{\frac{1}{2}} + 2\eta_p M\right) + R_2\varepsilon\\
&=: \Delta.
\end{aligned}
$$

By manipulating the inequality (36), the following is obtained:

$$\left| e^{\lambda_0 t} \|w_t\|_{L^2(\mu^\dagger)} - \eta_p r \right| \le \Delta \int_0^t e^{\lambda_0 \tau} \|w_\tau\|_{L^2(\mu^\dagger)} d\tau.$$

The application of Gronwall's inequality to the upper bound yields $e^{\lambda_0 t}\|w_t\|_{L^2(\mu^\dagger)} \le \eta_p r e^{\Delta t}$. Then it holds that

$$\begin{aligned}
\|w_t\|_{L^2(\mu^\dagger)} &\ge \eta_p r e^{-\lambda_0 t} - \Delta e^{-\lambda_0 t} \int_0^t e^{\lambda_0 \tau} \|w_t\|_{L^2(\mu^\dagger)} \, d\tau \\
&\ge \eta_p r e^{-\lambda_0 t} \left( 1 - \Delta \int_0^t e^{\Delta \tau} \, d\tau \right) \\
&= \eta_p r e^{-\lambda_0 t}(2 - e^{\Delta t}).
\end{aligned} \tag{37}$$

The left side of (37) is upper-bounded as

$$\begin{aligned}
\|w_t\|_{L^2(\mu^\dagger)} &\le \left\| \tilde{Y}_t - \mathrm{Id} \right\|_{L^2(\mu^\dagger} + \|Y_t - \mathrm{Id}\|_{L^2(\mu^\dagger} \\
&< 2\sqrt{2} T_{\mathrm{thres}}^{\frac{1}{2}} F_{\mathrm{thres}}^{\frac{1}{2}} + 2\eta_p M + \eta_p r \\
&\le 4\sqrt{2} T_{\mathrm{thres}}^{\frac{1}{2}} F_{\mathrm{thres}}^{\frac{1}{2}},
\end{aligned}$$

where in the last line it follows from $T_{\mathrm{thres}}^{\frac{1}{2}} F_{\mathrm{thres}}^{\frac{1}{2}} = \tilde{O}(\delta)$, $\eta_p M = o(\delta)$, $\eta_p = o(\delta)$.
On the other hand, using $\eta_p M T_{\mathrm{thres}} = o(1)$, $\varepsilon T_{\mathrm{thres}} = o(1)$ and the definition of $F_{\mathrm{thres}}$, we have

$$\begin{aligned}
t\Delta &\le T_{\mathrm{thres}} \Delta \\
&= ((L_2 + L_3)(\sqrt{2} T_{\mathrm{thres}}^{\frac{1}{2}} F_{\mathrm{thres}}^{\frac{1}{2}} + 2\eta_p M) + R_2 \varepsilon) T_{\mathrm{thres}} \\
&= \sqrt{2}(L_2 + L_3) T_{\mathrm{thres}}^{\frac{3}{2}} F_{\mathrm{thres}}^{\frac{1}{2}} + 2(L_2 + L_3)\eta_p M T_{\mathrm{thres}} + R_2 \varepsilon T_{\mathrm{thres}} \\
&= \frac{1}{3} \log \frac{3}{2} + \frac{1}{3} \log \frac{3}{2} + \frac{1}{3} \log \frac{3}{2} \\
&= \log \frac{3}{2}.
\end{aligned}$$

Then the right side of (37) is lower-bounded as

$$\begin{aligned}
\eta_p r e^{-\lambda_0 t}(2 - e^{\Delta t}) &\ge \frac{\eta_p r}{2} e^{\delta t} \\
&\ge \frac{\eta_p r}{2} (e\delta t)^{\frac{1}{2}} e^{\frac{\delta}{2} t},
\end{aligned}$$

where we used in the second line the inequality $e^{\frac{x}{2}} \le \sqrt{e} x^{\frac{1}{2}}$ as $x = \delta t$. Letting $t = T_{\mathrm{thres}}$ and transforming (37) yields

$$\begin{aligned}
e^{\frac{\delta}{2} T_{\mathrm{thres}}} &< \frac{8\sqrt{2}}{\sqrt{e}} \frac{F_{\mathrm{thres}}^{\frac{1}{2}}}{\delta^{\frac{1}{2}} \eta_p r} \\
&\le \frac{16 L_1^{\frac{1}{2}}}{\sqrt{e}} \frac{M}{\delta^{\frac{1}{2}} r}.
\end{aligned}$$

where the second line is implied by

$$\begin{aligned}
\eta_p M &= \frac{2 F_{\mathrm{thres}}}{\varepsilon + \sqrt{\varepsilon^2 + 2 L_1 F_{\mathrm{thres}}}} \\
&\ge \frac{2 F_{\mathrm{thres}}}{2\varepsilon + \sqrt{2 L_1 F_{\mathrm{thres}}}} \\
&\ge \frac{1}{2} \sqrt{\frac{2 F_{\mathrm{thres}}}{L_1}}.
\end{aligned}$$

Here it follows from the definition of $\eta$ and the assumption $\delta^2 \ge (L_2 + L_3)\varepsilon$. Since we set $T_{\mathrm{thres}} = \frac{2}{\delta} \log \left( \frac{16 L_1^{\frac{1}{2}} M}{\sqrt{e} \delta^{\frac{1}{2}} r} \right)$, this leads to a contradiction. $\qquad\square$

*Proof of Proposition 5.4.* From the discussion provided after Lemma D.8, by setting $M = \left(\frac{eC}{e-1}\left(1 + 2\log\frac{2}{\zeta'}\right)\right)^{\frac{1}{2}} = \tilde{O}(1)$, it holds that $\|\xi\|_{L^2(\mu)} \le M = \tilde{O}(1)$ with probability $1 - \frac{\zeta'}{2}$. At this point, by choosing the hyperparameters as in (30), we have $\eta_p = \tilde{O}\left(\delta^{\frac{3}{2}} \wedge \frac{\delta^3}{\varepsilon}\right)$, $T_{\text{thres}} = \tilde{O}\left(\frac{1}{\delta}\right)$, $F_{\text{thres}} = \tilde{O}(\delta^3)$, and Lemma E.1, Lemma E.2 can be applied. There exists $t \in [0, T_{\text{thres}}]$ such that

$$
\begin{aligned}
\mathrm{P}\left(F(\mu_0) - F(\mu_t) \le 2F_{\text{thres}}\right) &\ge \mathrm{P}\left(F(\mu_0) - F(\mu_t) \le 2F_{\text{thres}}, \left|\langle\psi_0, \xi\rangle_{L^2(\mu^\dagger)}\right| \le M\right) \\
&\ge \int_{\mathbb{R}\setminus\left(-\frac{\sqrt{2\pi}|\lambda_0|\zeta'}{4}, \frac{\sqrt{2\pi}|\lambda_0|\zeta'}{4}\right)} \frac{1}{\sqrt{2\pi}|\lambda_0|} \exp\left(-\frac{x^2}{2\lambda_0^2}\right) dx - \mathrm{P}\left(\left|\langle\psi_0, \xi\rangle_{L^2(\mu^\dagger)}\right| \ge M\right) \\
&\ge 1 - \frac{1}{\sqrt{2\pi}|\lambda_0|} \cdot 2 \cdot \frac{\sqrt{2\pi}|\lambda_0|\zeta'}{8} - \sqrt{2}e^{-\frac{M^2}{4R_1^2}} \\
&\ge 1 - \frac{\zeta'}{4} - \frac{\zeta'}{4} \\
&= 1 - \frac{\zeta'}{2},
\end{aligned}
$$

where several previously established results are used. We provide a detailed explanation below:

- The first line is straightforward.

- The second line is followed by Lemma E.2 as shown below. The key point of Lemma E.2 is that when considering two points respectively perturbed but the $\psi_0$-direction of perturbation differs by a fixed amount, the WGF dynamics lead to at least one point having an decrease of the objective greater than $2F_{\text{thres}}$.

  Let $\Omega$ be the sample space associated with the randomness of the Gaussian process $\xi$, and define a random variable $X$ as $X = \langle\psi_0, \xi\rangle_{L^2(\mu^\dagger)}$.

  Take $\omega_1, \omega_2 \in \Omega$ such that $|X(\omega_1)| \vee |X(\omega_2)| \le M, \xi(\omega_2) = \xi(\omega_1) + r\psi_0$, where $r$ is a real constant satisfying the assumption in Lemma E.2. We consider applying Lemma E.2 in this setting.

  Since $|X(\omega_1)| \le M, |X(\omega_2)| \le M$ implies $|r| = |X(\omega_1) - X(\omega_2)| \le 2M$, from Lemma E.2, we obtain that if $|r| \ge |r_0|$, $r_0 = \frac{\sqrt{2\pi}|\lambda_0|\zeta'}{8}$, then at least one of the samples $\omega_1, \omega_2$ satisfies $F(\mu_0) - F(\mu_t) \ge 2F_{\text{thres}}$ . $---(\star)$

  Based on this, consider the following two cases:

  - There exists a point $x_0 \in [-M, M]$ such that $X(\omega) = x_0 \implies F(\mu_0) - F(\mu_t) < 2F_{\text{thres}}$.
  - For all points $x_0 \in [-M, M], X(\omega) = x_0 \implies F(\mu_0) - F(\mu_t) \ge 2F_{\text{thres}}$.

  In the former case, by $(\star)$, $X \in [-M, M] \setminus (x_0 - r_0, x_0 + r_0) \implies F(\mu_0) - F(\mu_t) \ge 2F_{\text{thres}}$ holds. Therefore, in either case, the following holds:

  $$
  \begin{aligned}
  &\mathrm{P}\left(F(\mu_0) - F(\mu_t) \le 2F_{\text{thres}}, \left|\langle\psi_0, \xi\rangle_{L^2(\mu^\dagger)}\right| \le M\right) \\
  &\ge \mathrm{P}\left(X \in [-M, M] \setminus (x_0 - r_0, x_0 + r_0)\right) \\
  &= \int_{[-M,M]\setminus(x_0-r_0, x_0+r_0)} \frac{1}{\sqrt{2\pi}\lambda_0} e^{-\frac{x^2}{2\lambda_0}} dx \\
  &\ge \int_{\mathbb{R}\setminus(x_0-r_0, x_0+r_0)} \frac{1}{\sqrt{2\pi}\lambda_0} e^{-\frac{x^2}{2\lambda_0}} dx - \int_{\mathbb{R}\setminus(-M, M)} \frac{1}{\sqrt{2\pi}\lambda_0} e^{-\frac{x^2}{2\lambda_0}} dx
  \end{aligned}
  $$

  Here, the first term is minimized at $x_0 = 0$, which justifies the bound used in the second line above.

- The third line : The lower bound of the first term follows from the fact that the Gaussian PDF reaches its maximum $\frac{1}{\sqrt{2\pi}|\lambda_0|}$ at $x = 0$.

The lower bound of the second term comes from the Gaussian tail bound. (This follows from Proposition D.6 by taking $X = \langle \psi_0, \xi \rangle_{L^2(\mu^\dagger)}$, $\varphi(x) = \exp\left(\frac{x^2}{4\lambda_0^2}\right)$.)

- The fourth line follows from the definition of $M$.

Thus, with probability $1 - \frac{\zeta'}{2}$, we have

$$F(\mu_0) - F(\mu_t) \geq 2F_{\text{thres}}.$$

Finally, combining with Lemma E.1, the following holds:

$$
\begin{aligned}
F(\mu^\dagger) - F(\mu_{T_{\text{thres}}}) &= F(\mu\dagger) - F(\mu_0) + F(\mu_0) - F(\mu_{T_{\text{thres}}}) \\
&\geq -F_{\text{thres}} + 2F_{\text{thres}} \\
&= F_{\text{thres}}.
\end{aligned}
$$

This occurs with probability more than $1 - \left(\frac{\zeta'}{2} + \frac{\zeta'}{2}\right) = 1 - \zeta'$. $\qquad\square$

## E.2 Discrete-time Convergence Analysis

In this section, we prove the convergence of discrete-time PWGF (Theorem 5.5). For the discretization, the following Lipschitz continuity of the Wasserstein gradient is utilized to bridge the gap with continuous time.

**Lemma E.3.** *Let $F : \mathcal{P}_2(\mathbb{R}^d) \to \mathbb{R}$ be sufficiently smooth and suppose that the Wasserstein gradient $\nabla_\mu F$ be $L_1$-Lipschitz continuous.[6] Then, it holds that for any $\mu, \nu \in \mathcal{P}_2(\mathbb{R}^d)$, $\gamma \in \Gamma(\mu, \nu)$,*

$$F(\nu) - F(\mu) \leq \int \langle \nabla_\mu F(\mu, x), y - x \rangle \gamma(\mathrm{d}x\mathrm{d}y) + \frac{L_1}{2} \int \|x - y\|^2 \gamma(\mathrm{d}x\mathrm{d}y).$$

*Proof.* Let $\mu_t = ((1-t)p_1 + tp_2)\#\gamma \in \mathcal{P}_2(\mathbb{R}^d)$, $\gamma_t = (\mathrm{Id} \times (1-t)p_1 + tp_2))\#\gamma \in \Gamma(\mu, \mu_t)$. The application of Lemma C.6 implies

$$
\begin{aligned}
\frac{\mathrm{d}}{\mathrm{d}t}F(\mu_t) &= \int \langle \nabla_\mu F(\mu_t, (1-t)x + ty), y - x \rangle \gamma(\mathrm{d}x\mathrm{d}y) \\
&= \int \langle \nabla_\mu F(\mu, x), y - x \rangle \gamma(\mathrm{d}x\mathrm{d}y) \\
&\quad + \int \langle \nabla_\mu F(\mu_t, (1-t)x + ty) - \nabla_\mu F(\mu, x), y - x \rangle \gamma(\mathrm{d}x\mathrm{d}y) \\
&\leq \int \langle \nabla_\mu F(\mu, x), y - x \rangle \gamma(\mathrm{d}x\mathrm{d}y) \\
&\quad + L_1\left(\int \|x - y\|^2 \gamma_t(\mathrm{d}x\mathrm{d}y)\right)^{\frac{1}{2}}\left(\int \|x - y\|^2 \gamma(\mathrm{d}x\mathrm{d}y)\right)^{\frac{1}{2}} \\
&= \int \langle \nabla_\mu F(\mu, x), y - x \rangle \gamma(\mathrm{d}x\mathrm{d}y) + L_1 t \int \|x - y\|^2 \gamma(\mathrm{d}x\mathrm{d}y).
\end{aligned}
$$

In the third line, we use the Cauchy-Schwarz inequality and Lipschitz continuity. Then, we have

$$
\begin{aligned}
F(\nu) - F(\mu) &= \int_0^1 \frac{\mathrm{d}}{\mathrm{d}t}F(\mu_t)\mathrm{d}t \\
&= \int \langle \nabla_\mu F(\mu, x), y - x \rangle \gamma(\mathrm{d}x\mathrm{d}y) + \frac{L_1}{2} \int \|x - y\|^2 \gamma(\mathrm{d}x\mathrm{d}y).
\end{aligned}
$$

$\qquad\square$

The following proposition is a discrete version of Gronwall's inequality, which will be used in subsequent proofs.

---

[6]See Assumption 3.

**Proposition E.4.** *Let $\{a_k\}_{k \geq 0}$ be real-valued sequence and $b \neq 0$, $c \in \mathbb{R}$. Then,*

$$a_k - c \leq b \sum_{l=0}^{k-1} a_l \quad (\forall k \geq 0) \quad \Longrightarrow \quad a_k - a_0 \leq c(b+1)^k.$$

*Proof.* Letting $d_k := \sum_{l=0}^{k} a_k$, we have

$$d_k + \frac{c}{b} \leq a_k + d_{k-1} + \frac{c}{b} \leq (b+1)d_{k-1} + c + \frac{c}{b}$$
$$= (b+1)(d_{k-1} + \frac{c}{b}) \leq (b+1)^{k+1}\frac{c}{b}.$$

Then it holds that

$$a_k \leq c + bd_{k-1} \leq c + b\frac{c}{b}\big((b+1)^k - 1\big) = c(b+1)^k.$$

$\square$

The following proposition corresponds to Lemma 5.3 in continuous time and serves to prove two key points: the evaluation of the decrease in $F$ when the gradient is large, and the ability to reduce the objective function near saddle points.

**Proposition E.5.** *Let $\{\mu^{(l)}\}_{l=0}^{k}$ be a sequence of probability measures generated by discrete-time PWGF (Algorithm 2) with step size $\eta \leq \frac{1}{L_1}$. Then it holds that*

$$F(\mu^{(0)}) - F(\mu^{(k)}) \geq \frac{\eta}{2} \sum_{l=0}^{k-1} \left\| \nabla_\mu F(\mu^{(l)}) \right\|_{L^2(\mu^{(l)})}^2.$$

*Proof.* By Lemma E.3, it holds that for any $l = 0, \cdots, k-1$,

$$F(\mu^{(l+1)}) - F(\mu^{(l)}) \leq \int \left\langle \nabla_\mu F(\mu^{(l)}, x), y - x \right\rangle \Big( \text{Id} \times (\text{Id} - \eta \nabla_\mu F(\mu^{(l)})) \Big) \# \mu^{(l)}(\mathrm{d}x\mathrm{d}y)$$
$$+ \frac{L_1}{2} \int \|x - y\|^2 \Big( \text{Id} \times (\text{Id} - \eta \nabla_\mu F(\mu^{(l)})) \Big) \# \mu^{(l)}(\mathrm{d}x\mathrm{d}y)$$
$$= -\eta \left( 1 - \frac{L_1 \eta}{2} \right) \left\| \nabla_\mu F(\mu^{(l)}) \right\|_{L^2(\mu^{(l)})}^2$$
$$\leq -\frac{\eta}{2} \left\| \nabla_\mu F(\mu^{(l)}) \right\|_{L^2(\mu^{(l)})}^2.$$

Then we have

$$F(\mu^{(k)}) - F(\mu^{(0)}) = \sum_{l=0}^{k-1} \Big( F(\mu^{(l+1)}) - F(\mu^{(l)}) \Big)$$
$$\leq -\frac{\eta}{2} \sum_{l=0}^{k-1} \left\| \nabla_\mu F(\mu^{(l)}) \right\|_{L^2(\mu^{(l)})}^2.$$

$\square$

In discrete-time PWGF, we take hyperparameters $\eta_p$, $F_{\text{thres}}$, $k_{\text{thres}}$, $\eta$ as follows:

$$M = \left(\frac{eR_1{}^2}{e-1}\left(1 + 2\log\frac{2}{\zeta'}\right)\right)^{\frac{1}{2}} \vee 2R_1\left(\log\frac{4\sqrt{2}}{\zeta'}\right)^{\frac{1}{2}} = \tilde{O}(1),$$

$$\eta \le \frac{1}{L_1} = O(1),$$

$$k_{\text{thres}} = \frac{2}{\log(1 + \eta\delta)}\log\left(\frac{16\sqrt{2}L_1^{\frac{1}{2}}\eta^{\frac{1}{2}}M}{\sqrt{er}\log^{\frac{1}{2}}(1 + \eta\delta)}\right) = \tilde{O}\left(\frac{1}{\delta}\right), \tag{38}$$

$$F_{\text{thres}} = \frac{\eta^{-3}k_{\text{thres}}^{-3}}{18(L_2 + L_3)^2}\log^2\frac{3}{2} = \tilde{O}(\delta^3),$$

$$\eta_p = \frac{2F_{\text{thres}}}{M(\varepsilon + \sqrt{\varepsilon^2 + 2L_1 F_{\text{thres}}})} = \tilde{O}\left(\frac{\delta^3}{\varepsilon} \wedge \delta^{\frac{3}{2}}\right).$$

Since $\eta_p$ and $M$ are set in the same way as continuous time (30), Lemma E.1 holds in discrete time as well. It should also be noted that $F_{\text{thres}}$ is defined to correspond to $T_{\text{thres}}$ and $\eta k_{\text{thres}}$.

Next, we present the discrete-time counterpart of Lemma E.2.

**Proposition E.6.** *Let $\mu^\dagger$ be an $(\varepsilon, \delta)$-saddle point, $\mu^{(0)} := (\text{Id} + \eta_p\xi)\#\mu^\dagger$, and the hyperparameters $\eta_p, F_{\text{thres}}$ be taken as in (38). Furthermore, we set $\tilde{\mu}^{(0)} = (\text{Id} + \eta_p\tilde{\xi})\sharp\mu^\dagger$, where $\tilde{\xi} = \xi + r\psi_0$, $r \in \mathbb{R}$ is a constant, and $\psi_0$ is the eigenvector of the Hessian operator $H_\mu$ corresponding to the smallest eigenvalue $\lambda_0$.*
*Letting $\mu^{(k)}t$, $\tilde{\mu}^{(k)}$ be WGF initialized at $\mu^{(0)}, \tilde{\mu}^{(0)}$ respectively, then $\|\xi\|_{L^2(\mu^\dagger)} \le M$ and $\frac{\sqrt{2\pi}|\lambda_0|\zeta'}{4} \le |r| \le 2M$ implies that there exists $k = 0, \cdots, k_{\text{thres}}]$ satisfying*

$$(F(\mu^{(0)}) - F(\mu^{(k)})) \vee (F(\tilde{\mu}^{(0)}) - F(\tilde{\mu}^{(k)})) \ge 2F_{\text{thres}}.$$

*Proof.* We give a proof by contradiction. Assume that for any $0 \le k \le k_{\text{thres}}$,

$$(F(\mu^{(0)}) - F(\mu^{(k)})) \vee (F(\tilde{\mu}^{(0)}) - F(\tilde{\mu}^{(k)})) \vee \Delta_F(\tilde{\mu}^{(0)}, \tilde{\mu}^{(k)}) < 2F_{\text{thres}}.$$

The following vector fields are used to evaluate how $\Delta_F(\mu^{(0)}, \mu^{(k)})$ and $\Delta_F(\tilde{\mu}^{(0)}, \tilde{\mu}^{(k)})$ evolve:

$$Y^{(k)} = (\text{Id} - \eta\nabla_\mu F(\mu^{(k-1)})) \circ \cdots \circ (\text{Id} - \eta\nabla_\mu F(\mu^{(0)})) \circ (\text{Id} + \eta_p\xi),$$

$$\tilde{Y}^{(k)} = (\text{Id} - \eta\nabla_\mu F(\tilde{\mu}^{(k-1)})) \circ \cdots \circ (\text{Id} - \eta\nabla_\mu F(\tilde{\mu}^{(0)})) \circ (\text{Id} + \eta_p\tilde{\xi}).$$

Here, we note that

$$\left\|Y^{(k)} - \text{Id}\right\|_{L^2(\mu^{(0)})} \le \eta_p\|\xi\|_{L^2(\mu^\dagger)} + \sum_{l=0}^{k-1}\left\|Y^{(l+1)} - Y^{(l)}\right\|_{L^2(\mu^{(l)})}$$

$$\le \eta_p\|\xi\|_{L^2(\mu^\dagger)} + \eta\sum_{l=0}^{k-1}\left\|\nabla_\mu F(\mu^{(l)})\right\|_{L^2(\mu^{(l)})}$$

$$= \eta_p\|\xi\|_{L^2(\mu^\dagger)} + \eta k\sum_{l=0}^{k-1}\frac{1}{k}\left\|\nabla_\mu F(\mu^{(l)})\right\|_{L^2(\mu^{(l)})}$$

$$\le \eta_p\|\xi\|_{L^2(\mu^\dagger)} + \eta k\left(\sum_{l=0}^{k-1}\frac{1}{k}\left\|\nabla_\mu F(\mu^{(l)})\right\|_{L^2(\mu^{(l)})}^2\right)^{\frac{1}{2}}$$

$$= \eta_p\|\xi\|_{L^2(\mu^\dagger)} + \sqrt{2}\eta^{\frac{1}{2}}k^{\frac{1}{2}}\left(\frac{\eta}{2}\sum_{l=0}^{k-1}\left\|\nabla_\mu F(\mu^{(l)})\right\|_{L^2(\mu^{(l)})}^2\right)^{\frac{1}{2}}$$

$$\le \eta_p\|\xi\|_{L^2(\mu^\dagger)} + \sqrt{2}\eta^{\frac{1}{2}}k^{\frac{1}{2}}(F(\mu^{(0)}) - F(\mu^{(k)})^{\frac{1}{2}}$$

$$\le \eta_p M + 2\eta^{\frac{1}{2}}k_{\text{thres}}^{\frac{1}{2}}F_{\text{thres}}^{\frac{1}{2}}. \tag{39}$$

The third line is due to Jensen's inequality and the fifth line is due to Proposition E.5. Similarly,

$$\left\|\tilde{Y}^{(k)} - \mathrm{Id}\right\|_{L^2(\mu^\dagger)} \le \eta_p M + \eta_p r + 2\eta^{\frac{1}{2}} k_{\text{thres}}^{\frac{2}{2}} F_{\text{thres}}^{\frac{1}{2}}. \tag{40}$$

We demonstrate that either $\mu^{(k)}$ or $\tilde{\mu}^{(k)}$ significantly decreases the objective function. To this end, we define the following vector field as a measure of the "distance" between $\mu^{(k)}$ and $\tilde{\mu}^{(k)}$:

$$w^{(k)} := \tilde{Y}^{(k)} - Y^{(k)}.$$

$w^{(k)}$ follows the recurrence relation given by the following:

$$w^{(k+1)} - w^{(k)} = -\eta \nabla_\mu F(\tilde{\mu}^{(k)}) \circ \tilde{Y}^{(k)} + \eta \nabla_\mu F(\mu^{(k)}) \circ Y^{(k)}$$

$$= -\eta \int_0^1 \frac{\mathrm{d}}{\mathrm{d}h} \left( \nabla_\mu F(\nu_h) \circ Y_h^{(k)} \right) \mathrm{d}h$$

$$= -\eta H_{\mu^\dagger} w^{(k)} - \eta \Delta^{(k)} w^{(k)}, \tag{41}$$

where we set $Y_h^{(k)} = (1-h)Y^{(k)} + h\tilde{Y}^{(k)}$, $\nu_h = Y_h^{(k)} \# \mu^\dagger = ((1-h)Y^{(k)} + h\tilde{Y}^{(k)}) \# \mu^\dagger$ and

$$\Delta^{(k)} = \int_0^1 \Delta_h^{(k)} \mathrm{d}h,$$

$$\Delta_h^{(k)} f(x) = \int \left( \nabla_\mu^2 F(\nu_h, Y_h(x), Y_h(y)) - \nabla_\mu^2 F(\mu^\dagger, x, y) \right) f(y) \mu^\dagger(\mathrm{d}y)$$

$$+ (\nabla \nabla_\mu F(\nu_h, Y_h(x)) - \nabla \nabla_\mu F(\mu^\dagger, x)) f(x)$$

$$+ \nabla \nabla_\mu F(\mu^\dagger, x) f(x).$$

The recurrence formula (41) yields

$$w^{(k)} = (1 - \eta H_{\mu^\dagger})^k w^{(0)} - \eta \sum_{l=0}^{k-1} (1 - \eta H_{\mu^\dagger})^{k-l-1} \Delta^{(l)} w^{(l)}$$

$$= (1 - \eta \lambda_0)^k w^{(0)} - \eta \sum_{l=0}^{k-1} (1 - \eta H_{\mu^\dagger})^{k-l-1} \Delta^{(l)} w^{(l)}.$$

Here, we use the fact that $w^{(0)} = \tilde{Y}^{(0)} - Y^{(0)} = \eta_p r \psi_0$. Then, we have

$$\left| \left\| w^{(k)} \right\|_{L^2(\mu^\dagger)} - (1 - \eta \lambda_0)^k \eta r \right| \le \left\| w^{(k)} - (1 - \eta \lambda_0)^k w^{(0)} \right\|_{L^2(\mu^\dagger)}$$

$$\le \eta \sum_{l=0}^{k-1} \left\| 1 - \eta H_{\mu^\dagger} \right\|_{L^2(\mu^\dagger)}^{k-l-1} \left\| \Delta^{(l)} \right\|_{L^2(\mu^\dagger)} \left\| w^{(l)} \right\|_{L^2(\mu^\dagger)}$$

$$\le \eta \Delta \sum_{l=0}^{k-1} (1 - \eta \lambda_0)^{k-l-1} \left\| w^{(l)} \right\|_{L^2(\mu^\dagger)},$$

$$\therefore \left| (1 - \eta \lambda_0)^{-k} \left\| w^{(k)} \right\|_{L^2(\mu^\dagger)} - \eta_p r \right| \le \eta \Delta \sum_{l=0}^{k-1} (1 - \eta \lambda_0)^{-l-1} \left\| w^{(l)} \right\|_{L^2(\mu^\dagger)},$$

where the constant $\Delta$ upper bounds the norm of $\Delta^{(k)}$ and set in the same manner as in continuous time:

$$\Delta := (L_2 + L_3)(2\eta^{\frac{1}{2}} k_{\text{thres}}^{\frac{1}{2}} F_{\text{thres}}^{\frac{1}{2}} + 2\eta_p M) + R_2 \varepsilon \ge \left\| \Delta^{(k)} \right\|_{L^2(\mu^\dagger)}.$$

Using the discrete version of Gronwall's inequality (Proposition E.4) with $a_k = (1 - \eta \lambda_0)^{-k} \left\| w^{(k)} \right\|_{L^2(\mu^\dagger)}$, $b = \frac{\eta \Delta}{1 - \eta \lambda_0}$, $c = \eta_p r$, we obtain the following:

$$(1 - \eta \lambda_0)^{-l} \left\| w^{(l)} \right\|_{L^2(\mu^\dagger)} \le \eta_p r \left( 1 + \frac{\eta \Delta}{1 - \eta \lambda_0} \right)^l.$$

From this, we have

$$(1-\eta\lambda_0)^{-k}\left\|w^{(k)}\right\| \geq \eta_p r - \eta\Delta \sum_{l=0}^{k-1}(1-\eta\lambda_0)^{-l-1}\left\|w^{(l)}\right\|_{L^2(\mu^\dagger)}$$

$$\geq \eta_p r\left(1 - \frac{\eta\Delta}{1-\eta\lambda_0}\sum_{l=0}^{k-1}\left(1+\frac{\eta\Delta}{1-\eta\lambda_0}\right)^l\right)$$

$$= \eta_p r\left(1 - \frac{\eta\Delta}{1-\eta\lambda_0}\frac{\left(1+\frac{\eta\Delta}{1-\eta\lambda_0}\right)^k - 1}{\frac{\eta\Delta}{1-\eta\lambda_0}}\right)$$

$$= \eta_p r\left(2 - \left(1+\frac{\eta\Delta}{1-\eta\lambda_0}\right)^k\right)$$

$$\geq \eta_p r\left(2 - (1+\eta\Delta)^k\right)$$

$$\geq \eta_p r(2 - \exp(\eta k_{\text{thres}}\Delta))$$

$$\geq \frac{\eta_p r}{2},$$

where the last line holds as $\eta k_{\text{thres}}\Delta \leq \log\frac{3}{2}$ in the same manner as in continuous time,

$$\eta k_{\text{thres}}\Delta \leq \eta k_{\text{thres}}\left((L_2+L_3)(2\eta^{\frac{1}{2}}k_{\text{thres}}^{\frac{1}{2}}F_{\text{thres}}^{\frac{1}{2}}+2\eta_p M)+R_2\varepsilon\right)$$

$$\leq 2(L_2+L_3)(\eta k_{\text{thres}})^{\frac{3}{2}}F_{\text{thres}}^{\frac{1}{2}}+2(L_2+L_3)\eta_p M\eta k_{\text{thres}}+R_2\eta k_{\text{thres}}\varepsilon$$

$$\leq \frac{1}{3}\log\frac{3}{2}\cdot 3 = \log\frac{3}{2}.$$

Then we have

$$\left\|w^{(k)}\right\| \geq \frac{\eta_p r}{2}(1-\eta\lambda_0)^k$$

$$\geq \frac{\eta_p r}{2}(1+\eta\delta)^k$$

$$\geq \frac{\eta_p r}{2}\sqrt{e}k^{\frac{1}{2}}(1+\eta\delta)^{\frac{k}{2}}\log^{\frac{1}{2}}(1+\eta\delta). \tag{42}$$

On the other hand,

$$\left\|w^{(k)}\right\| \leq \left\|\tilde{Y}^{(k)}-\text{Id}\right\|_{L^2(\mu^\dagger)}+\left\|Y^{(k)}-\text{Id}\right\|_{L^2(\mu^\dagger)}$$

$$< 4\eta^{\frac{1}{2}}k_{\text{thres}}^{\frac{1}{2}}F_{\text{thres}}^{\frac{1}{2}}+2\eta_p M+\eta_p r$$

$$\leq 8\eta^{\frac{1}{2}}k_{\text{thres}}^{\frac{1}{2}}F_{\text{thres}}^{\frac{1}{2}}, \tag{43}$$

where in the second line (39) and (40) are used, and in the third line $\eta^{\frac{1}{2}}k_{\text{thres}}^{\frac{1}{2}}F_{\text{thres}}^{\frac{1}{2}} = \tilde{O}(\delta)$, $\eta_p M = o(\delta)$, $\eta_p r = o(\delta)$. Letting $k = k_{\text{thres}}$, it follows from (42), (43) and $\eta_p M \geq \frac{1}{2}\sqrt{\frac{2F_{\text{thres}}}{L_1}}$ that

$$(1+\eta\delta)^{\frac{k_{\text{thres}}}{2}} < \frac{16\eta^{\frac{1}{2}}F_{\text{thres}}^{\frac{1}{2}}}{\sqrt{e}\eta_p r\log^{\frac{1}{2}}(1+\eta\delta)}$$

$$\leq \frac{16\sqrt{2}L_1^{\frac{1}{2}}\eta^{\frac{1}{2}}M}{\sqrt{e}r\log^{\frac{1}{2}}(1+\eta\delta)}.$$

This leads to a contradiction, as we had set $k_{\text{thres}}$ as in (38). $\qquad\square$

The following proposition corresponds to the discrete-time version of Proposition 5.4.

**Proposition E.7.** *Let $\varepsilon$, $\delta$, $\zeta' > 0$ be chosen such that $(L_2 + L_3)\,\varepsilon \leq \delta^2$. Suppose $\mu^{\dagger} \in \mathcal{P}_2^a(\mathbb{R}^d)$ satisfies $\left\|\nabla_{\mu}F(\mu^{\dagger})\right\|_{L^2(\mu^{\dagger})} < \varepsilon$ and $\lambda_0 := \lambda_{\min}H_{\mu^{\dagger}} \leq -\delta$. x Considering $\xi \sim \mathrm{GP}(0, k_{\mu})$ and setting $\mu_0 = (\mathrm{Id} + \eta_p \xi)\sharp \mu^{\dagger}$ as the initial value of the discrete time PWGF flow $\mu^{(k)}$, with parameters $\eta = O(1)$, $\eta_p = \tilde{O}\!\left(\delta^{\frac{3}{2}} \wedge \frac{\delta^3}{\varepsilon}\right)$, $k_{\mathrm{thres}} = \tilde{O}\!\left(\frac{1}{\delta}\right)$, and $F_{\mathrm{thres}} = \tilde{O}(\delta^3)$, the following holds with probability $1 - \zeta'$:*

$$F(\mu^{\dagger}) - F(\mu^{(k_{\mathrm{thres}})}) \geq F_{\mathrm{thres}}.$$

*Proof.* From the discussion at the beginning of the previous section, by setting $M \leq \left(\frac{eC}{e-1}\left(1 + 2\log\frac{2}{\zeta'}\right)\right)^{\frac{1}{2}} = \tilde{O}(1)$, it holds that $\|\xi\|_{L^2(\mu)} \leq M = \tilde{O}(1)$ with probability $1 - \frac{\zeta'}{2}$. By choosing the hyperparameters as in (38), we have $\eta = O(1)$, $\eta_p = \tilde{O}\!\left(\delta^{\frac{3}{2}} \wedge \frac{\delta^3}{\varepsilon}\right)$, $k_{\mathrm{thres}} = \tilde{O}\!\left(\frac{1}{\delta}\right)$, $F_{\mathrm{thres}} = \tilde{O}(\delta^3)$, and Lemma E.1 and Proposition E.6 can be applied. In a similar manner to the proof of Proposirion 5.4 (continuous version), it follows that there exists $0 \leq k \leq k_{\mathrm{thres}}$ such that

$$\mathrm{P}\left(F(\mu^{(0)}) - F(\mu^{(k)}) \geq 2F_{\mathrm{thres}}\right) \geq 1 - \frac{\zeta'}{2},$$

Thus, with probability $1 - \frac{\zeta'}{2}$, we have

$$F(\mu^{(0)}) - F(\mu^{(k_{\mathrm{thres}})}) \geq F(\mu^{(0)}) - F(\mu^{(k)}) \geq 2F_{\mathrm{thres}}.$$

Combining with Lemma E.1, the following holds:

$$\begin{aligned}
F(\mu^{\dagger}) - F(\mu^{(k_{\mathrm{thres}})}) = F(\mu^{\dagger}) - F(\mu^{(0)}) + F(\mu^{(0)}) - F(\mu^{(k_{\mathrm{thres}})}) \\
\geq -F_{\mathrm{thres}} + 2F_{\mathrm{thres}} \\
= F_{\mathrm{thres}}.
\end{aligned}$$

This occurs with probability more than $1 - \left(\frac{\zeta'}{2} + \frac{\zeta'}{2}\right) = 1 - \zeta'$. $\qquad\square$

With the above preparations, we finally prove the convergence of discrete-time PWGF to a second-order stationary point.

**Theorem 5.5.** *Let the initial point be $\mu_0 \in \mathcal{P}_2(\mathbb{R}^d)$ and denote $\Delta F = F(\mu_0) - \inf_{\mu \in \mathcal{P}_2(\mathbb{R}^d)} F(\mu)$. Set $\eta = O(1)$ and let $\varepsilon$, $\delta$, $\eta_p$, $T_{\mathrm{thres}}$, $F_{\mathrm{thres}}$ be chosen as in Theorem 5.2. Then, discrete time PWGF halts after*

$$k = \tilde{O}\left(\Delta F\left(\frac{1}{\varepsilon^2} + \frac{1}{\delta^4}\right)\right)$$

*steps and reaches an $(\varepsilon, \delta)$-second-order stationary point with probability $1 - \zeta$.*

*Proof.* Let $\varepsilon$, $\delta$, $\zeta > 0$ be chosen arbitrarily chosen such that $(L_2 + L_3)\,\varepsilon \leq \delta^2$, and set $\zeta' > 0$ such that $\zeta'$ is polynomial in $\frac{1}{\delta}$ and $\zeta$ up to logarithmic factors, provided later. By the settings of $\eta$, $\eta_p$, $k_{\mathrm{thres}}$, and $F_{\mathrm{thres}}$ as in Proposition E.7, we have $\eta = O(1)$, $\eta_p = \tilde{O}\!\left(\delta^{\frac{3}{2}} \wedge \frac{\delta^3}{\varepsilon}\right)$, $k_{\mathrm{thres}} = \tilde{O}\!\left(\frac{1}{\delta}\right)$, and $F_{\mathrm{thres}} = \tilde{O}(\delta^3)$.

From Proposition E.7, perturbations occur at most $m := \lceil \frac{\Delta F}{F_{\mathrm{thres}}} \rceil$ times. Thus, the probability of failure after $m$ perturbations is at most $1 - (1 - \zeta')^m \leq m\zeta'$. Setting $\zeta' = \frac{\zeta}{m}$ ensures that the algorithm reaches an $(\varepsilon, \delta)$-second order stationary point with probability at least $1 - \zeta$.

Discrete time PWGF determines whether the objective decreases by at least $F_{\mathrm{thres}}$ after a certain number of iterations $k_{\mathrm{thres}}$ following the application of a perturbation. Then, we define the period between the application of a perturbation and this evaluation as *State* 1, and all other times as *State* 0. Let $k_0$ denote the total time spent in State 0, where $\|\nabla_{\mu}F(\mu)\|_{L^2(\mu)} \geq \varepsilon$. By Lemma 5.3, the decrease in the objective function during this time is at least $\varepsilon^2 k_0$, implying $k_0 \leq \frac{\Delta F}{\varepsilon^2}$. Moreover, the total time $k_1$ in State 1 is upper bounded by $k_1 \leq m T_{\mathrm{thres}} = \frac{\Delta F k_{\mathrm{thres}}}{F_{\mathrm{thres}}} = \tilde{O}\!\left(\frac{1}{\delta^4}\right)$. Hence, the algorithm halts in $k_0 + k_1 = \tilde{O}\!\left(\Delta F\left(\frac{1}{\varepsilon^2} + \frac{1}{\delta^4}\right)\right)$ iterations. $\qquad\square$

# F    Numerical Experiments

## F.1    ICFL Functional

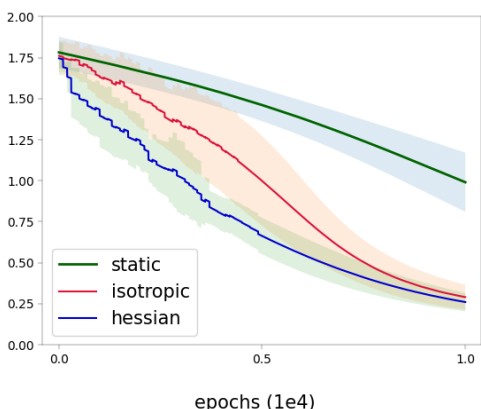

epochs (1e4)

Figure 1: Trajectories of the training loss for no-noise ("static"), isotropic noise ("isotropic") and Hessian guided noise ("hessian") settings.

We conducted numerical experiments using the loss functional of in-context learning of Transformers from Kim & Suzuki (2024) as the objective functional. See below for details.

We compared the dynamics of the loss function under three variants of WGF; WGF without noise (static), WGF with isotropic noise (isotropic), and WGF with noise guided by the Hessian (hessian).

As Figure 1 shows, the loss decreases gradually in the "static" case, whereas the "isotropic" and "hessian" cases exhibit a significant reduction in loss, leading to saturation. Furthermore, the Hessian-based noise demonstrates a more efficient decrease in loss.

**Experimental details.**    We provide a minimal explanation of the loss function used in our numerical experiments for in-context learning in Transformers. For an in-depth exposition on the problem setup, derivation of the loss functional, and an analysis of the loss landscape, we refer to the work by Kim & Suzuki (2024).

We consider a mean field two-layer neural network with a sigmoid activation function, which takes $l$-dimensional data inputs and $k$-dimensional data outputs :

$$h_\mu(z) = \int h_x(z)\mu(\mathrm{d}x) = \int a\sigma(w^\top z)\mu(\mathrm{d}x) \quad (x = (a, w) \in \mathbb{R}^k \times \mathbb{R}^l),$$

where $z \in \mathbb{R}^l$ is a given data and follows a certain distribution. We also define the following matrices :

$$\Sigma_{\mu,\nu} = \mathrm{E}_z\big[h_\mu(z)h_\nu(z)^\top\big] \quad (\mu,\nu \in \mathcal{P}_2(\mathbb{R}^{k+l})).$$

We consider performing in-context learning using $h_\mu$ as feedforward layer, followed by a reparametrized linear self-attention mechanism which can be described by a single attention matrix $W \in \mathbb{R}^{k \times k}$. The optimal value of $W$ is determined to satisfy $\Sigma_{\mu^o,\mu}W = \Sigma_{\mu^o,\mu}\Sigma_{\mu,\mu}^{-1}$ with given $\mu$. Thus, the objective with optimal $W$ is derived as follows:

$$F(\mu) = \frac{1}{2}\mathrm{E}\Big[\big\|h_{\mu^o}(z) - \Sigma_{\mu^o,\mu}\Sigma_{\mu,\mu}^{-1}h_\mu(z)\big\|^2\Big], \tag{44}$$

where $\mu^o \in \mathcal{P}_2(\mathbb{R}^{k+l})$ is the true feature and $\zeta_{\mu^o,\mu}(z) = h_{\mu^o}(z) - \Sigma_{\mu^o,\mu}\Sigma_{\mu,\mu}^{-1}h_\mu(z)$. The Wasserstein gradient of the objective (44) is computed as

$$\nabla_\mu F(\mu, a, w) = \begin{pmatrix} -\Sigma_{\mu,\mu}^{-1}\Sigma_{\mu,\mu^o}\mathrm{E}_z\big[\sigma(w^\top z)\zeta_{\mu^o,\mu}(z)\big] \\ a^\top\Sigma_{\mu,\mu}^{-1}\Sigma_{\mu,\mu^o}\mathrm{E}_z\big[\sigma'(w^\top z)\zeta_{\mu^o,\mu}(z)z^\top\big] \end{pmatrix}.$$

**Algorithm 3** PWGD ( time/ space discrete )

---

initialize $x_1^{(0)}, ...x_N^{(0)}, \mu^{(0)} \leftarrow \frac{1}{N}\sum_{j=1}^{N}\delta_{x_j^{(0)}}$

**for** $k = 0, 1, ...$ **do**

    **if** $\left\|\nabla_\mu F(\mu^{(k)})\right\|_{L^2(\mu^{(k)})} \leq \varepsilon$ and $k - k_{\rm p} > k_{\rm thres}$ **then**

        $\xi \sim {\rm GP}(0, k_{\mu^{(k)}})$

        $(\xi_1, ...\xi_N) \leftarrow (\xi(x_1^{(k)}), ..., \xi(x_N^{(k)}))$

        $x_j^{(k)} \leftarrow x_j^{(k)} + \eta_p \xi_j \quad (j = 1, ..., N)$

        $\mu^{(k)} \leftarrow \frac{1}{N}\sum_{j=1}^{N}\delta_{x_j^{(k)}}$

        $k_p \leftarrow k$

    **end if**

    **if** $k = k_{\rm p} + k_{\rm thres}$ and $F(\mu^{(k_{\rm p})}) - F(\mu^{(k)}) \leq F_{\rm thres}$ **then**

        **return** $\mu^{(k_{\rm p})}$

    **end if**

    $x_j^{(k+1)} \leftarrow x_j^{(k)} - \eta\nabla_\mu F(\mu)(\mu^{(k)}, x_j^{(k)}),\ (j = 1, ..., N),\ \mu^{(k+1)} \leftarrow \frac{1}{N}\sum_{j=1}^{N}\delta_{x_j^{(k+1)}}$

**end for**

---

Furthermore, Hessian $\nabla_\mu^2 F$ at a first-order optimal point $\mu$ is computed as

$$\nabla_\mu^2 F(\mu, a, w, b, v) = \begin{pmatrix} H_{11} & H_{12} \\ H_{21} & H_{22} \end{pmatrix}$$

where

$$
\begin{aligned}
&H_{11}(\mu, a, w, b, v) \\
&= \nabla_a \nabla_b^\top \frac{\delta^2 F}{\delta\mu^2}(\mu, a, w, b, v) \\
&= \left({\rm E}_z[\sigma(w^\top z)\sigma(v^\top z)] - {\rm E}_z[\sigma(w^\top z)h_\mu(z)]^\top \Sigma_{\mu,\mu}^{-1} {\rm E}_z[\sigma(v^\top z)h_\mu(z)]\right)\Sigma_{\mu,\mu}^{-1}\Sigma_{\mu,\mu^\circ}\Sigma_{\mu^\circ,\mu}\Sigma_{\mu,\mu}^{-1} \\
&\quad + {\rm E}_z[\sigma(w^\top z)\zeta_{\mu^\circ,\mu}(z)]^\top {\rm E}_z[\sigma(v^\top z)\zeta_{\mu^\circ,\mu}(z)]\Sigma_{\mu,\mu}^{-1}
\end{aligned}
$$

and

$$
\begin{aligned}
&H_{12}(\mu, a, w, b, v) = H_{21}(\mu, b, v, a, w)^\top \\
&= \nabla_a \nabla_v^\top \frac{\delta^2 F}{\delta\mu^2}(\mu, a, w, b, v) \\
&= \Sigma_{\mu,\mu}^{-1}\Sigma_{\mu,\mu^\circ}\Sigma_{\mu^\circ,\mu}\Sigma_{\mu,\mu}^{-1}b \\
&\quad \cdot \left({\rm E}_z\left[\sigma(w^\top z)\sigma'(v^\top z)z^\top\right] - {\rm E}_z\left[\sigma(w^\top z)h_\mu(z)\right]^\top \Sigma_{\mu,\mu}^{-1}{\rm E}_z\left[h_\mu(z)\sigma'(v^\top z)z^\top\right]\right) \\
&\quad - \Sigma_{\mu,\mu}^{-1}b{\rm E}_z\left[\sigma(w^\top z)\zeta_{\mu^\circ,\mu}(z)^\top\right]{\rm E}_z\left[\zeta_{\mu^\circ,\mu}(z)\sigma'(v^\top z)z^\top\right] \\
&\quad + \left({\rm E}_z\left[\sigma(w^\top z)h_\mu(z)^\top\right]\Sigma_{\mu,\mu}^{-1}b\right)\Sigma_{\mu,\mu}^{-1}\Sigma_{\mu,\mu^\circ}{\rm E}_z\left[\zeta_{\mu^\circ,\mu}(z)\sigma'(v^\top z)z^\top\right],
\end{aligned}
$$

as well as

$$
\begin{aligned}
&H_{22}(\mu, a, w, b, v) \\
&= \left(a^\top \Sigma_{\mu,\mu}^{-1}\Sigma_{\mu,\mu^\circ}\Sigma_{\mu^\circ,\mu}\Sigma_{\mu,\mu}^{-1}b\right) \\
&\quad \cdot \left({\rm E}_z\left[z\sigma'(w^\top z)\sigma'(v^\top z)z^\top\right] - {\rm E}_z\left[z\sigma'(w^\top z)h_\mu(z)^\top\right]\Sigma_{\mu,\mu}^{-1}{\rm E}_z\left[h_\mu(z)\sigma'(v^\top z)z^\top\right]\right) \\
&\quad - \left(a^\top\Sigma_{\mu,\mu}^{-1}b\right){\rm E}_z\left[z\sigma'(w^\top z)\zeta_{\mu^\circ,\mu}(z)^\top\right]{\rm E}_z\left[\zeta_{\mu^\circ,\mu}(z)\sigma'(w^\top z)z^\top\right] \\
&\quad + {\rm E}_z\left[z\sigma'(w^\top z)h_\mu(z)^\top\right]\Sigma_{\mu,\mu}^{-1}ba^\top\Sigma_{\mu,\mu}^{-1}\Sigma_{\mu,\mu^\circ}{\rm E}_z\left[\zeta_{\mu^\circ,\mu}(z)\sigma'(v^\top z)z^\top\right] \\
&\quad + {\rm E}_z\left[z\sigma'(w^\top z)\zeta_{\mu^\circ,\mu}(z)^\top\right]\Sigma_{\mu^\circ,\mu}\Sigma_{\mu,\mu}^{-1}ba^\top\Sigma_{\mu,\mu}^{-1}{\rm E}_z\left[h_\mu(z)\sigma'(v^\top z)z^\top\right].
\end{aligned}
$$

The experimental setup is as follows. We compared the dynamics of the loss function under three variants of WGF; WGF without noise (static), WGF with isotropic noise (isotropic), and PWGF

(hessian). To ensure a fair comparison of the three algorithms, no stopping criteria were incorporated into the algorithms.

The input and output dimensions were set to $l = 20,\ k = 5$. We approximated the measure using $400$ neurons and generated $800$ i.i.d. input data points $z$ from the standard normal distribution $\mathcal{N}(0, 1)$ for each coordinate. The optimization probability measure was randomly initialized and we conducted fiveive experiments under the same conditions, plotting the mean and standard deviation. We used parameters as $\eta_p = 0.015, k_{\text{thres}} = 100$. In addition, SGD was used in the optimization process with the learning rate $\eta = 10^{-7}$.

### F.2 Matrix-Decomposition Functional

Next, we conducted experiments using the matrix decomposition setting presented in Example 1. Details of the objective function, including analytical expressions for the gradient and Hessian, as well as a proposition suggesting strict benignity of the matrix decomposition objective, are provided in Appendix G.

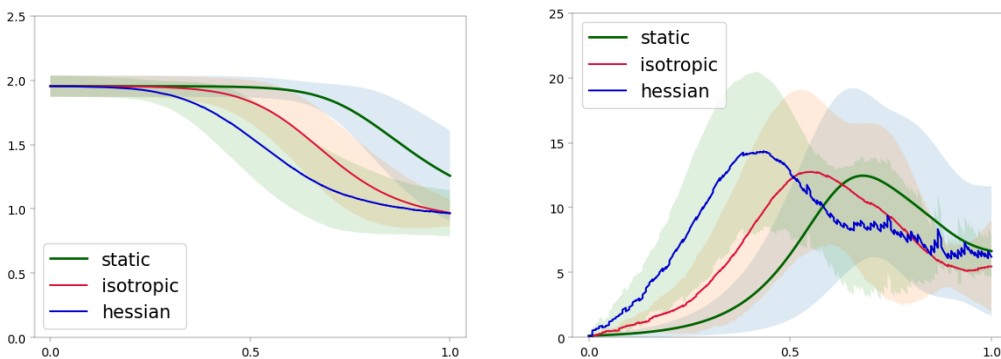

Figure 2: Trajectories of the training loss and the norm of the gradient for no-noise ("static"), isotropic noise ("isotropic") and Hessian guided noise ("hessian") settings.

Due to the stochastic nature of the algorithms, we report the mean and standard deviation over 10 runs, with the corresponding error bars. As shown in Figure 2, the Hessian and isotropic noise injection methods achieve faster objective reduction and exhibit earlier peaks in the gradient norm, demonstrating a more efficient escape from the initial critical point. In contrast, the perturbation-free method tends to stagnate for longer periods. The Hessian method shows the best performance, although the performance of isotropic noise is comparable. The effectiveness of isotropic noise can be attributed to the fact that the infinite-dimensional nature of the problem has not yet manifested due to the number of particles used in the approximation.

In practice, at points where the gradient norm is small, but the point is not a saddle, adding noise may hinder the gradient descent. This issue is particularly pronounced for the method with Hessian noise, where the magnitude of the noise depends on the local curvature. Consequently, whether to inject noise should be determined adaptively in combination with the criteria discussed above.

**Experimental details.** The input and output dimensions are $l = 15, k = 5$. We approximate the measure using $400$ neurons and generate $800$ i.i.d. input data points $z$ from the standard normal distribution $\mathcal{N}(0, 1)$ for each coordinate. Similarly to the ICFL case, SGD was used in the optimization process and the learning rate $\eta = 10^{-6}$. We set $k_{\text{thres}} = 100,\ \eta_p = 3 \times 10^{-3},\ F_{\text{thres}} = 10^{-2}$.

# G  Landscape Analysis of Matrix Decomposition

We analyze matrix factorization (Example 1) as an example of non-convex stochastic optimization:

$$
\begin{aligned}
F(\mu) &= \frac{1}{2}\mathrm{E}_z[\|h_\mu(z)h_\mu(z)^\top - h_{\mu^o}(z)h_{\mu^o}(z)^\top\|^2] \\
&= \frac{1}{2}E_z\big[\|M_\mu(z) - M_{\mu^o}(z)\|^2\big] \\
&= \frac{1}{2}E_z\big[\|\Delta M_\mu(z)\|^2\big],
\end{aligned}
$$

where we set

$$
\begin{aligned}
h_\mu(z) &= \int h_{a,w}(z)\mu(\mathrm{d}a\mathrm{d}w) \\
h_{a,w}(z) &= h_{a,w}(z) \\
&= a\sigma(w^\top z) \\
M_\mu(z) &= h_\mu(z)h_\mu(z)^\top \\
\Delta M_\mu(z) &= M_\mu(z) - M_{\mu^o}(z)
\end{aligned}
$$

The Wasserstein gradient is computed as:

$$
\begin{aligned}
\nabla_\mu F(\mu, a_1, a_2, w) &= 2\mathrm{E}_z\left[\nabla_{a,w}h_{a,w}(z)\Delta M_\mu(z)h_\mu(z)\right] \\
&= \begin{pmatrix} 2\mathrm{E}_z\left[\sigma(w^\top z)\Delta M_\mu(z)h_\mu(z)\right] \\ 2\mathrm{E}_z\left[z\sigma'(w^\top z)h_\mu(z)^\top \Delta M_\mu(z)\right]a_1 \end{pmatrix}
\end{aligned}
\tag{45}
$$

The Hessian is computed as:

$$
\nabla_\mu^2 F(\mu, x, y) = \nabla_\mu^2 F(\mu, a, w, b, v)
\tag{46}
$$

$$
= \mathrm{E}_z\left[\nabla_x h_x(z)\left(2M_\mu(z) + \|h_\mu(z)\|^2 I_k - M_{\mu^o}(z)\right)(\nabla_y h_y(z))^\top\right]
\tag{47}
$$

From (45) and (47),

- For any $\mu \in \mathcal{P}_2(\mathbb{R}^l)$, $\mu = \delta_0 \otimes \tilde{\mu}$ is a strict saddle point.

- $\mu = (\pm\mathrm{Id}_{\mathbb{R}^k}) \times (\mathrm{Id}_{\mathbb{R}^l})\#\mu^o$ is a global optima.

Furthermore, by a similar argument to Ge et al. (2017), we can deduce the following proposition. This proposition asserts that for an $\varepsilon$-stationary point $\mu$ which is not a global minimizer, the objective function can be strictly decreased. This suggests that $F$ possesses strict benignity.

**Proposition G.1.** *Let $\mu \in \mathcal{P}(\mathbb{R}^d)$ be $\varepsilon$-stationary and not a global optima, i.e; satisfy $\|\nabla_\mu F(\mu)\| \le \varepsilon$ and $F(\mu) \neq 0$. If $h_\mu(z) \ge 0$ a.s. [7], Assumption 2 and $W_2(\mu, \mu^o) \le C$ hold, then there exists a curve $\mu_t$ s.t. $\mu_0 = \mu$ and at $t = 0$,*

$$
\frac{\mathrm{d}^2}{\mathrm{d}t^2}F(\mu_t) \le -\mathrm{E}_z[\|\left(h_{\mu_t}h_{\mu_t}^\top - h_{\mu^o}h_{\mu^o}^\top\right)(z)\|_F^2] + O(\varepsilon).
$$

*Proof.* We define a curve $\mu_t$ by $\mu_t = (1 - t)\mu + t\mu^o$. Then we obtain at $t = 0$

$$
\begin{aligned}
\frac{\mathrm{d}}{\mathrm{d}t}F(\mu_t) &= \mathrm{E}_z\left[\mathrm{tr}\left(\left(\frac{\mathrm{d}}{\mathrm{d}t}(h_{\mu_t}h_{\mu_t}^\top - h_{\mu^o}h_{\mu^o}^\top)(h_{\mu_t}h_{\mu_t}^\top - h_{\mu^o}h_{\mu^o}^\top)\right)(z)\right)\right] \\
&= \mathrm{E}_z\left[\mathrm{tr}\left(\left(\frac{\mathrm{d}}{\mathrm{d}t}h_{\mu_t}h_{\mu_t}^\top + h_{\mu_t}\frac{\mathrm{d}}{\mathrm{d}t}h_{\mu_t}^\top\right)(h_{\mu_t}h_{\mu_t}^\top - h_{\mu^o}h_{\mu^o}^\top)(z)\right)\right],
\end{aligned}
\tag{48}
$$

---

[7]This condition can be regarded as an extension of non-negative matrix factorization.

$$\frac{\mathrm{d}^2}{\mathrm{d}t^2}F(\mu_t) = \mathrm{E}_z\left[\mathrm{tr}\left(\left(\frac{\mathrm{d}^2}{\mathrm{d}t^2}h_{\mu_t}h_{\mu_t}^\top + h_{\mu_t}\frac{\mathrm{d}^2}{\mathrm{d}t^2}h_{\mu_t}^\top\right)(h_{\mu_t}h_{\mu_t}^\top - h_{\mu^\circ}h_{\mu^\circ}^\top)(z)\right)\right]$$

$$+ \mathrm{E}_z\left[\mathrm{tr}\left(\left(2\frac{\mathrm{d}}{\mathrm{d}t}h_{\mu_t}\frac{\mathrm{d}}{\mathrm{d}t}h_{\mu_t}^\top\right)^\top(h_{\mu_t}h_{\mu_t}^\top - h_{\mu^\circ}h_{\mu^\circ}^\top)(z)\right)\right]$$

$$+ \mathrm{E}_z\left[\mathrm{tr}\left(\left(\frac{\mathrm{d}}{\mathrm{d}t}h_{\mu_t}h_{\mu_t}^\top + h_{\mu_t}\frac{\mathrm{d}}{\mathrm{d}t}h_{\mu_t}^\top\right)^\top\left(\frac{\mathrm{d}}{\mathrm{d}t}h_{\mu_t}h_{\mu_t}^\top + h_{\mu_t}\frac{\mathrm{d}}{\mathrm{d}t}h_{\mu_t}^\top\right)(z)\right)\right]$$

$$= \mathrm{E}_z\left[(4-3)\mathrm{tr}\left((h_{\mu_t}h_{\mu_t}^\top - h_{\mu^\circ}h_{\mu^\circ}^\top)^\top(h_{\mu_t}h_{\mu_t}^\top - h_{\mu^\circ}h_{\mu^\circ}^\top)(z)\right)\right]$$

$$+ 4\mathrm{E}_z\left[\mathrm{tr}\left(\left(\frac{\mathrm{d}}{\mathrm{d}t}h_{\mu_t}\frac{\mathrm{d}}{\mathrm{d}t}h_{\mu_t}^\top\right)^\top(h_{\mu_t}h_{\mu_t}^\top - h_{\mu^\circ}h_{\mu^\circ}^\top)(z)\right)\right]$$

$$+ \mathrm{E}_z\left[\mathrm{tr}\left(\frac{\mathrm{d}}{\mathrm{d}t}h_{\mu_t}\frac{\mathrm{d}}{\mathrm{d}t}h_{\mu_t}^\top\frac{\mathrm{d}}{\mathrm{d}t}h_{\mu_t}\frac{\mathrm{d}}{\mathrm{d}t}h_{\mu_t}^\top(z)\right)\right]$$

$$= \mathrm{E}_z\left[\mathrm{tr}\left(\frac{\mathrm{d}}{\mathrm{d}t}h_{\mu_t}\frac{\mathrm{d}}{\mathrm{d}t}h_{\mu_t}^\top\frac{\mathrm{d}}{\mathrm{d}t}h_{\mu_t}\frac{\mathrm{d}}{\mathrm{d}t}h_{\mu_t}^\top(z)\right)\right]$$

$$- 3\mathrm{E}_z\left[\mathrm{tr}\left((h_{\mu_t}h_{\mu_t}^\top - h_{\mu^\circ}h_{\mu^\circ}^\top)^\top(h_{\mu_t}h_{\mu_t}^\top - h_{\mu^\circ}h_{\mu^\circ}^\top)(z)\right)\right]$$

$$- 4\frac{\mathrm{d}}{\mathrm{d}t}F(\mu_t), \tag{49}$$

where we used, in the sixth line, the equations;

$$\left.\frac{\mathrm{d}^2}{\mathrm{d}t^2}\right|_{t=0}h_{\mu_t}(z) = 0,$$

$$\left.\frac{\mathrm{d}}{\mathrm{d}t}\right|_{t=0}h_{\mu_t}h_{\mu_t}^\top(z) + h_{\mu_t}\left.\frac{\mathrm{d}}{\mathrm{d}t}\right|_{t=0}h_{\mu_t}^\top(z) = (h_\mu - h_{\mu^\circ})h_\mu^\top(z) + h_\mu(h_\mu - h_{\mu^\circ})^\top(z)$$

$$= -(h_\mu h_\mu^\top - h_{\mu^\circ}h_{\mu^\circ})(z) - (h_\mu - h_{\mu^\circ})(h_\mu - h_{\mu^\circ})^\top(z)$$

$$= -(h_\mu h_\mu^\top - h_{\mu^\circ}h_{\mu^\circ})(z) - \left.\frac{\mathrm{d}}{\mathrm{d}t}\right|_{t=0}h_{\mu_t}\left.\frac{\mathrm{d}}{\mathrm{d}t}\right|_{t=0}h_{\mu_t}^\top(z),$$

and in the ninth line;

$$\left.\frac{\mathrm{d}}{\mathrm{d}t}F(\mu_t)\right|_{t=0} = -\mathrm{E}_z\left[\mathrm{tr}\left((h_\mu h_\mu^\top - h_{\mu^\circ}h_{\mu^\circ}^\top)(h_\mu h_\mu^\top - h_{\mu^\circ}h_{\mu^\circ}^\top)(z)\right)\right]$$

$$-\mathrm{E}_z\left[\mathrm{tr}\left(\frac{\mathrm{d}}{\mathrm{d}t}h_{\mu_t}\frac{\mathrm{d}}{\mathrm{d}t}h_{\mu_t}^\top(h_{\mu_t}h_{\mu_t}^\top - h_{\mu^\circ}h_{\mu^\circ}^\top)(z)\right)\right]\Bigg|_{t=0},$$

which are derived from the definition of $\mu_t$.

Noting that

$$\left.\left\|\frac{\mathrm{d}}{\mathrm{d}t}h_{\mu_t}\frac{\mathrm{d}}{\mathrm{d}t}h_{\mu_t}^\top\right\|_F^2(z)\right|_{t=0} \leq 2\left\|h_{\mu_t}h_{\mu_t}^\top - h_{\mu^\circ}h_{\mu^\circ}^\top\right\|_F^2(z), \tag{50}$$

which is obtained by straightforward calculation and the assumption $h_\mu \geq 0$ a.s.;

$$2\left\|h_{\mu_t}h_{\mu_t}^\top - h_{\mu^\circ}h_{\mu^\circ}^\top\right\|_F^2(z) - \left.\left\|\frac{\mathrm{d}}{\mathrm{d}t}h_{\mu_t}\frac{\mathrm{d}}{\mathrm{d}t}h_{\mu_t}^\top\right\|_F^2(z)\right|_{t=0}$$

$$= 2\left\|h_\mu h_\mu^\top - h_{\mu^\circ}h_{\mu^\circ}^\top\right\|_F^2(z) - \left\|(h_{\mu^\circ} - h_\mu)(h_{\mu^\circ} - h_\mu)^\top\right\|_F^2(z)$$

$$= \|h_\mu\|^4(z) + \|h_{\mu^\circ}\|^4(z) - 2|h_\mu^\top h_{\mu^\circ}|^2(z) - \|h_\mu - h_{\mu^\circ}\|^4(z)$$

$$= \left(\|h_\mu\|^2(z) - \|h_{\mu^\circ}\|^2(z)\right)^2 + 4h_\mu^\top h_{\mu^\circ}\|h_\mu - h_{\mu^\circ}\|^2(z)$$

$$\geq 0$$

Then we obtain, from (49) and (50),

$$\left.\frac{d^2}{dt^2}\right|_{t=0} F(\mu_t) \leq -E\left[\left\|h_\mu h_\mu^\top - h_{\mu^o} h_{\mu^o}^\top\right\|_F^2(z)\right] - 4 \left.\frac{d}{dt}\right|_{t=0} F(\mu_t). \tag{51}$$

Finally we will show that

$$\left|\left.\frac{d}{dt}\right|_{t=0} F(\mu_t)\right| = 2\left|E_z\left[(h_{\mu^o} - h_\mu)^\top \delta M_\mu(z) h_\mu(z)\right]\right|$$

$$\leq \tilde{C}\|\nabla_\mu F(\mu)\|_{L^2(\mu)} = O(\varepsilon) \tag{52}$$

for some $\tilde{C} > 0$. For any $\gamma \in \Gamma_o(\mu, \mu^o)$,

$$\left|\left.\frac{d}{dt}\right|_{t=0} F(\mu_t)\right|$$
$$= 2\left|E_z\left[(h_{\mu^o} - h_\mu)^\top \Delta M_\mu(z)) h_\mu(z)\right]\right|$$
$$= 2\int E_z[(h_x(z) - h_y(z))^\top \Delta M_\mu(z) h_\mu(z)]\gamma(dxdy)$$
$$= 2\int (x-y)^\top E_z[\nabla h_{x+\theta(y-x)}(z)^\top \Delta M_\mu(z) h_\mu(z)]\gamma(dxdy)$$
$$= 2\int \left((x-y)^\top \nabla_\mu F(\mu, x) + (x-y)^\top \nabla \nabla_\mu F(\mu, x + \tilde{\theta}(y-x))(x-y)\right)\gamma(dxdy)$$
$$\leq 2W_2(\mu, \mu^o)\|\nabla_\mu F(\mu)\|_{L^2(\mu)} + W_2(\mu, \mu^o)^2 \sup_x \|\nabla \nabla_\mu F(\mu, x)\|$$
$$\leq 2(C + R_2 C^2)\|\nabla_\mu F(\mu)\|_{L^2(\mu)}$$

where Taylor's expansion is used and $\theta, \tilde{\theta} \in [0, 1]$ in the third and fourth lines, the Cauchy–Schwarz inequality and the definition of the 2-Wasserstein distance in the fifth line, and the assumptions $W_2(\mu, \mu^o) \leq C$ and Assumption 2 in the sixth line. Setting $\tilde{C} = 2(C + R_2 C^2)$, we obtain (52) and hence, from (51),

$$\left.\frac{d^2}{dt^2}\right|_{t=0} F(\mu_t) \leq -E\left[\left\|h_\mu h_\mu^\top - h_{\mu^o} h_{\mu^o}^\top\right\|_F^2(z)\right] + O(\varepsilon).$$

$$\square$$

