# OpenReview forum: "Hessian-guided Perturbed Wasserstein Gradient Flows for Escaping Saddle Points"
_NeurIPS.cc/2025/Conference — NeurIPS 2025 poster_

### Official Review · Reviewer_cKxd · 2025-06-26

**Clarity:** 3
**Significance:** 2
**Originality:** 2
**Rating:** 5
**Confidence:** 3

**Summary:**

The authors propose a novel algorithm perturbed Wasserstein Gradient flow (PWGF) to address the issue of how to escape saddle points in probability distribution optimization problems using Hessian perturbation. The idea is that, in order to escape a saddle point, it is more efficient to go along the direction of the smallest eigenvalue. Under reasonable assumptions, for problems satisfying a strict benignity propert, the authors establish polynomial-time convergence to second-order stationary points, both in continuous and discrete setups.

**Questions:**

- Can the authors provide experiments demonstrating the advantage of Hessian-based PWGF on realistic tasks that do not necessarily satisfy the strict benignity condition?

I will raise my score if more experiments can be provided.

**Ethical Concerns:**

["NO or VERY MINOR ethics concerns only"]

**Final Justification:**

The authors provided a detailed literature review on Hessian-based methods and interesting experiments on the matrix factorization problem using a mean-field neural network.

**Quality:**

3

**Strengths And Weaknesses:**

Strengths:

- The paper is well written and presents a detailed theoretical framework.

- The convergence analysis is technically sound.

Weaknesses:

- Literature on the benefit of the Hessian-based saddle-point escape method, even in the Euclidean spaces, is missing.

- Although the authors provide examples of non-convex objective functions that exhibit strict benignity, they are far from practice.

---

> ### Author Rebuttal · Authors · 2025-07-29
>
> Thank you for the insightful suggestions, they have helped us greatly to improve the manuscript! Below are our responses.
>
> **Weakness 1:**
>
> We have added an overview of Hessian-based methods in finite dimensions and related works in the manuscript.
>
> * **Hessian-based methods:** The most basic second-order optimization method that utilizes the Hessian of the loss is Newton's method, which however may not converge in general. Global complexity guarantees were first established in [NP06], which studied a cubic regularization of Newton method. More sophisticated methods such as ARC [CGT11a,CGT11b] minimize an adaptive cubic-regularized second-order approximation of the loss and achieve an iteration complexity of $O(\epsilon^{-3/2})$ for nonconvex optimization, which has been shown to be optimal in certain settings. More recently, [CRS17] proposed the TRACE algorithm which solves a regularized trust region problem at each step, also achieving an iteration complexity of $O(\epsilon^{-3/2})$. These have been adapted to constrained optimization in e.g. [NR20].
> * Alternatively, the Hessian-vector product can be used to run negative curvature descent methods [LY17,CR19] without computing the full Hessian inverse. Such methods can also be converted into first-order algorithms such as Neon2 [AZL18]. See the discussion in the cited works for a more comprehensive overview. We remark that it is unclear how to adapt these methods to the distributional setting as optimizing even a single step of the second-order approximation of the loss is in general intractable. Nevertheless, these represent potential alternative approaches to achieve second-order optimization in Wasserstein space.
>
> * **Perturbation methods:** since PWGF is primarily first-order gradient descent combined with random perturbations, the most relevant algorithms in finite dimensions are perturbed gradient descent and its variants, which we cover in the introduction.
>
> * **Noisy methods:** noise-injected updates such as stochastic gradient Langevin dynamics are also effective in optimizing particle systems with nonconvex objectives. However, the randomness vanishes in the mean-field limit since the dynamics converges to the *deterministic* WGF w.r.t. the lifted objective with entropy regularization, which does not help if the resulting landscape is nonconvex on Wasserstein space. Our method can be seen an effective way to inject noise in infinite dimensions using directed Gaussian processes.
>
>
> **Weakness 2/Question:**
>
> We have **added new experiments and a detailed landscape analysis** of the matrix factorization problem using a mean-field neural network (Example 1) in Appendix F.2 and G which suggest the presence of strict benignity and show that noise injection helps escape the initial critical point. Since NeurIPS this year does not allow updating the submission pdf nor adding anonymous links during rebuttal, we summarize the results in text here.
>
> * Motivation: Classical optimization problems such as matrix factorization and tensor decomposition are known to be affected by saddle points. These problems arise in practical scenarios such as image reconstruction, object detection via matrix sensing, and recommendation systems [ZHBL10, ZKLR13, Y08]. In Euclidean space, the landscape of the objective function for matrix factorization has been extensively studied [GHJY15, GJZ17]. In particular, it is known that saddle points exist, and that all non-saddle stationary points are global optima for these types of objectives. (We perform the corresponding mean-field landscape analysis in the revised Appendix G.) A major concern in such situations is that gradient updates can get trapped near saddle points. For example, passing through neighborhoods of saddle points using gradient-based methods can take arbitrarily long time [DJL+17]. Therefore, escaping saddle points is crucial in such settings.
> * Experimental details: The input and output dimensions are $l=15,k=5$. We approximate the measure using $400$ neurons and generate $800$ i.i.d. input data points $z$ from the standard normal distribution $\mathcal{N}(0,1)$ for each coordinate. Similarly to the ICFL case, SGD was used in the optimization process and the learning rate $\eta = 10^{-6}$. We set $k_{\mathrm{thres}} = 100, ~ \eta_p = 3\times 10^{-3}, ~ F_{\mathrm{thres}} = 10^{-2}$. We report the mean and standard deviation over 10 runs, with the corresponding error bars.
> * We find that both Hessian and isotropic noise injection methods achieve faster objective reduction and exhibit earlier peaks in the gradient norm, demonstrating a more efficient escape from the initial critical point. In contrast, the perturbation-free method tends to stagnate for longer periods. In particular, the Hessian method shows the best performance, although the performance of isotropic noise is comparable.
>
> We thank the reviewer again for their time and humbly ask to consider updating their score in light of the added results.
>
> **References**
>
> [AZL18] Zeyuan Allen-Zhu, Yuanzhi Li. NEON2: Finding Local Minima via First-Order Oracles. NeurIPS, 2018.
>
> [CGT11a] Cartis, C., Gould, N.I.M., Toint, Ph.L. Adaptive cubic regularisation methods for unconstrained optimization. Part I: Motivation, convergence and numerical results. Math. Program. 127, 245–295, 2011.
>
> [CGT11b] Cartis, C., Gould, N.I.M., Toint, Ph.L. Adaptive cubic regularisation methods for unconstrained optimization. Part II: Worst-case function- and derivative-evaluation complexity. Math. Program. 130, 295–319, 2011.
>
> [CR19] F. E. Curtis and D. P. Robinson. Exploiting negative curvature in deterministic and stochastic optimization. Mathematical Programming: Series A and B, Volume 176, Issue 1-2, 69-94, 2019.
>
> [CRS17] F. E. Curtis, D. P. Robinson, and M. Samadi. A trust region algorithm with a worst-case iteration complexity of $o(\epsilon^{−3/2})$ for nonconvex optimization. Mathematical Programming, 162(1-2):1–32, 2017.
>
> [LY17] Mingrui Liu, Tianbao Yang. On Noisy Negative Curvature Descent: Competing with Gradient Descent for Faster Non-convex Optimization. arXiv:1709.08571, 2017.
>
> [NP06] Y. Nesterov and B. T. Polyak. Cubic regularization of newton method and its global performance. Mathematical Programming, 108(1):177–205, 2006.
>
> [NR20] M. Nouiehed and M. Razaviyayn. A Trust Region Method for Finding Second-Order Stationarity in Linearly Constrained Nonconvex Optimization.
> SIAM Journal on Optimization 30(3):2501-2529, 2020.
>
> [ZHBL10] Bo Zhao, Justin P. Haldar, Cornelius Brinegar, Zhi-Pei Liang. Low rank matrix recovery for real-time cardiac MRI. In 2010 IEEE International Symposium on Biomedical Imaging: From Nano to Macro, pp.996–999. IEEE, 2010.
>
> [ZKLR13] Wenbin Zou, Kidiyo Kpalma, Zhi Liu, Joseph Ronsin. Segmentation driven low-rank matrix recovery for saliency detection. In 24th British machine vision conference (BMVC), pp.1–13, 2013.
>
> [Y08] Koren, Yehuda. Factorization meets the neighborhood: a multifaceted collaborative filtering model. In Proceedings of the 14th ACM SIGKDD international conference on knowledge discovery and data mining. p.426-434. 2008.
>
> [GHJY15] Rong Ge, Furong Huang, Chi Jin, Yang Yuan. Escaping From Saddle Points—Online Stochastic Gradient for Tensor Decomposition. COLT, 2015.
>
> [GJZ17] Rong Ge, Chi Jin, Yi Zheng. No Spurious Local Minima in Nonconvex Low Rank Problems: A Unified Geometric Analysis. ICML, 2017.
>
> [DJL+17] Simon S Du, Chi Jin, Jason Lee, Michael I Jordan, Aarti Singh, Barnabas Poczos. Gradient Descent Can Take Exponential Time to Escape Saddle Points. NeurIPS, 2017.

---

> > ### Comment · Reviewer_cKxd · 2025-08-01
> >
> > Thank you for the literature review and the added interesting experiments. I will raise my score accordingly.

---

### Official Review · Reviewer_hSh6 · 2025-07-02

**Clarity:** 3
**Significance:** 2
**Originality:** 2
**Rating:** 5
**Confidence:** 3

**Summary:**

This paper introduces Perturbed Wasserstein Gradient Flow (PWGF), an algorithm for optimizing nonconvex functionals over probability measures. PWGF enhances standard Wasserstein gradient flow by introducing a perturbation with a Hessian-based kernel and thus efficiently escaping saddle points. The authors also provide the computational complexity of the proposed PWGF method for finding the ε-second-order stationary point.

**Questions:**

I am not very familiar with Wasserstein Gradient Flow, as my background is primarily in continuous optimization. I have a question regarding the algorithm's design: Given that it frequently computes the Hessian, have the authors considered incorporating second-order methods to leverage this information for improved optimization performance?

**Ethical Concerns:**

["NO or VERY MINOR ethics concerns only"]

**Final Justification:**

The authors have provided detailed responses and additional experiments, which largely addressed my concerns.

**Limitations:**

See weaknesses.

**Paper Formatting Concerns:**

No.

**Quality:**

3

**Strengths And Weaknesses:**

**strength**:
1. A rigorous convergence analysis is provided. In particular, the authors show the computational complexity of the proposed PWGF method for finding the ε-second-order stationary point, under standard smoothness and strict-saddle assumptions.
2. The writing is clear and well-structured.

**weakness**:
1. The proposed algorithm requires computing Hessian or Hessian-vector products for each particle or for the distribution, which is computationally expensive.
2. The numerical experiments are only on the loss functional of in-context learning of Transformers. A few more complex examples or ablation studies would broaden the practical impact.

---

> ### Author Rebuttal · Authors · 2025-07-29
>
> Thank you for the insightful review! Our responses are as follows.
>
> **(Weakness 1) The proposed algorithm requires computing Hessian or Hessian-vector products which is computationally expensive.**
>
> Our primary goal is to initiate a theoretical investigation of perturbative methods for nonconvex probability distribution optimization, analogous to how perturbed GD provably achieves second-order optimality in finite dimensions, and hence we did not focus on computational optimality. Indeed, the use of Hessian-informed perturbations is natural: PGD in $d$ dimensions works with uniformly random perturbations since there will be a nonzero correlation with the desired escape direction with high probability, resulting in a $\text{polylog}(d)$ rate. However, this argument breaks down when $d=\infty$ (and also due to the non-Euclidean nature of Wasserstein space/WGF), and we need directed noise to efficiently escape the saddle. Our algorithm provides a first-of-its-kind construction of a concrete Gaussian process-based perturbation which provably attains second-order optimality in the challenging infinite-dimensional setting.
>
> Nonetheless, there are ways to improve the computational cost of the process. In finite dimensions, algorithms such as NEON2 [AZL18] can convert Hessian-vector product computations to first-order approximations. This idea may be leveraged to derive a fully first-order WGF algorithm with provable second-order optimality.
>
> **(Weakness 2) The numerical experiments are only on the loss functional of in-context learning of Transformers.  A few more complex examples or ablation studies would broaden the practical impact.**
>
> We have **added new experiments and a detailed landscape analysis** of the matrix factorization problem using a mean-field neural network (Example 1) in Appendix F.2 and G which suggest the presence of strict benignity and show that noise injection helps escape the initial critical point. Since NeurIPS this year does not allow updating the submission pdf nor adding anonymous links during rebuttal, we summarize the results in text here.
>
> * Experimental details: The input and output dimensions are $l=15,k=5$. We approximate the measure using $400$ neurons and generate $800$ i.i.d. input data points $z$ from the standard normal distribution $\mathcal{N}(0,1)$ for each coordinate. Similarly to the ICFL case, SGD was used in the optimization process and the learning rate $\eta = 10^{-6}$. We set $k_{\mathrm{thres}} = 100, ~ \eta_p = 3\times 10^{-3}, ~ F_{\mathrm{thres}} = 10^{-2}$. We report the mean and standard deviation over 10 runs, with the corresponding error bars.
> * We find that both Hessian and isotropic noise injection methods achieve faster objective reduction and exhibit earlier peaks in the gradient norm, demonstrating a more efficient escape from the initial critical point. In contrast, the perturbation-free method tends to stagnate for longer periods. In particular, the Hessian method shows the best performance, although the performance of isotropic noise is comparable.
>
> **(Question) Given that it frequently computes the Hessian, have the authors considered incorporating second-order methods?**
>
> It seems nontrivial to directly apply second-order methods to nonconvex distributional optimization as it is unclear how to minimize even the local regularized second-order approximation of the loss functional; the kernel we compute for PWGF cannot be directly used for this purpose, and we are unaware of any existing works which study this problem. (A Wasserstein Newton's flow is studied in [WL20], but only obtain local convergence guarantees assuming strong convexity, and do not consider the nonconvex/saddle problem.) Nonetheless, second-order methods represent a potential alternative approach from our paper. We have provided a review of Hessian-based second-order methods in Euclidean space in our response to Reviewer cKxd, which may be leveraged for this purpose.
>
> [AZL18] Zeyuan Allen-Zhu and Yuanzhi Li. NEON2: Finding Local Minima via First-Order Oracles. NeurIPS, 2018.
>
> [WL20] Yifei Wang and Wuchen Li. Information Newton’s flow: second-order optimization method in probability space. arXiv:2001.04341, 2020.

---

### Official Review · Reviewer_FN5C · 2025-07-03

**Clarity:** 4
**Significance:** 3
**Originality:** 3
**Rating:** 5
**Confidence:** 3

**Summary:**

This paper introduces Perturbed Wasserstein Gradient Flow (PWGF), an algorithm for optimizing non-convex functionals over the space of probability measures. PWGF augments the usual Wasserstein gradient flow by injecting Hessian-guided Gaussian perturbations near saddle points, ensuring efficient escape and convergence to second-order stationary points. The main convergence result shows that PWGF reaches an $(\epsilon,\delta)$-second-order stationary point in $\tilde{O}(1/\epsilon^2 + 1/\delta^2)$ time steps with high probability, and under a strict “benignity” assumption, it further converges to a global optimum.

**Questions:**

1. Can you provide matching lower bounds (even in simpler settings) to argue that the derived rate is unimprovable?

2.  Is there a way to extend the analysis to the N-particle system, quantifying how the error between particle flow and the true flow scales with N and time?

3. Given the cost of sampling from the Gaussian process, are there efficient approximations that preserve the escape guarantees?

**Ethical Concerns:**

["NO or VERY MINOR ethics concerns only"]

**Final Justification:**

See my reply to the rebuttal.

**Limitations:**

Yes.

**Paper Formatting Concerns:**

None.

**Quality:**

4

**Strengths And Weaknesses:**

Strengths

1. The paper provides a meticulous infinite-dimensional analysis, including (i) second-order optimality conditions in Wasserstein space (Prop. 3.2–3.3), (ii) a novel construction of the Hessian-based kernel, and (iii) a clear proof of polynomial-time escape from saddles.

2. Unlike isotropic noise, the Hessian-guided GP perturbations are directed along the most negative curvature, mirroring perturbed-gradient-descent strategies in Euclidean space.

Weakness:

1. It would be nice if the paper could touch base on some practical issues as detailed in the question/limitation box below.

---

> ### Author Rebuttal · Authors · 2025-07-29
>
> Thank you for your detailed review and insightful questions! Below are our responses.
>
> **(Q1) Can you provide matching lower bounds (even in simpler settings) to argue that the derived rate is unimprovable?**
>
> Our primary goal was to initiate a theoretical investigation of perturbative methods for nonconvex probability distribution optimization and achieve a polynomial rate which is already first-of-its-kind, so we did not focus on optimality. Naive WGF can of course take arbitrarily long time to escape a saddle depending on how close it was initialized to the saddle point, and so global rates are impossible to obtain as in finite dimensions.
>
> In fact, with $\delta^2\sim\epsilon$, the derived rate is $\widetilde{O}(\epsilon^{-2})$ which we believe is potentially suboptimal. In finite dimensions, sophisticated second-order methods such as ARC [CGT11a] which minimizes an adaptive cubic-regularized local second-order approximation of the loss, or TRACE [CRS17] which solves a regularized trust region problem at each step, achieve an iteration complexity of $O(\epsilon^{-3/2})$ for nonconvex optimization, which has been shown to be optimal in certain settings [CGT11b]. These results (neither lower bounds nor the proposed methods) cannot be translated directly to the distributional optimization problem, however, so we anticipate more work is needed to rigorously establish optimal bounds and develop matching algorithms.
>
> **(Q2) Is there a way to extend the analysis to the N-particle system, quantifying how the error between particle flow and the true flow scales with N and time?**
>
> Existing analyses of $N$-particle approximation error (the so-called mean-field propagation of chaos) either blow up exponentially in time via a naive Gronwall-type bound [MMN18, MMM19, DBDFS20], or rely on uniform log-Sobolev arguments to prove uniform-in-time bounds, which only hold for strongly convex systems with Langevin noise and generally require exponential runtime [CLRW24, SWN23, KZC+24]. We can easily obtain a bound of the former type by the same arguments; however, none of these approaches are desirable as we aim to achieve polynomial iteration complexity for general nonconvex problems, for which little is known besides the recent work of [GWB25], which requires a local strong convexity assumption as well as other highly technical conditions. Nonetheless, we agree that studying the particle discretization is a natural and important direction of future work.
>
> **(Q3) Given the cost of sampling from the Gaussian process, are there efficient approximations that preserve the escape guarantees?**
>
> One natural strategy is to approximate the Hessian stochastically, for instance by computing it using a subsample of the data or particles. In the finite-dimensional setting, perturbation-based methods near saddle points have been extended to SGD [L19]. Under suitable assumptions on the variance, such stochastic approximations may retain the desired escape guarantees.
>
> In addition, in finite dimensions, the Hessian-vector product can be used to run negative curvature descent methods [LY17,CR19] without computing the full Hessian inverse as in the usual second-order methods. Such methods can also be converted into first-order algorithms such as Neon2 [AZL18]. These ideas may be leveraged to develop a more efficient or even fully first-order version of PWGF.
>
> [GWB25] Margalit Glasgow, Denny Wu and Joan Bruna. Propagation of Chaos in One-hidden-layer Neural Networks beyond Logarithmic Time. arXiv:2504.13110, 2025.
>
> [MMN18] Song Mei, Andrea Montanari, and Phan-Minh Nguyen. A mean field view of the landscape of two-layer neural networks. Proceedings of the National Academy of Sciences, 115(33):E7665–E7671, 2018.
>
> [MMM19] Song Mei, Theodor Misiakiewicz, and Andrea Montanari. Mean-field theory of two-layers neural networks: dimension-free bounds and kernel limit. COLT, 2019.
>
> [DBDFS20] Valentin De Bortoli, Alain Durmus, Xavier Fontaine, and Umut Simsekli. Quantitative propagation of chaos for sgd in wide neural networks. NeurIPS, 2020.
>
> [CLRW24] Fan Chen, Yiqing Lin, Zhenjie Ren, and Songbo Wang. Uniform-in-time propagation of chaos for kinetic mean field langevin dynamics. Electronic Journal of Probability, 29:1–43, 2024.
>
> [SWN23] Taiji Suzuki, Denny Wu, and Atsushi Nitanda. Convergence of mean-field langevin dynamics: Time and space discretization, stochastic gradient, and variance reduction. NeurIPS, 2023.
>
> [KZC+24] Yunbum Kook, Matthew S Zhang, Sinho Chewi, Murat A Erdogdu, and Mufan (Bill) Li. Sampling from the mean-field stationary distribution. arXiv preprint arXiv:2402.07355, 2024.
>
> [AZL18] Zeyuan Allen-Zhu and Yuanzhi Li. NEON2: Finding Local Minima via First-Order Oracles. NeurIPS, 2018.
>
> [CGT11a] Cartis, C., Gould, N.I.M., Toint, Ph.L. Adaptive cubic regularisation methods for unconstrained optimization. Part I: Motivation, convergence and numerical results. Math. Program. 127, 245–295, 2011.
>
> [CGT11b] Cartis, C., Gould, N.I.M., Toint, Ph.L. Optimal Newton-type methods for nonconvex smooth optimization problems. Technical report ERGO 11-009, School of Mathematics, University of Edinburgh.
>
> [CR19] F. E. Curtis and D. P. Robinson. Exploiting negative curvature in deterministic and stochastic optimization. Mathematical Programming: Series A and B, Volume 176, Issue 1-2, 69-94, 2019.
>
> [CRS17] F. E. Curtis, D. P. Robinson, and M. Samadi. A trust region algorithm with a worst-case iteration complexity of $o(\epsilon^{−3/2})$ for nonconvex optimization. Mathematical Programming, 162(1-2):1–32, 2017.
>
> [L19] Zhize Li. SSRGD: Simple Stochastic Recursive Gradient Descent for Escaping Saddle Points. NeurIPS, 2019.
>
> [LY17] Mingrui Liu, Tianbao Yang. On Noisy Negative Curvature Descent: Competing with Gradient Descent for Faster Non-convex Optimization. arXiv:1709.08571, 2017.

---

> > ### Comment · Reviewer_FN5C · 2025-08-05
> >
> > Thank you for the response. I keep my positive score.

---

### Official Review · Reviewer_LaL6 · 2025-07-23

**Clarity:** 4
**Significance:** 3
**Originality:** 3
**Rating:** 5
**Confidence:** 4

**Summary:**

The paper performs a theoretical analysis of escaping saddle points via perturbation in Wasserstein-2 space analogous to Jin et al. 2017. The key here is selecting a hessian guided kernel in the gaussian process.

**Questions:**

1. For the discrete time analysis, what if we use backward euler instead of forward as in JKO-scheme? What would the algorithm be like and can we get similar convergence?

2. Why is it important to escape saddle points in WGF? Is there a particular practical scenario that this makes a major issue or do you have any intuition on when saddle points occur frequently?

3. Do you have any intuition on the choice of the kernel? Could you explain maybe by showing the analogy in euclidean space?

**Ethical Concerns:**

["NO or VERY MINOR ethics concerns only"]

**Limitations:**

yes

**Quality:**

3

**Strengths And Weaknesses:**

Strengths

The analysis is very neat and the writings are very structural. Although I cannot check every proof, I think key theoretical results are clear and valid.


Weaknesses

Just like WGF, from an algorithmic perspective, this is conceptual rather than empirical. The specific implementation depends a lot on what functional you use and how well it is estimated. I would suggest adding more empirical demonstrations if possible just to show the potential empirical values. Maybe using MMD as the functional could be easier.

---

> ### Author Rebuttal · Authors · 2025-07-29
>
> Thank you for the thorough review and positive evaluation of our contributions! Our responses are as follows.
>
> **(Weakness) I would suggest adding more empirical demonstrations if possible just to show the potential empirical values.**
>
> We have **added new experiments and a detailed landscape analysis** of the matrix factorization problem using a mean-field neural network (Example 1) in Appendix F.2 and G which suggest the presence of strict benignity and show that noise injection helps escape the initial critical point. Since NeurIPS this year does not allow updating the submission pdf nor adding anonymous links during rebuttal, we summarize the results in text here.
>
> * Experimental details: The input and output dimensions are $l=15,k=5$. We approximate the measure using $400$ neurons and generate $800$ i.i.d. input data points $z$ from the standard normal distribution $\mathcal{N}(0,1)$ for each coordinate. Similarly to the ICFL case, SGD was used in the optimization process and the learning rate $\eta = 10^{-6}$. We set $k_{\mathrm{thres}} = 100, ~ \eta_p = 3\times 10^{-3}, ~ F_{\mathrm{thres}} = 10^{-2}$. We report the mean and standard deviation over 10 runs, with the corresponding error bars.
> * We find that both Hessian and isotropic noise injection methods achieve faster objective reduction and exhibit earlier peaks in the gradient norm, demonstrating a more efficient escape from the initial critical point. In contrast, the perturbation-free method tends to stagnate for longer periods. In particular, the Hessian method shows the best performance, although the performance of isotropic noise is comparable.
>
> **(Q1) For the discrete time analysis, what if we use backward euler instead of forward as in JKO-scheme? What would the algorithm be like and can we get similar convergence?**
>
> Unfortunately, it seems difficult to extend the analysis to backward Euler in the discrete time setting. The discrete-time analysis is discussed in Appendix E. Using the definition of the JKO scheme, we can derive a result similar to Proposition E.5. However, it is difficult to establish Proposition E.6, which is key to the proof of saddle point escape. This is because the transport map $Y^{(k)}$ (defined by $\mu^{(k)} = Y^{(k)} \sharp \mu_0$) is generally not available in implicit schemes such as the backward method, and unless $Y^{(k)}$ is expressed explicitly, it is difficult to carry out any Hessian-based arguments.
>
> In terms of stability, implicit schemes such as the JKO scheme are generally considered to be more stable than the explicit scheme used in this paper. However, for the class of smooth objective functionals we consider, stability does not pose a problem, and the theoretical analysis remains valid.
>
> **(Q2) Why is it important to escape saddle points in WGF? Is there a particular practical scenario that this makes a major issue or do you have any intuition on when saddle points occur frequently.**
>
> Classical optimization problems such as matrix factorization and tensor decomposition are known to be affected by saddle points. These problems arise in practical scenarios such as image reconstruction, object detection via matrix sensing, and recommendation systems [ZHBL10, ZKLR13, Y08]. In Euclidean space, the landscape of the objective function for matrix factorization has been extensively studied [GHJY15, GJZ17]. In particular, it is known that saddle points exist, and that all non-saddle stationary points are global optima for these types of objectives. A major concern in such situations is that gradient updates can get trapped near saddle points. For example, passing through neighborhoods of saddle points using gradient-based methods can take arbitrarily long time [DJL+17]. Therefore, escaping saddle points is crucial in such settings.
>
> It is highly likely that analogous problems in Wasserstein space (e.g. mean-field formulations of the above problems) suffer from the same issues, as we demonstrated in our new neural matrix factorization analysis and numerical experiments. Moreover, recent distributional landscape analyses (including the MMD and ICFL settings introduced in the paper) show that new types of saddle points or benignity properties can manifest in fundamentally distributional problems. On the other hand, existing mean-field studies have been mostly constrained to the linearly convex setting, such as shallow neural networks; our secondary goal is to more deeply study second-order behavior of WGF to understand wider families of deep learning methods.
>
> **(Q3) Do you have any intuition on the choice of the kernel? Could you explain maybe by showing the analogy in euclidean space?**
>
> The intuition is that the kernel is chosen so that points move in directions where the objective function is unstable and tends to decrease. At a saddle point, the minimal eigenvalue of the Hessian is negative, and the objective function is locally concave along the corresponding eigenvector direction. For example in Euclidean space, one can look at the function $f(x,y) = x^2 - y^2$ around the origin, where the $-y$ direction exhibits this concavity. Therefore, by using noise to perturb the point in that direction, we aim to move it where the gradient is steep and the objective function decreases. In the above example, we would want the noise to contain the $(0,1)$ direction, which is the eigenvector corresponding to the negative eigenvalue of $\text{Hess}(f)$.
>
> In finite $d$-dimensional settings, it is known that adding isotropic noise is sufficient and $\text{polylog}(d)$ updates suffice to escape via a geometric argument: the region of points which fail to escape is shaped like a pancake and thus has small volume. However, in infinite-dimensional curved spaces, this naive argument does not work and the noise must be modified into a more tractable form. To this end, we consider Gaussian noise defined via a Hessian-based kernel. This "directed" Gaussian noise is fundamentally connected to the eigenstructure of the Hessian as discussed in Appendix D, and faithfully reflects the above intuition even in Wasserstein space.
>
> **References**
>
> [ZHBL10] Bo Zhao, Justin P. Haldar, Cornelius Brinegar, Zhi-Pei Liang. Low rank matrix recovery for real-time cardiac MRI. In 2010 IEEE International Symposium on Biomedical Imaging: From Nano to Macro, pp.996–999. IEEE, 2010.
>
> [ZKLR13] Wenbin Zou, Kidiyo Kpalma, Zhi Liu, Joseph Ronsin. Segmentation driven low-rank matrix recovery for saliency detection. In 24th British machine vision conference (BMVC), pp.1–13, 2013.
>
> [Y08] Koren, Yehuda. Factorization meets the neighborhood: a multifaceted collaborative filtering model. In Proceedings of the 14th ACM SIGKDD international conference on knowledge discovery and data mining. p.426-434. 2008.
>
> [GHJY15] Rong Ge, Furong Huang, Chi Jin, Yang Yuan. Escaping From Saddle Points—Online Stochastic Gradient for Tensor Decomposition. COLT, 2015.
>
> [GJZ17] Rong Ge, Chi Jin, Yi Zheng. No Spurious Local Minima in Nonconvex Low Rank Problems: A Unified Geometric Analysis. ICML, 2017.
>
> [DJL+17] Simon S Du, Chi Jin, Jason Lee, Michael I Jordan, Aarti Singh, Barnabas Poczos. Gradient Descent Can Take Exponential Time to Escape Saddle Points. NeurIPS, 2017.

---

### Decision · Program_Chairs · 2025-09-17

**Decision:**

Accept (poster)

**Comment:**

The reviewers found the contribution solid and were positive about the paper. They also made several suggestions that can improve the paper, such as discussing the motivation of the work and including more numerical results. Please take the reviewers' suggestions into account as you prepare the camera-ready version of the paper.